# Met1-specific motifs conserved in OTUB subfamily of green plants enable rice OTUB1 to hydrolyse Met1 ubiquitin chains

Lining Lu [1,2,9] ✉, Xiaoguo Zhai [1,9], Xiaolong Li [3,9], Shuansuo Wang [4,5,9], Lijun Zhang[1,9], Luyang Wang[1,9], Xi Jin[1,9], Lujun Liang [2], Zhiheng Deng[2], Zichen Li[2], Yanfeng Wang [3], Xiangdong Fu [5], Honggang Hu [2,6], Jiawei Wang [7], Ziqing Mei [8] ✉, Zhengguo He[1] ✉ & Feng Wang [3] ✉

Linear (Met1-linked) ubiquitination is involved inflammatory and innate immune signaling. Previous studies have characterized enzymes regulating the addition and removal of this modification in mammalian systems. However, only a few plant-derived deubiquitinases targeting Met1-linked ubiquitin chains have been reported and their mechanism of action remains elusive. Here, using a dehydroalanine-bearing Met1-diubiquitin suicide probe, we discover OTUB1 from *Oryza sativa* (OsOTUB1) as a Met1-linked ubiquitin chain-targeting deubiquitinase. By solving crystal structures of apo OsOTUB1 and an OsOTUB1/Met1-diubiquitin complex, we find that Met1 activity is conferred by Met1-specific motifs in the S1' pocket of OsOTUB1. Large-scale sequence alignments and hydrolysis experiments provide evidence that these motifs are a general determinant of Met1 activity in the OTUB subfamily across species. Analysis of the species distribution of OTUBs capable of hydrolysing Met1-linked ubiquitin chains shows that this activity is conserved in green plants (*Viridiplantae*) and does not exist in metazoans, providing insights into the evolutionary differentiation between primitive plants and animals.

Ubiquitin can be attached to either one of the seven lysine residues (Lys6, Lys11, Lys27, Lys29, Lys33, Lys48, and Lys63) or the N-terminal methionine (Met1) of another ubiquitin, allowing the formation of distinct ubiquitin chain configurations[1], and different types of ubiquitin chains confer functional diversity to their linking proteins[2]. Unlike

ubiquitin chains linked with isopeptide bonds, linear ubiquitin chains (also known as Met1-linked chains, referred to hereafter as the "Met1 chains") are connected by peptide bonds, the construction of which is catalysed by the linear ubiquitin chain assembly complex (LUBAC) in mammalian cells[3–6]. Met1 chains promote the activation of NF-κB

[1]State Key Laboratory for Conservation and Utilization of Subtropical Agro-bioresources, College of Life Science and Technology, Guangxi University, Nanning 530004, PR China. [2]Tsinghua-Peking Center for Life Sciences, Key Laboratory of Bioorganic Phosphorus Chemistry & Chemical Biology (Ministry of Education), Center for Synthetic and Systems Biology, State Key Laboratory of Chemical Oncogenomics (Shenzhen), Department of Chemistry, Tsinghua University, Beijing 100084, PR China. [3]Key Laboratory of Molecular Medicine and Biotherapy in the Ministry of Industry and Information Technology, Department of Biology, School of Life Science, Beijing Institute of Technology, Beijing 100081, PR China. [4]Shanxi Key Laboratory of Minor Crops Germplasm Innovation and Molecular Breeding, Shanxi Agricultural University, Taiyuan 030031, PR China. [5]The State Key Laboratory of Plant Cell and Chromosome Engineering, Institute of Genetics and Developmental Biology, Chinese Academy of Sciences, Beijing 100101, PR China. [6]Institute of Translational Medicine, Shanghai University, Shanghai 200444, PR China. [7]State Key Laboratory of Membrane Biology, Tsinghua University, Beijing 100084, PR China. [8]School of Chemistry and Biological Engineering, University of Science and Technology Beijing, Beijing 100083, PR China. [9]These authors contributed equally: Lining Lu, Xiaoguo Zhai, Xiaolong Li, Shuansuo Wang, Lijun Zhang, Luyang Wang, Xi Jin. ✉e-mail: thu20151@sina.com; Marina1977@163.com; hezhengguo2019@163.com; wfeng@bit.edu.cn

signalling and autophagy by recruiting related downstream factors[4,7]. Deubiquitinases (DUBs) catalyse the hydrolysis of Met1 chains to negatively regulate the ubiquitination process, often modulating cell signalling. OTULIN (OTU deubiquitinase with linear linkage specificity) and CYLD (CYLD lysine 63 deubiquitinase) are the only two types of DUBs that have been reported to hydrolyse the Met1 chains in mammalian cells[8]. More recently, RavD from a pathogenic microorganism (*Legionella pneumophila*) was also observed to specifically depolymerize Met1 chains, weakening the host immune response[9].

The S1 and S1' sites in DUBs are responsible for their specificity for the poly-Ub linkage: the S1 site guides the C-terminus of the distal ubiquitin to the active centre, while the S1' site determines linkage selectivity by accommodating the distinct proximal Ub moiety. In some cases, the inactive configurations of S1 and/or S1' rearrange and become active upon the binding of Ub substrate[10–14]. OTULIN engages in substrate-assisted catalysis wherein the proximal Ub of Met1-diub triggers catalytic triad rearrangement for activation. In contrast, the catalytic centre and the overall structure of RavD do not undergo visible conformational changes upon binding to Met1-diUb, leading to the conclusion that the full activity of RavD is not predicated on substrate assistance[9]. In plants, physiological roles of Met1 chains and directly correlated DUBs have rarely been reported[15–17]. With the exception of *Arabidopsis*, no DUBs digesting Met1 chains have been reported in other plants. Importantly, how plant DUBs selectively recognize and hydrolyse Met1 linkages is still unclear.

In this work, we report the identification of OsOTUB1, a homologue of human OTUB1 (hOTUB1) capable of hydrolysing Met1 chains (Met1 activity), by the screening of DUBs in *Oryza sativa* (rice) using a dehydroalanine (DHA)-bearing Met1-diUb probe. OsOTUB1 is the first DUB known to deubiquitylate Met1 chains to be identified in *rice*. Structural evidence demonstrates that the Met1-specific motifs (containing the N-handle motif and C-handle motif) in the S1' pocket of OsOTUB1 confer Met1 activity, probably by permitting the S1' pocket to accommodate the proximal Ub in Met1-diUb. Furthermore, large-scale consensus sequence alignments and species distribution analysis imply the prevalence of Met1-specific motifs in the OTUB subfamily across species. The conservation of these motifs in plants other than animals might provide a novel molecular criterion for evolutionary classification, distinguishing primitive plants from animals.

## Results

### The discovery of plant DUBs that hydrolyse Met1 chains using an efficiently prepared Met1-diUb suicide probe

To search for plant-based DUBs capable of cleaving Met1 chains, we designed and prepared a Met1-diUb active probe and used it to retrieve DUBs from northern japonica *rice* Zhonghua 11 (ZH11), as described previously[18]. Weber et al. reported an activity-based Met1-diUb probe containing DHA used for in vitro and in vivo functional analysis of OTULIN[19]. In this work, the authors prepared a biotinylated probe by total chemical synthesis and obtained an N-terminally His-tagged probe by combining bacterial protein expression with chemical synthesis. Inspired by their work, we wondered whether we could directly acquire the biotinylated probe in one pot, which would eliminate multiple time-consuming procedures including chemical synthesis, HPLC purification, freeze drying, refolding, etc. We developed a one-pot strategy capable of delivering approximately 100 mg of the probe within 10 h (Fig. 1a). First, we recombinantly expressed and purified Met1-diUb bearing the N-terminal Avi-tag (GLNDIFEAQKIEWHE)[20] and incorporating a Gly76Cys mutation at the distal ubiquitin (Avi-Met1-diUb (G76C$^{distal}$)). Then, the pH value of a solution of purified Avi-Met1-diUb (G76C$^{distal}$) was adjusted to 9.5 using a mixture of 50 mM Tris•HCl and 150 mM NaCl. Subsequently, the biotin ligase BirA and 2,5-dibromohexanediamide, an alkaline elimination reagent, were incubated with Avi-Met1-diUb (G76C$^{distal}$) at 37 °C. BirA specifically added biotin to Lys of Avi-tag, while 2,5-

dibromohexanediamide desulfurized Cys76 of Avi-Met1-diUb (G76C$^{distal}$) into DHA[21,22]. After nearly 8 h, Avi-Met1-diUb (G76C$^{distal}$) was almost 100% transformed into the biotinylated Met1-diUb probe bearing DHA (biotin-Avi-Met1-diUb-DHA) (Supplementary Fig. 1a, Methods). To verify the efficacy of biotin-Avi-Met1-diUb-DHA, we tested its crosslinking activity with the catalytic domain of OTULIN (OTULIN-cat, residues 80-352) over a time gradient (Supplementary Fig. 1c). The results showed that biotin-Avi-Met1-diUb-DHA crosslinked with nearly 90% of the OTULIN-cat in 20 min, indicating that biotin-Avi-Met1-diUb-DHA has an efficient crosslinking ability. Then, the crosslinked product (biotin-Avi-Met1-diUb-DHA-OTULIN-cat) was purified, and its binding ability was assessed with streptavidin beads (Supplementary Fig. 1d). The results showed that streptavidin beads efficiently enriched biotin-Avi-Met1-diUb-DHA-OTULIN-cat after 30 min of incubation. In summary, biotin-Avi-Met1-diUb-DHA is well-suited to screen for plant DUBs with Met1 activity.

To clarify whether DUBs with Met1 activity exist in the plant kingdom, we used biotin-Avi-Met1-diUb-DHA to retrieve DUBs from ZH11 in the seedling stage (14 days). *Rice* seedlings were milled in liquid nitrogen, and their proteins were dissolved in lysis buffer. After centrifugation, biotin-Avi-Met1-diUb-DHA was incubated with the supernatant at 37 °C to facilitate a crosslinking reaction. After enrichment by streptavidin beads and extensive washing, the proteins on beads were denatured in protein loading dye by heating and then separated by SDS-PAGE. Finally, each band on the gel was cut and degraded by trypsin. The digested peptides were separated by HPLC and analysed by Orbitrap-MS/MS. Finally, the MS/MS spectra were searched against rice.fasta downloaded from UniProt. The search criteria used were a fixed modification of carbamidomethyl (C); a variable modification of oxidation (M); precursor ion mass tolerances of 20 ppm, and a fragment ion mass tolerance of 0.02 Da. The peptide false discovery rate (FDR) was calculated using Percolator provided by Proteome Discoverer (PD). Peptide spectrum matches (PSM) were assumed to be correct for q values lower than 1%. FDR was determined based on PSMs when searched against the reverse decoy database. Peptides only assigned to a given protein group were considered unique. In two independent experiments, five ubiquitin-related proteins were identified with a score≥10, including 3 DUBs (OsUCH14, OsOTUB1 and OsUCH-L), OsE1 and OsPolyUb (Fig. 1c). Coincidentally, these ubiquitin-related proteins are related to those found in human cells by Weber[19]. For example, the human-derived USP5, UCHL3, E1 and PolyUb identified by Weber correspond to OsUCH14, OsUCH-L, OsE1 and OsPolyUb, respectively[19]. The mass spectrometry proteomics data have been deposited to the ProteomeXchange Consortium via the PRIDE[23] partner repository with the dataset identifiers PXD032822 and 10.6019/PXD032822. Here, we noticed that in addition to DUBs, PolyUb and E1 were both detected in Weber's work and our study. Mass spectrometry can easily and inevitably detect Ub peptides because the Met1-diUb probe covalently crosslinks the target protein. The E1 enzyme can bind and activate not only the C-terminus of ubiquitin but also the proximal Ub in the ubiquitin chains[24,25]. Therefore, it is reasonable that the Met1-diUb probe can bind and extract the E1 enzyme in the lysate.

Among the three DUB hits, the human homologues of OsUCH14 and OsUCH-L are USP5 and UCHL3, respectively. Previous studies have demonstrated that USP5 can hydrolyse 6 types of ubiquitin chains, including Met1, without any linkage selectivity[26,27]. UCHL3 mainly removes the C-terminal small peptide conjugates of ubiquitin or short C-terminal extensions of polymeric ubiquitin precursors[28]. Our in vitro experiments confirmed that OsUCH14 and OsUCH-L have no apparent linkage preference (Supplementary Fig. 2a). To discover more DUBs with Met1-type linkage selectivity, OsUCH14 and OsUCH-L were not considered for further study. OsOTUB1 is the human homologue of hOTUB1, which has been proven to strictly digest Lys48-type ubiquitin

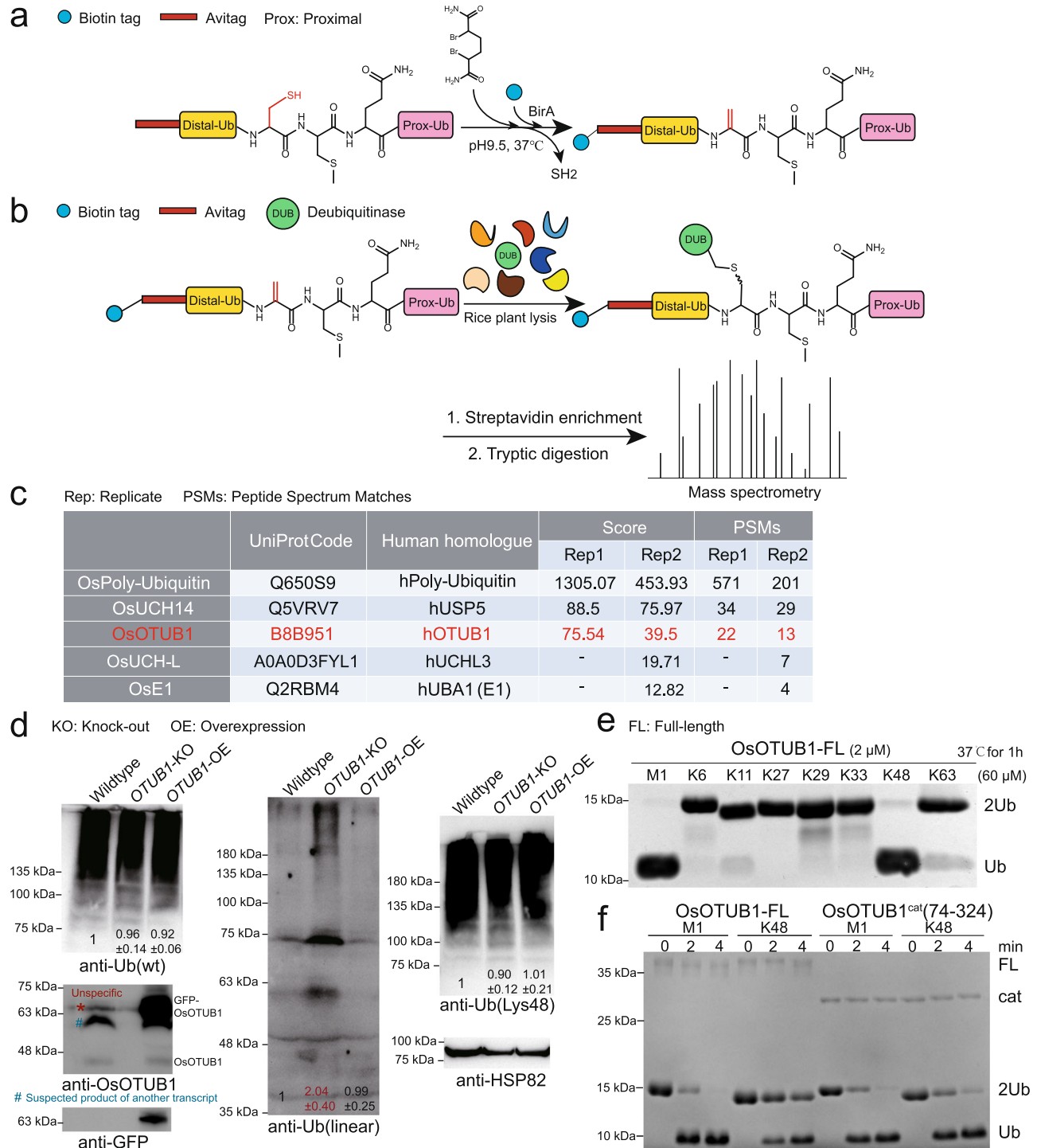

**Fig. 1 | Identification of OsOTUB1 and depolymerization activity test in vivo and in vitro. a** The one-pot synthesis of Biotin-Avi-Met1-diUb-DHA; **b** The identification of DUBs that hydrolyze the Met1 chain in *Oryza Sativa* using Biotin-Avi-Met1-diUb-DHA; **c** Ubiquitin related hits by mass spectrometry (*N* = 2 biologically independent experiments); **d** Effects of *OsOTUB1* over-expression and knockout on the abundance of mono-Ub, Met1- and Lys48 chains, and OsOTUB1 at the seedling stage, the numbers indicate the relative amount to the wildtype *rice* (defined to 1) based on three replicates monitored by ImageJ (*N* = 3 biologically independent experiments),

\* indicates the unspecific band, # indicates the suspected product of another transcript; **e** Hydrolysis activity against 8 linkage-type diUbs by recombinantly expressed full-length OsOTUB1 (*N* = 3 biologically independent experiments); **f** Time-scale hydrolysis activity of OsOTUB1 (full-length and the catalytic domain) against M1- and K48-diUb (*N* = 3 biologically independent experiments). KO, knock out; OE, overexpression; FL, full-length; cat, catalytic. Source data are provided as a Source Data file.

chains but not Met1-type ubiquitin chains[29]. The unusual selectivity of OsOTUB1 for the Met1 type aroused our interest.

To confirm the Met1 activity of OsOTUB1 in vivo, we compared the differences in the abundance of Met1 chains between *OsOTUB1* knockout and *OsOTUB1* overexpressing transgenic seedlings. In the

overexpression *rice*, *GFP* was C-terminally expressed with *OsOTUB1*. In the use of antibodies against OsOTUB1 and GFP, the western blot results showed that (Fig. 1d) the expression of OsOTUB1 was significantly elevated and completely absent in the overexpression and knockout seedlings, respectively. An antibody against Met1 chains was

used to analyse the abundance of Met1 chains in the seedlings, and an antibody against Ub was used to check whether the total amount of Ub was affected (Fig. 1d). The band intensity was quantified by ImageJ software[30]. The results showed that the intensities of the bands corresponding to Ub in wild-type, depleted, and overexpressed *OsOTUB1* seedlings were almost the same, indicating that OsOTUB1 does not affect the total Ub in *rice*. Using the Met1 chain antibody, the intensity of the band corresponding to Met1 chains was significantly increased in *OsOTUB1* knockout seedlings, which was approximately 204% that of the WT, indicating that the activity of OsOTUB1 contributes to the digestion of Met1 chains in vivo. In contrast, the *OsOTUB1*-overexpressing seedlings showed no obvious decrease in Met1 chains (99% of that of WT), the reason for which is under further investigation. However, the effect of OsOTUB1 on Lys48 chains was seemingly not significant, since the abundance of Lys48 chains in both *OsOTUB1* knockout and overexpression seedlings was similar to that of the WT (Fig. 1d). Interestingly, the effects of OsOTUB1 on Met1 and Lys48 chains became insignificant at the young panicle differentiation stage. Based on three replicates, there were no obvious differences among the WT, *OTUB1*-KO and *OTUB1*-OE lines for both Met1 and Lys48 chains (Supplementary Fig. 2b).

To further confirm the Met1 activity of OsOTUB1 in vitro, we recombinantly expressed full-length OsOTUB1 (FL-OsOTUB1) and assessed its hydrolytic activity against eight types of diUb (Fig. 1e). In this assay, 2 μM FL-OsOTUB1 and 60 μM diUbs were mixed and incubated for 1 h at 37 °C, followed by SDS-PAGE separation and Coomassie brilliant blue staining. FL-OsOTUB1 was found to effectively hydrolyse Lys48 diUb (<90%) but not Lys63 diUb (<10%) under the same conditions. This is consistent with a previous finding of Wang et al., who proved that FL-OsOTUB1 preferentially cleaves Lys48 tetraUb[18]. Additionally, OsOTUB1 weakly digested Lys11 diUb (Fig. 1e). Unexpectedly, under the same conditions, FL-OsOTUB1 almost completely hydrolysed Met1-diUb. To further confirm the linkage preference of FL-OsOTUB1, we compared the time-scale activity differences of FL-OsOTUB1 against Met1- and Lys48-diUb (Fig. 1f). Under the same conditions, FL-OsOTUB1 hydrolysed 80% of Met1-diUb at 2 min but only approximately 30% of Lys48-diUb, further demonstrating that FL-OsOTUB1 preferentially hydrolysed Met1 chains. Notably, FL-OsOTUB1 appeared smeared on the gel (Fig. 1f, Supplementary Fig. 2c, e), suggesting that it may be prone to degrade into variants with similar molecular weights, thus challenging the following quantification and enzymatic essay – perhaps due to the instability of the flexible, glycine-rich region on the N-terminus of the full-length protein (Supplementary Fig. 3). Accordingly, we deleted the 73 residues at the N-terminus to obtain a truncated version of OsOTUB1 comprising only the sequence ranging from residues 74 to 324 but still incorporating the catalytic domain (OsOTUB1-cat). Biochemical experiments (Fig. 1f) showed that OsOTUB1-cat appeared on the gel as a single band, indicating that it was more stable. However, OsOTUB1-cat showed marginal preference between Met1- and Lys48-diUb (hydrolysing almost the same percentage of diUbs within 4 min, Fig. 1f), indicating that the N-terminal part of OsOTUB1-FL plays roles in discriminating Met1-diUb and Lys48-diUb, the mechanism of which needs further investigation. Since we aimed to uncover the mechanism by which OsOTUB1 hydrolyses the Met1 chains, OsOTUB1-cat was used in all subsequent work, unless otherwise stated. In addition, to ensure uniformity, all the following enzymatic digestion experiments were carried out at 37 °C, and the concentrations of DUB and Ub chains were 2 μM and 60 μM, respectively, unless otherwise noted.

In hOTUB1, the N-terminal peptide (A25-N45) forms an α helix that binds to and stabilizes the conformation of the proximal Ub in Lys48-diUb, thereby promoting cleavage by hOTUB1[31, 32], which was also confirmed by our data, both in the presence of hUBCH5B and OsUBCH5B (Supplementary Fig. 2c). Inspired by this phenomenon, we

asked whether the activity of OsOTUB1 towards Lys48-diUb can be promoted by OsUBCH5B or hUBCH5B. Therefore, the activity of OsOTUB1 against Lys48-diUb was monitored (molar ratio 1:30), in the presence of OsUBCH5B or hUBCH5B (25 μM). The results showed that (Supplementary Fig. 2c), in the absence of E2, approximately 30% of Lys48-diUb was digested by OsOTUB1 in 5 min. Under the same conditions, >50% and >90% of Lys48-diUb was digested in the presence of hUBCH5B and OsUBCH5B in 5 min, respectively, demonstrating that the activity of OsOTUB1 against Lys48-diUb can be promoted by both hUBCH5B and OsUBCH5B. However, the sequence comparison shows that (Supplementary Fig. 3) the N-terminal sequence of OsOTUB1 is significantly different from that of hOTUB1. In OsOTUB1, the N-terminal sequence is rich in residues that are prone to form random coils, such as Gly, Ser and Pro, suggesting that this region would not form an α helix to bind and stabilize the proximal Ub in Lys48-diUb. To reveal whether the activity promotion by E2 is dependent on the N-terminal sequence of OsOTUB1, the activity of OsOTUB1-cat (OsOTUB1 with deletion of the N-terminus) against Lys48-diUb was monitored in the presence of OsUBCH5B or hUBCH5B. The results showed that (Supplementary Fig. 2d), without E2, approximately 70% of Lys48-diUb was digested by OsOTUB1-cat in 4 min. While under the same conditions, >90% and almost 100% of Lys48-diUb was digested in the presence of hUBCH5B and OsUBCH5B in 4 min, respectively, demonstrating that stimulation of OsOTUB1 against Lys48-diUb by both hUBCH5B and OsUBCH5B is not (or not completely) dependent on the N-terminal sequence. The mechanism by which OsUBCH5B and hUBCH5B enhance the activity of OsOTUB1 needs to be further revealed. However, in sharp contrast, neither OsUBCH5B nor hUBCH5B affected OsOTUB1's activity against Met1-diUb (Supplementary Fig. 2e), implying that OsOTUB1 adopts a different way to interact with Met1-diUb and Lys48-diUb, similar to the conclusion demonstrated above that the N-terminal part of OsOTUB1 plays roles in discriminating Met1-diUb and Lys48-diUb (Fig. 1f).

## OsOTUB1-cat is likely induced to be active by distal Ub, but may not be activated by proximal Ub

To better understand the mechanism underlying the OsOTUB1-cat-mediated degradation of Met1 chains, the crystal structure of apo OsOTUB1-cat was first sought. OsOTUB1-cat crystals were grown in a reservoir containing 0.15 M potassium bromide and 30% w/v polyethylene glycol monomethyl ether 2000. Using the catalytic domain structure of hOTUB1 (hOTUB1-cat) (PDB code 2ZFY[29]) as the template, the crystal structure of OsOTUB1-cat at a resolution of 2.27 Å (PDB code 6K9N, Fig. 2a, Supplementary Table 1, Supplementary Fig. 4a) was finally obtained after molecular replacement and refinement. The overall structure of OsOTUB1-cat was found to comprise 11 α-helices (α) and 5 β-sheets (β) and bear an active region at the interface between the α3 helix and the β5 sheet separating the S1 and S1' pockets – almost identical to hOTUB1-cat (RMSD = 1.46 Å) (Fig. 2b). Catalytic Cys121 is located at the end of the C-terminus of the α3 helix (Fig. 2a). The imidazole ring of the catalytic His317 adopts a nearly vertical conformation with Cys121, and ε²N is 6.8 Å away from the sulfur atom of C121 – much farther than the distance between the catalytic His and Cys in the common DUBs (3-5 Å)[33, 34], and too far for ε²N to deprotonate the catalytic Cys. In summary, the catalytic triad configuration in OsOTUB1 is almost identical to that of hOTUB1-cat (Fig. 2b); therefore, the structural basis for their contrasting activities remains unknown.

To better understand how OsOTUB1-cat is activated and accomplishes the hydrolysis of Met1 chains, we attempted to resolve the crystal structure of OsOTUB1-cat crosslinked with Met1-diUb bearing DHA but not Avi (Met1-diUb-DHA) (OsOTUB1-cat-Met1-diUb-DHA). Met1-diUb-DHA was first prepared in a manner analogous to that used for the one-pot preparation of biotin-Avi-Met1-diUb-DHA and then incubated with OsOTUB1-cat at 37 °C, to allow the crosslinking reaction to proceed. Diffraction-quality crystals were finally obtained by

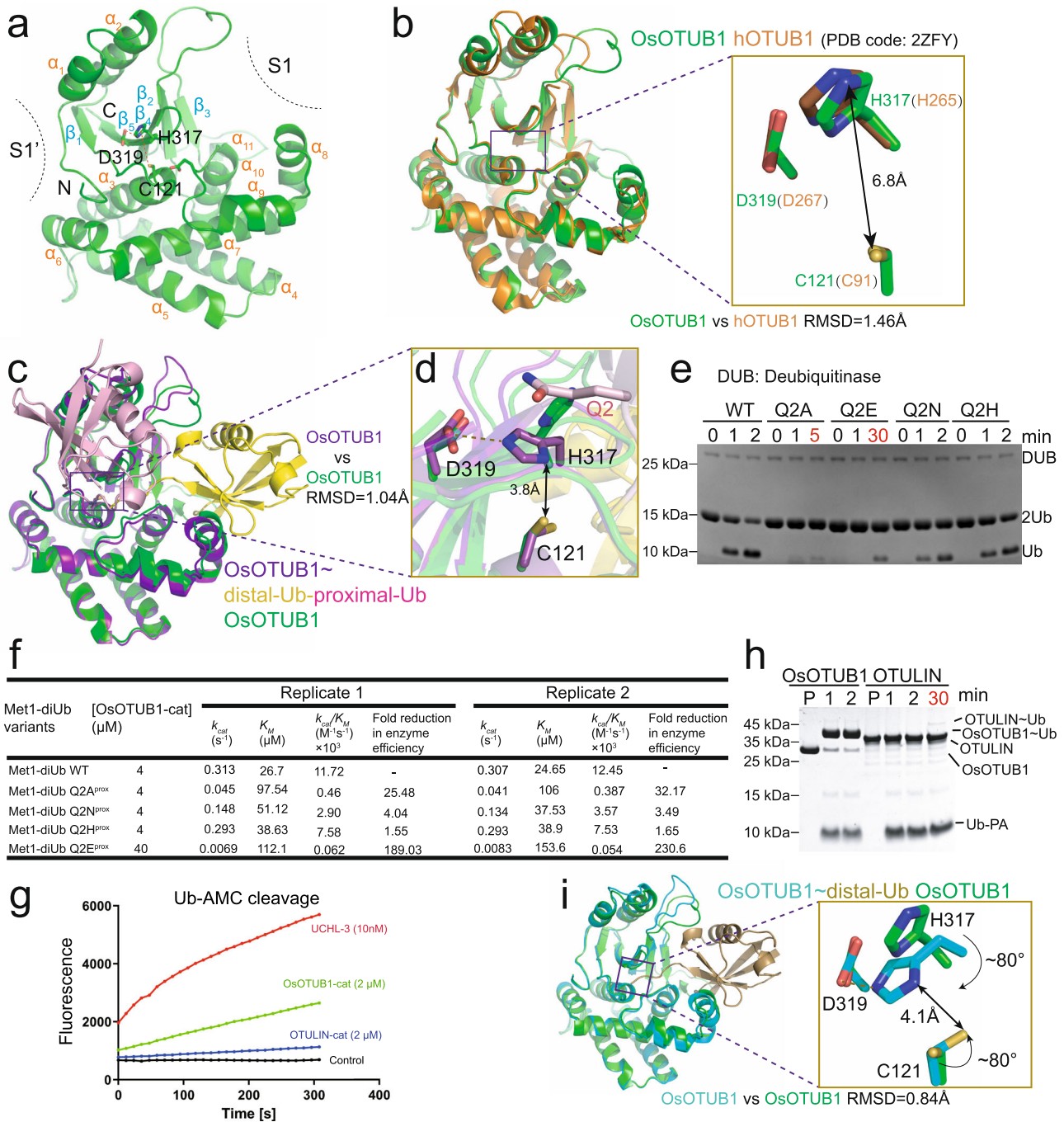

**Fig. 2 | OsOTUB1-cat is likely induced to be active by distal Ub, but may not be activated by proximal Ub. a** Crystal structure of OsOTUB1-cat; **b** Structural alignment between OsOTUB1-cat and hOTUB1-cat; **c** Crystal structure of OsOTUB1-cat-Met1-diUb-DHA and its alignment with OsOTUB1-cat; **d** Structural alignment between catalytic centers of OsOTUB1-cat and OsOTUB1-cat-Met1-diUb-DHA; **e** Effects of Q2 mutation in the proximal Ub on Met1-diUb hydrolytic activity ($N = 3$ biologically independent experiments); **f** Two replicates' kinetics data of OsOTUB1-cat against Met1-diUb variants ($N = 2$ biologically independent experiments, individual data shown as separate values); **g** Time-scale Ub-AMC cleavage ($N = 3$ biologically independent experiments); **h** Time scale Ub-PA crosslinking activity of OsOTUB1-cat and OTULIN-cat ($N = 3$ biologically independent experiments); **i** Crystal structure of OsOTUB1-cat-Ub-PA and its alignment with OsOTUB1-cat. Source data are provided as a Source Data file.

crystallization from 0.2 M sodium chloride, 0.1 M BIS-TRIS pH 5.5, and 25% w/v polyethylene glycol 3350. The structures of apo OsOTUB1-cat and Ub (PDB code 1UBQ) were used as templates. After molecular replacement and refinement, the complex crystal structure at a resolution of 2.34 Å (PDB code 6KBE, Fig. 2c, Supplementary Table 1, Supplementary Fig. 4b) was obtained.

The structures of OsOTUB1-cat-Met1-diUb-DHA and apo OsOTUB1-cat (Fig. 2d) were compared in an attempt to infer the mechanism for the activation of OsOTUB1-cat. We found that the γ

amide of Gln2 in the proximal Ub ($Q2^{prox}$) is parallel to the imidazole ring of the catalytic His317 in the complex. Additionally, the orientation of the imidazole ring changes from being perpendicular to the catalytic Cys121 to being parallel to it, reducing the distance between the sulfur atom of catalytic Cys121 and the imidazole ring of catalytic His 317 from 6.8 Å to 3.8 Å, low enough for the imidazole ring to deprotonate the thiol (Fig. 2d). This scenario suggests that the side chain of $Q2^{prox}$ may activate OsOTUB1 through a physical pushing of the imidazole ring of His317, relocating it from the inactive state to the

active state. To preliminarily explore this hypothesis, the effect of mutations at Q2$^{prox}$ on the activity of OsOTUB1-cat was screened. Here, Q2$^{prox}$ was mutated to residues with short side chains (here, Met1-diUb Q2A$^{prox}$ and Met1-diUb Q2N$^{prox}$), comparable-length side chains (here, Met1-diUb Q2H$^{prox}$) or negative charges since it might electrostatically stabilize the imidazole ring of His317 in the inactive state (here, Met1-diUb Q2E$^{prox}$), followed by enzymatic evaluation. The results showed that (Fig. 2e), compared with WT Met1-diUb (approximately 50% was hydrolysed after 1 min), Met1-diUb Q2A$^{prox}$ significantly impaired the activity of OsOTUB1-cat (merely negligible product after 5 min), Met1-diUb Q2N$^{prox}$ moderately reduced the activity of OsOTUB1-cat (approximately 10% was hydrolysed after 1 min), Met1-diUb Q2H$^{prox}$ almost recovered the activity of OsOTUB1-cat (approximately 30% was hydrolysed after 1 min), while Met1-diUb Q2E$^{prox}$ much more severely impeded the activity of OsOTUB1-cat (approximately 5% was hydrolysed even after 30 min). These results indicated that the activity of OsOTUB1-cat increases as the length of the side chain gradually extends, while it is much more significantly inhibited when the side chain bears a negative charge, seemingly supporting the hypothesis that Q2$^{prox}$ activates OsOTUB1-cat by relocating the imidazole ring of His317.

However, we still did not dissect the binding function and the catalytic role of Q2$^{prox}$. Therefore, we measured the kinetics of OsOTUB1-cat in the presence of Met1-diUb WT, Met1-diUb Q2A$^{prox}$, Met1-diUb Q2N$^{prox}$, Met1-diUb Q2H$^{prox}$ and Met1-diUb Q2E$^{prox}$. In this assay, the initial velocity of OsOTUB1-cat is represented by the generation of the product, which was monitored by high-performance liquid chromatography (HPLC). The initial amount of product was converted from the area of its peak traced by HPLC. Here, the relationship between the molar amount of product and its peak area in HPLC was established through a standard curve (Supplementary Fig. 5). The kinetics results from two replicates showed that (Fig. 2f), compared with OsOTUB1-cat in the presence of Met1-diUb WT, the turnover number ($k_{cat}$) of OsOTUB1-cat showed 1.05–1.07-fold, 2.11–2.29-fold, 6.96–7.49-fold and 36.99–45.36-fold reductions in the presence of Met1-diUb Q2H$^{prox}$, Met1-diUb Q2N$^{prox}$, Met1-diUb Q2A$^{prox}$ and Met1-diUb Q2E$^{prox}$, respectively. This is consistent with the activity of OsOTUB1-cat against different mutants (Fig. 2e), demonstrating that mutation of Q2$^{prox}$ significantly inhibits the Met1 activity of OsOTUB1-cat by reducing its $k_{cat}$. However, the affinity ($K_M$) of OsOTUB1-cat towards Met1-diUb mutants was not almost unchanged or elevated, as seen in OTULIN, which was previously demonstrated to be activated by proximal Ub[35], but was significantly reduced: compared to OsOTUB1-cat with the Met1-diUb WT, the affinity ($K_M$) of OsOTUB1-cat showed 1.45–1.58-fold, 1.52–1.91-fold, 3.65–4.30-fold and 4.20–6.23-fold reductions in the presence of Met1-diUb Q2H$^{prox}$, Met1-diUb Q2N$^{prox}$, Met1-diUb Q2A$^{prox}$ and Met1-diUb Q2E$^{prox}$, respectively. These results implied that mutations in Q2$^{prox}$ may affect the affinity of OsOTUB1-cat for the substrate, thereby altering the catalytic ability of OsOTUB1-cat. However, we still cannot completely rule out the possibility that Q2$^{prox}$ may play roles in activating OsOTUB1-cat due to the seemingly unparallel decrease between $k_{cat}$ and $K_M$, especially for Met1-diUb Q2E$^{prox}$ ($k_{cat}$ decreased 36.99–45.36-fold while $K_M$ elevated only 4.20–6.23-fold).

Given that the activity of some DUBs is induced by distal Ub[10,13,14], we examined the role of distal Ub in activating OsOTUB1-cat. Accordingly, we investigated whether OsOTUB1-cat could hydrolyse Ub-7-amino-4-methylcoumarin (Ub-AMC), since it was previously reported to be bound by the S1 pocket of hOTUB1[32], and the free coumarin generated through deubiquitination produces readily detectable fluorescence. The results showed that (Fig. 2g), under the action of OsOTUB1-cat, the fluorescence intensity of AMC continued to increase over time, demonstrating that OsOTUB1-cat effectively hydrolysed Ub-AMC. This result was similar to that conducted by UCHL3, which produced more free coumarin over time (Fig. 2g) and was previously demonstrated to be activated by distal Ub[10]. Simultaneously, we examined the ability of

OsOTUB1-cat to crosslink with C-terminally propargylated Ub (Ub-PA) (Fig. 2h). The results showed that OsOTUB1-cat crosslinked >90% Ub-PA in 1 min, consistent with the results of the Ub-AMC hydrolysis experiment. However, under the same conditions, OTULIN-cat, which also has Met1 activity, behaved differently: the fluorescence intensity generated by Ub-AMC hydrolysis was approximately equal to the background level (Fig. 2g), and only negligible Ub-PA crosslinked products were detected, even after 30 min (Fig. 2h). The contrasting activities of OsOTUB1-cat and OTULIN-cat against mono-Ub suggest that the activation of OsOTUB1-cat may be different from that of OTULIN-cat. Previous studies have reported that OTULIN activation relies solely on the proximal Ub, preventing it from hydrolysing Ub-AMCs or crosslinking with Ub-PA[35]. Therefore, we suspect that OsOTUB1-cat is activated by the distal Ub. To verify this, we resolved the crystal structure of OsOTUB1-cat crosslinked with Ub-PA (OsOTUB1-cat-Ub-PA). OsOTUB1-cat-Ub-PA was obtained by incubating OsOTUB1-cat with Ub-PA (Methods) at 37 °C. After purification, conventional crystallization conditions were screened, and crystals meeting the diffraction requirements were finally obtained using 0.2 M sodium chloride, 0.1 M HEPES pH 7.5, and 25% w/v polyethylene glycol 3350. Using structures of apo OsOTUB1-cat and Ub as templates, the complex structure was finally resolved at a resolution of 2.34 Å after molecular replacement and refinement (PDB code 6K9P, Fig. 2i, Supplementary Table 1, Supplementary Fig. 4c). Based on this structure, binding of the distal Ub was found to induce a deflection of approximately 80 degrees and a distance reduction from 6.8 Å to 4.1 Å between side chains of the catalytic Cys121 and His317, forming an activated conformation. However, given the condition differences between the crystal and solution, the snap shot of the crystal structure of apo OsOTUB1-cat that reflects the inactivation state (Fig. 2b) may not be that in solution. Therefore, the results proved that the distal Ub likely activates OsOTUB1-cat by inducing major conformational changes. In line with the conclusion that OsOTUB1-cat activity is likely induced by distal Ub but may not be induced by proximal Ub, we found that residues around the catalytic centre of OsOTUB1-cat-Ub-PA are already at almost the same positions as those in the structure of OsOTUB1-cat-Met1-diUb-DHA (Supplementary Fig. 6).

## Two motifs in the S1' pocket determine the Met1 activity of OsOTUB1-cat

As the residues binding to the proximal Ub of Met1-diUb in OsOTUB1 are also conserved in hOTUB1 (Supplementary Fig. 3), it appears that there might be an extra determinant to confer Met1 activity on OsO-TUB1 other than hOUTB1. To identify more binding sites for the proximal Ub of Met1-diUb in the S1' pocket, we analysed the key residues in OsOTUB1-cat involved in the interaction with Met1-diUb in depth. Based on the comparison of the primary sequences of hOTUB1 and OsOTUB1, the structural alignment between apo hOTUB1-cat and OsOTUB1-cat-Met1-diUb-DHA showed two main variations (Supplementary Fig. 9): 1) the P87 of hOTUB1-cat at the entrance of the catalytic centre is different from G117 in OsOTUB1-cat (Fig. 3a); and 2) the E63, D64 and D65 residues of hOTUB1-cat (EDD), located in loop between α2 and α3, are different from S93, G94 and S95 in OsOTUB1-cat (SGS) (Fig. 3b).

To examine the effect of these two motifs on the Met1 activity of OsOTUB1-cat, we first mutated the two motifs separately and simultaneously to the corresponding residues in hOTUB1-cat to obtain the separate mutants OsOTUB1-cat-G117P and OsOTUB1-cat-EDD and the double mutant OsOTUB1-cat-G117P-EDD and investigated their hydrolytic activities against Met1-diUb. The results showed (Fig. 3c) that mutation of any motif resulted in the detection of only negligible quantities of hydrolysis product even after 4 h, demonstrating that both motifs are essential for the Met1 activity of OsOTUB1-cat. To further examine the role of these two motifs, we mutated the two motifs of hOTUB1-cat separately and simultaneously to the corresponding residues in OsOTUB1-cat to obtain the

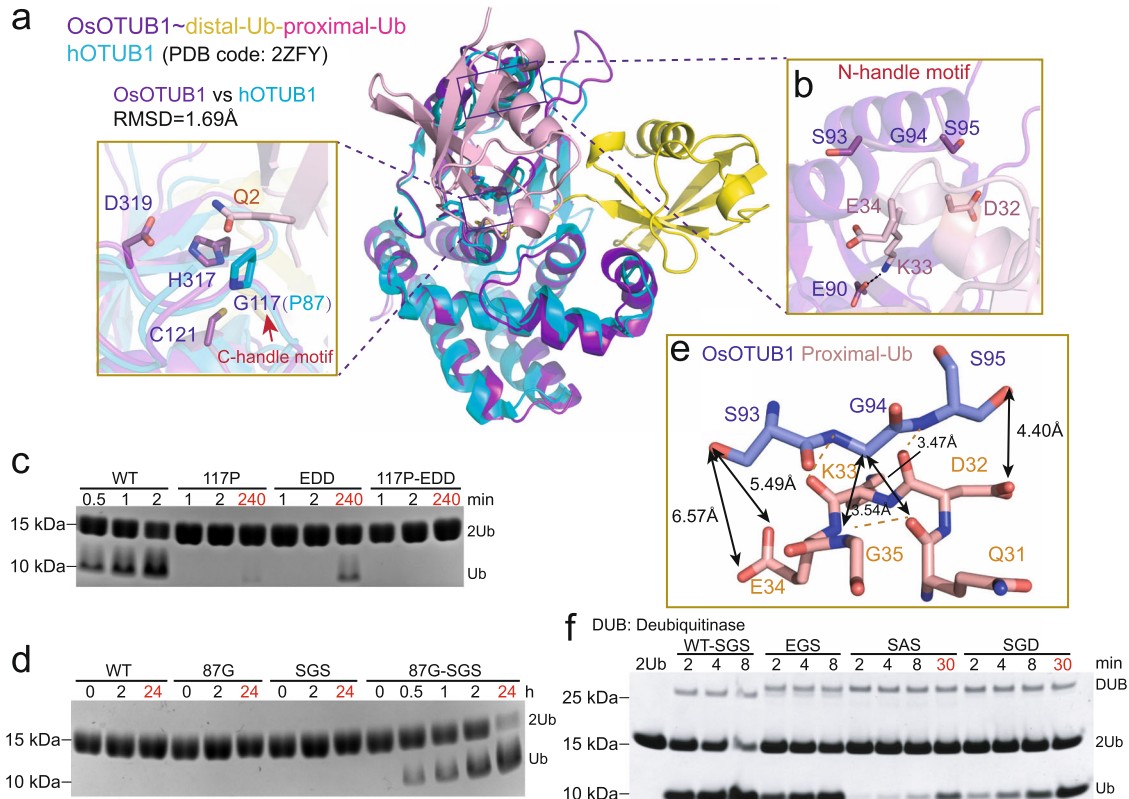

**Fig. 3 | Two motifs in S1' pocket determine Met1 linkage selectivity of OsOTUB1.**
**a** Structural alignment between C-handle motifs of OsOTUB1-cat-Met1-diUb-DHA and hOTUB1-cat; **b** Structural alignment between N-handle motifs of OsOTUB1-cat-Met1-diUb-DHA and hOTUB1-cat; **c** Effects of mutations in two motifs on the hydrolysis activity of OsOTUB1-cat against Met1-diUb ($N = 3$ biologically independent experiments); **d** Effects of mutations in two motifs on the hydrolysis activity of hOTUB1-cat against Met1-diUb ($N = 3$ biologically independent experiments); **e** Detail interaction between N-handle motif of OsOTUB1 and proximal Ub; **f** Effects of mutations in N-handle motif on the hydrolysis activity against Met1-diUb ($N = 3$ biologically independent experiments). Source data are provided as a Source Data file.

individual mutants hOTUB1-cat-P87G and hOTUB1-cat-SGS and the double mutant hOTUB1-cat-P87G-SGS and examined their hydrolytic activities against Met1-diUb. The results showed (Fig. 3d) that neither of the two single mutants could catalyse the formation of hydrolysis products, similar to WT hOTUB1-cat (no product was detected even after 24 h). However, under the action of hOTUB1-cat-P87G-SGS, bands corresponding to hydrolysates appeared on SDS-PAGE after 30 min (approximately 20% Met1-diUb), the intensities of which increased with time (approximately 50% Met1-diUb was digested after 2 h), indicating that mutations to both motifs are capable of conferring the Met1 activity of hOTUB1-cat. This result implies that hOTUB1 has the potential to cleave the Met1-type ubiquitin chain due to the conservation of residues responsible for binding to proximal Ub in Met1-diUb (Supplementary Fig. 3), while N- and C-handle motifs control this switch. Based on their order in the primary sequence, we named the SGS/EDD motif the N-handle motif, and the G/P motif the C-handle motif.

Based on these results, it is interesting to speculate that the ancestor of hOTUB1 (which may be similar to OsOTUB1 with broad linkage selectivity) should exhibit hydrolysis activity against both Met1- and Lys48 chains. However, under the pressure of natural selection, N- and C-handle motifs gradually evolved to distinct functional elements by gene duplication and mutation fixation. One of the derivatives, hOTUB1, evolved to an architecture that could not accommodate proximal Ub, eventually losing Met1 activity. However, the residue configuration in the S1' pocket responsible for proximal Ub binding did not change during this period. Once the entry restriction of proximal Ub into the S1' pocket is released (P87G-SGS), hOTUB1 can regain Met1 activity.

## The N- and C-handle motifs with XGY and G sequence modes, respectively, constitute the Met1-specific motif

To understand the mechanism by which these two motifs affect Met1 activity, we structurally analysed their role in the interaction between the S1' pocket and the proximal Ub. 1) Analysis of the C-handle motif. A key characteristic of Met1 chains is the repulsion between the bulky side chain of the Q2 residue of the proximal Ub and interacting proteins[35]. A similar repulsion cannot occur in OsOTUB1-cat since G117 at the entrance of the catalytic centre does not bear a side chain and cannot hinder the entry of the proximal Ub. However, because hOTUB1 bears the bulky and rigid P87 side chain, the proximal Ub is inaccessible, and therefore, it does not exhibit Met1 activity (Fig. 3a). 2) Analysis of the N-handle motif. The SGS of OsOTUB1-cat does not bear any charge in a neutral environment, while the EDD of hOTUB1-cat carries negative charges in a neutral environment. Analysis of the sequence of proximal Ub revealed that D32, K33 and E34 in proximal Ub (DKE) are adjacent to the SGS/EDD region. Given that the side chain of K33 is stabilized inside Ub, DKE would also bear a negative charge in a neutral environment, and therefore would be electrostatically repulsed by the EDD motif of hOTUB1-cat, resulting in the inactivity of hOTUB1-cat against Met1 chains.

To further clarify how the N-handle motif affects the binding of OsOTUB1-cat/hOTUB1-cat to proximal Ub and the extent to which this binding is dependent on electrostatic effects, we analysed the interaction of the N-handle motif (S93, G94, S95) in OsOTUB1-cat with proximal Ub (Fig. 3e). We found that 1) the distances between the hydroxyl group of the side chain in S93 and the carboxyl groups of the side chain in E34 are 5.49 Å and 6.57 Å, respectively. Such distances are much larger than the distance usually associated with

hydrogen bonds (≤4.0 Å); therefore, in OsOTUB1-cat, the introduction of even an acidic amino acid with a long side chain, such as glutamate (S93E), would still have no effect on Met1 activity. 2) The closest distance between the hydroxyl group of the side chain in S95 and the carboxyl group of the side chain in D32 is 4.40 Å, a distance expected to be conducive to the formation of a hydrogen bond. Thus, the introduction of an acidic amino acid with a short side chain, such as aspartic acid (S95D), would have a significant effect on Met1 activity. 3) The α C of G94 faces the loop structure of the proximal Ub, 3.47 Å and 3.54 Å from the carbonyl oxygen in the backbone of Q31 and the nitrogen in the backbone of G35, respectively. This sterically crowded environment seems to exclude the entry of any side chain groups, such as methyl. To verify the above conjectures, we separately introduced S93E, G94A and S95D into the N-handle motif of OsOTUB1-cat and examined the hydrolysis activity of these mutants against Met1-diUb. The results showed (Fig. 3f) that OsOTUB1-cat-S93E hydrolysed almost 50% of Met1-diUb at 4 min, comparable to that of OsOTUB1-cat (nearly 60% of Met1-diUb at 4 min) and indicating that the impact of the S93E mutation is negligible. OsOTUB1-cat-S95D hydrolysed almost 50% of Met1-diUb after only 30 min, indicating that the effect of the S95D mutation was significant. OsOTUB1-cat-G94A hydrolysed less than 20% of Met1-diUb even after 30 min, and is therefore much less active than OsOTUB1-cat, demonstrating the major influence of the G94A mutation on catalytic activity. Based on these results, we concluded that in the N-handle motif, the van der Waals-free nature of the side chain in G94 and the electrically neutral nature of the side chain in S95 are conducive to the development of Met1 activity, whereas S93 has a much smaller effect. Therefore, we can conclude that the Met1 activity-competent sequence in the N-handle motif is XGY, where X denotes any residue, and Y denotes any nonacidic residue.

To compare the effects of the N-handle and C-handle motifs on the Met1 activity of OsOTUB1-cat, we assessed the activity of OsOTUB1-cat-G117P, OsOTUB1-cat-EDD, and OsOTUB1-cat-G117P-EDD against Met1- polyUb in the presence of increased concentrations of enzyme and for prolonged hydrolysis periods (Supplementary Fig. 10a). Since long-chain polyUb was reported and confirmed by our preliminary assay to be more liable to DUB, thus readily facilitating comparisons of the importance of Met1 activity between the two motifs, Met1-tetraUb was used as the substrate. The results showed that, when the concentration of DUB was increased to 20 μM (DUB: Met1-tetraUb=4:3), OsOTUB1-cat-G117P hydrolysed much less Met1-tetraUb than OsOTUB1-cat-EDD over time, confirming the conclusion that the C-handle motif more significantly influences the Met1 activity of OsOTUB1-cat than the N-handle motif, consistent with the results shown in Fig. 3c.

Given that alterations in N-handle and C-handle motifs consistently alter Met1 activity for OsOTUB1 and hOTUB1 (Fig. 3c, d), we wondered whether the same alterations in N-handle and C-handle motifs would also have the same effect on Lys48 activity for OsOTUB1 and hOTUB1. The results showed that (Supplementary Fig. 10b), compared with hOTUB1-cat (approximately 25% Lys48-diUb was digested in 60 min), P87G significantly promoted the Lys48 activity of hOTUB1-cat (more than 90% Lys48-diUb was digested in 60 min), while SGS mutation obviously reduced the activity of hOTUB1-cat (less than 10% Lys48-diUb was digested in 60 min). Additionally, introduction of the P87G mutation (87G-SGS) could only provide minimal help (no more than 15% Lys48-diUb was digested in 60 min). These results indicated that P87G-EDD is more suitable for hOTUB1-cat Lys48 activity, while the activity impairment caused by the SGS mutation overwhelmed the advantage provided by the P87G mutation. For OsOTUB1-cat (Supplementary Fig. 10c), compared with OsOTUB1-cat (approximately 50% Lys48-diUb was digested in 2 min), G117P slightly weakened the Lys48 activity of OsOTUB1-cat (approximately 30% Lys48-diUb was digested in 2 min), while the EDD mutation obviously

reduced the activity of OsOTUB1-cat (less than 10% Lys48-diUb was digested in 2 min). Additionally, the introduction of the G117P mutation (117P-EDD) further inhibited Lys48 activity (negligible Lys48-diUb was digested in 2 min). These results indicated that both mutations (G117P and EDD) impeded the Lys48 activity of OsOTUB1-cat. Conclusively, the C-handle motif (G/P) contributes almost equally to the Lys48 activity of hOTUB1-cat and OsOTUB1-cat (G favours while P disfavours), while the effect of the N-handle motif (SGS/EDD) on hOTUB1-cat is opposite to that on OsOTUB1-cat: mutation from EDD to SGS can weaken hOTUB1-cat but enhance OsOTUB1-cat. These results indicate that hOTUB1-cat adopts a different strategy than OsOTUB1-cat to bind to and digest Lys48-diUb. However, the exact molecular mechanism needs further investigation. Conclusively, the N-handle motif and C-handle motif with XGY and G sequence modes, respectively, are Met1-specific but not Lys48-specific, which may result from the fact that OsOTUB1-cat and hOTUB1-cat-P87G-SGS adopt different strategies to bind to Met1-diUb and Lys48-diUb, probably due to the distinct configuration between the two linkage-type chains.

## The influence of the N-handle and C-handle motifs on Met1 activity is observed in the OTUB subfamily from other species

The above results established that the Met1 activity of OsOTUB1-cat and hOTUB1-cat is determined by the characteristics of the residues of the N-handle and C-handle motifs of their S1' pockets. To examine the generality of this observation, DUBs in the OTUB subfamily from other species were analysed. Sequences of the C65 peptidase family (from the database Pfam), consisting of 1847 sequences from 874 species (PF10275, updated to January 2020, Supplementary Data 1), which contains the OTUB subfamily, were retrieved and sequentially analysed by MEGA-X (Fig. 4a). First, sequences without catalytic activity were excluded based on analysis of the catalytic triad, leaving 57% of the sequences to be analysed in the next step (Fig. 4a–i). Second, these active sequences were studied to determine which incorporated the C-handle motif (G-only) due to the necessity of this motif for Met1 activity confirmed above; this was found in 68% of the active sequences (Fig. 4a–ii, after excluding 32% of NonG sequences). Third, these G-only sequences were studied to determine which precisely incorporated three residues in the N-handle motif (G-only-ordered), because sequences of this pattern can be analysed by our current model; this was found in 41% of the active sequences (Fig. 4a–ii). Fourth, the G-only ordered sequences were classified into five groups based on the residue composition in the N-handle motif: SGS, X'GS, XGN, XAD, and V, occupying 6%, 6%, 2%, 24%, and 3% of the active sequences, respectively, where X' represents any residue other than S, X represents any residue, and V (various) represents an irregular sequence pattern (Fig. 4a–iii). Finally, the X'GS and XAD sequences were subclassified into NGS, K/R-GS, G/A-GS, and V1-GS (Fig. 4a–iv) as well as K/R-AD, N/Q-AD, S/T-AD, and V2-AD (Fig. 4a–v) based on the similarity of the chemical properties of side chains. In these sequences, V1 (various 1) represents residues other than N, K, R, G and A, and V2 (various 2) represents residues other than K, R, N, Q, S and T. A total of 11 sequence patterns were obtained.

Statistical analysis of the G-only ordered sequences showed that those incorporating irregular patterns (V, V1 and V2) comprised only a small proportion (13.02%). To simplify the analysis, we excluded all sequences with irregular patterns. Those with specific patterns, such as SGS, NGS, K/R-GS, G/A-GS, XGN, K/R-AD, N/Q-AD and S/T-AD, occupied 86.98% and 35.66% of G-only ordered sequences and the active sequences, respectively. MEGA-X was used to calculate evolutionary distances between sequences under every pattern and OsOTUB1-cat, thus identifying the most evolutionarily remote sequences. Given that most of the sequences analysed originated from fungi, additional species were selected to broaden the range of species as much as possible. In total, fourteen candidate sequences were identified for

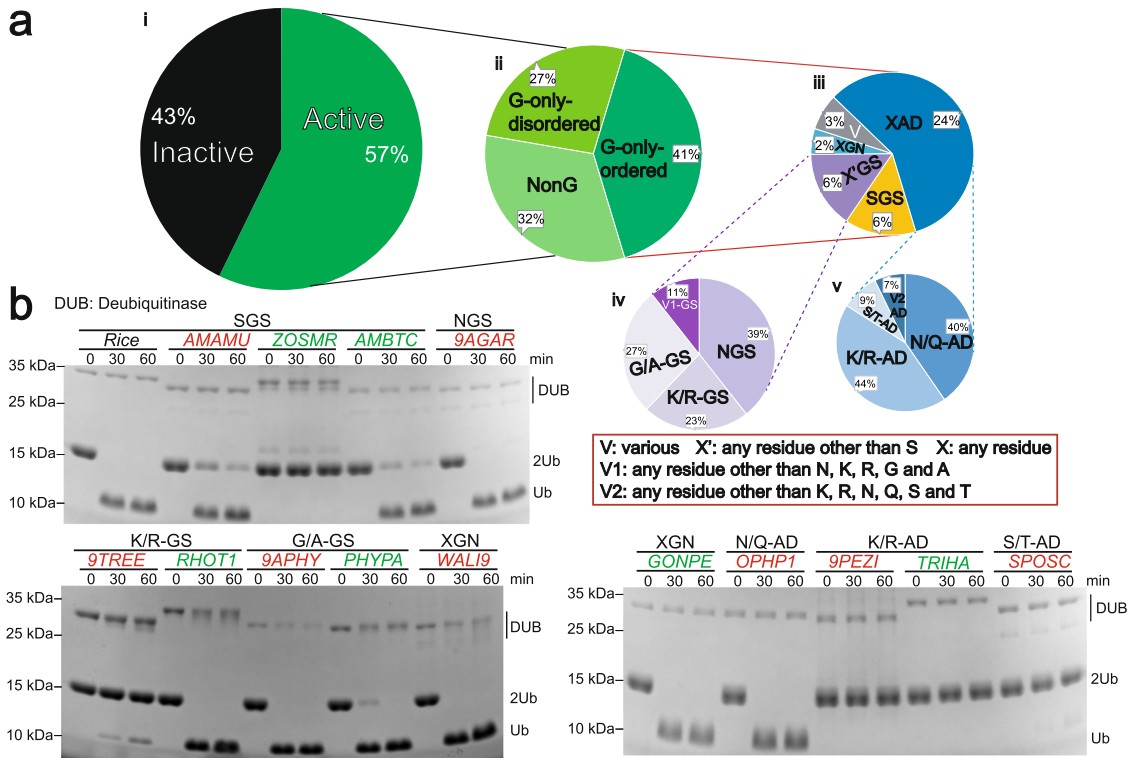

**Fig. 4 | Met1-specific motif rule is prevalent in members of OTUB subfamily.**
**a** Classification of C65 peptidase family based on characterization of catalytic triad and residues in two motifs of S1' pocket; **b** Hydrolysis activity against Met1-diUb by representative DUBs in every category (N = 3 biologically independent experiments). DUBs from species in red are the evolutionarily remotest ones from OsO-TUB1, while species in green are the ones other than fungus. *AMAMU, Amanita muscaria Koide* BX008; *ZOSMR, Zostera marina; AMBTC, Amborella trichopoda;*

*9AGAR, Hypholoma sublateritium* FD-334 SS-4; *9TREE, Kwoniella pini* CBS 10737; *RHOT1, Rhodosporidium toruloides* NP11; *9APHY, Daedalea quercina* L-15889; *PHYPA, Physcomitrella patens subsp. patens; WALI9, Wallemia ichthyophaga,* EXF-994/CBS 113033; *GONPE, Gonium pectoral; OPHP1, Ophiostoma piceae,* UAMH 11346; *9PEZI, Pseudogymnoascus* VKM F-4519; *TRIHA, Trichoderma harzianum; SPOSCS, Porothrix schenckii* 1099-18. Source data are provided as a Source Data file.

recombinant expression and assessment of their hydrolysis activity against Met-diUb: *AMAMU (Amanita muscaria Koide* BX008), *ZOSMR (Zostera marina (Eelgrass))* and *AMBTC (Amborella trichopoda)* in SGS mode; *9AGAR (Hypholoma sublateritium* FD-334 SS-4) in NGS mode; *9TREE (Kwoniella pini* CBS 10737) and *RHOT1 (Rhodosporidium toruloides* (strain NP11) *(Yeast))* in K/R-GS mode; *9APHY (Daedalea quercina* L-15889) and *PHYPA (Physcomitrella patens subsp. patens (Moss))* in G/A-GS mode; *WALI9 (Wallemia ichthyophaga* (strain EXF-994/CBS 113033)) and *GONPE (Gonium pectorale (green alga))* in XGN mode; *OPHP1 (Ophiostoma piceae* (strain UAMH 11346) *(Sap stain fungus))* in N/Q-AD mode; *9PEZI (Pseudogymnoascus sp.* VKM F-4519 (FW-2642)) and *TRIHA (Trichoderma harzianum (Hypocrea lixii))* in K/R-AD mode; *SPOSC (Sporothrix schenckii* 1099-18) in S/T-AD mode. The results showed (Fig. 4b) that, under almost the same enzymatic concentration, almost all the sequences incorporating the patterns SGS, NGS, K/R-GS, G/A-GS and XGN hydrolysed Met1-diUb, of which OTUBs from *9AGAR, RHOT1, 9APHY, PHYPA, WALI9 and GONPE* had the highest activity, cleaving 100% Met1-diUb within 30 min. Those from *AMAMU* and *AMBTC* had moderate activity, digesting 80% of Met1-diUb within 30 min, and those from 9TREE had weak activity, hydrolysing less than 10% of Met1-diUb within 30 min. These results demonstrated that sequences comprising the XGY pattern can hydrolyse Met1-diUb, confirming the association between XGY and Met1 activity. In contrast, almost all the sequences incorporating the XAD pattern (against the XGY pattern), including K/R-AD, N/Q-AD and S/T-AD, hydrolysed undetectable or negligible amounts of Met1-diUb within 60 min, thus verifying the XGY model again from the reverse aspect. Therefore, in general, the hydrolytic activity of an unidentified DUB against Met1-diUb might be judged from its N-handle motif of the XGY pattern and a

C-handle motif of G. However, there are also some exceptions here: 1) we could not detect any activity of the sequence from *ZOSMR* (belonging to SGS), although we rechecked the coding sequence, repurified the protein several times and tested the activity even when the protein concentration was elevated many times; 2) the sequence from *OPHP1* (belonging to N/Q-AD) showed effective catalytic activity (cleaving 100% of Met1-diUb in 30 min). Therefore, it is conclusive that the XGY pattern principle may be applied to determine hydrolysis activity against Met1 chains across OTUB subfamily members. Yet, this is not the whole story. Other factors that still need to be explored may exist.

Of the peptidases in the C65 family, 341 are from fungi, 255 are from metazoans, 76 are from green plants (*Viridiplantae*) (PF10275, updated to January 2020, Supplementary Data 1), and 149 sequences incorporate the XGY-G pattern. To analyze the evolutionary pathway of the XGY-G pattern, we analysed the distribution of species in which these 149 sequences were found (Supplementary Data 2). The results showed that these 149 sequences comprised six obsolete sequences and 143 valid sequences, originating from 130 species including 66 species of green plants, ranging from unicellular plants, such as green algae (*GONPE, Gonium pectorale*), to multicellular plants, such as Indian *rice* (*Oryza sativa* subsp. *indica*), domestic barley (*Hordeum vulgare* subsp. *vulgare*), soybeans (*Glycine max*) and cocoa (*Theobroma cacao*), together comprising 86.84% (66/76) of the total green plants in the C65 peptidase family. Thus, OTUBs from most green plants incorporate the XGY-G pattern, which means that OTUBs with Met1 activity are conserved in almost all green plants – from lower plants to higher plants. However, in contrast, sequences incorporating the XGY-G pattern were not found in any metazoan. Therefore, we

speculate that these differences in the sequences of the OTUBs in plants and animals and hence their contrasting propensities to hydrolyse Met1 chains already existed when the evolutionary paths of plants and animals diverged. Additionally, we note that sequences that conform to the XGY-G pattern are also present in 64 species of fungi, of which only 18.77% overlap with fungal species in the C65 peptidase family, indicating that OTUBs with Met1 activity are likely not conserved in fungi. However, sequences derived from most plant-parasitic fungi incorporate the XGY-G pattern, such as those from the brown rot fungus (*Gloeophyllum trabeum (strain ATCC 11539/FP-39264/Madison 617)*), *Tilletia walker*, white rot fungus (*Ceriporiopsis subvermispora, strain B*), and dwarf bunt fungus (*Tilletia controversa*)), implying that genes encoding OTUBs with Met1 activity in fungi might have been transferred from plants during their coevolution with plants, for example, during their resistance against or adaptation to the plant immunity system[8, 9, 36, 37]. Based on the above analysis, we speculated that OTUBs with Met1 activity are unique in plants.

## Discussion

Met1 chains have been intensively studied in animals and microorganisms but rarely studied in plants, perhaps due to the lack of high-throughput probes. In this paper, we describe a one-pot method for preparing a Met1-diUb probe carrying DHA and then the application of this probe in screening DUBs in *rice*. Our synthetic method was designed based on the recombinant expression of Met1-diUb, with a yield of approximately 100 mg of crystallization-competent probe in approximately 8 h under conventional conditions, enough to acquire relatively large quantities of crosslinked protein complex samples for structural analysis using X-ray or cryo-EM techniques. Using probes prepared by this method, OTUB1, which hydrolyses Met1 chains and other proteins, was captured in *rice*, implying the existence of regulators that recognize Met1 chains in plants. In recent studies, knockout or downregulation of OsOUTUB1 stabilized the transcription factor OsSPL14 by enhancing its Lys63 polyubiquitination, and resulted in improved plant architectural characteristics, such as a reduced tiller number, an increased grain number per panicle, and increased grain weight and yield[18]. Our data demonstrated that knockout or overexpression of OsOTUB1 could increase or reduce the abundance of Met1 chains in *rice*. Thus, it is speculated that manipulation of the gene encoding OsOTUB1 might also affect the traits of *rice* by interfering with the dynamics of Met1 chains or Lys48 chains, in addition to the modification of Lys63 polyubiquitination.

Since both OsOTUB1 and OTULIN hydrolyse Met1 chains, we structurally compared the binding of proximal Ub by OsOTUB1 with OTULIN to discover their similarities and differences. In OsOTUB1-cat-M1-diUb-DHA, the distal and proximal Ub moieties occupy the S1 and S1' pockets, respectively (Fig. 2c). The binding of OsOTUB1 to Met1-diUb is accompanied by the distortion of Met1-diUb when compared with the extended and compact structure of Met1-diUb[38, 39] (Supplementary Fig. 7c), which is similar to that in OTULIN(C129A)-Met1-diUb[34]. Surprisingly, compared with OTULIN(C129A)-Met1-diUb, proximal Ub in OsOTUB1-cat-M1-diUb-DHA adopts almost the same conformation (Supplementary Fig. 7a), while distal ubiquitin exhibits a spatial deflection with approximately 80°, which is probably caused by different orientation of the α helix responsible for binding of distal ubiquitin (Supplementary Fig. 7b). The S1 pocket interacts extensively with distal Ub. The hydrophobic patch of OsOTUB1-cat, formed by F225, F226, F229, L233, V256, I258, I259, L269, V271, Y273 and Y313, interacts with the hydrophobic patch of distal Ub, consisting of L8, I44, V70, L71 and L73 (Supplementary Fig. 7d). Additionally, N267, D275, and H289 of OsOTUB1-cat interact with Q40 of the distal Ub via hydrogen bonds (Supplementary Fig. 7e). However, binding of distal ubiquitin mediated by hydrophobic interactions is regularly seen in OTU family members, such as OTULIN (Met1-type specific), hOTUB2 (preferring Lys48- and Lys63-type) and Cezanne (preferring Lys11-

type)[12, 35, 40]. OsOTUB1 and OTULIN adopt nearly identical conformations to accommodate proximal Ub, with an RMSD of 1.76 Å (Supplementary Fig. 8). They both interact with proximal Ub via top, middle and bottom handlers. However, multiple obvious differences between their interactions with proximal Ub can be found in detail, both in the configuration and interaction number. In the top-handler, OsOTUB1 employs E90 to interact with K33prox, while OTULIN extensively interacts with D32prox, K33prox and E34prox by E95, R97, G98, T100 and K102, directly or indirectly. In the middle-handler, OsOTUB1 makes use of R314 to directly bind to E16prox, while OTULIN uses the catalytic residue N341 directly, and the other 5 residues (Y91, E335, I333, D336 and R338) indirectly bind with E16prox. OsOTUB1 binds to proximal Ub through a total of 8 hydrogen bonds, while OTULIN employs 16 hydrogen bonds to interact with proximal Ub. Taken together, although OsOTUB1 and OTULIN share a similar mode in accommodating proximal Ub, they employ residues with different orientations and quantities to accomplish these interactions.

OTUBs that hydrolyse Met1 chains are widely distributed across plants. It has been reported that hOTUB1 specifically hydrolyses Lys48 chains[29], and hOTUB2 hydrolyses both Lys48 and Lys63 chains[40], but neither can hydrolyse Met1 chains. Here, we report the association of sequence characteristics in the Met1-specific motif with Met1 activities. Variation in the C-handle motif of the S1' pocket has been previously reported to inhibit the hydrolysis of Lys63 chains by hOTUB1 but not hOTUB2[29]. Both the N- and C-handle motifs determine Met1 chain selection. The XGY mode in the N-handle motif is demonstrated to be widely distributed in *Viridiplantae* but not in *metazoans*, suggesting that it is plant specific. Additionally, the XGY pattern is also observed broadly in plant parasitic fungi, implying that DUBs with Met1 activity and incorporating the XGY pattern are probably involved in plant immunity, similar to RavD from the animal parasite *Legionella pneumophila*[9]. In the OTU family, the C-handle motif also correlates with the selectivity of the Lys63 chains, in addition to the selectivity of the Met1 chains. Therefore, the N-handle motif is more widely distributed in plants with a specific feature of Met1 selectivity than the C-handle motif.

In this manuscript, OsOTUB1 was proven to efficiently hydrolyse Lys48 chains, in addition to Met1 chains. To determine how OsOTUB1 recognizes and digests Lys48 chains, we tried several strategies to crystallize OsOTUB1 in complex with Lys48 chains but ultimately failed. Fortunately, Juang et al. reported[31] the crystal structure of Ub-UbcH5b^{C85S}-hOTUB^{Δ1-24} complexed with free Ub (Ub^{distal}-UbcH5b^{C85S}-hOTUB1^{Δ1-24}-Ub^{prox}), which includes two Ubs adopting the Lys48-diUb configuration and may provide clues for the interaction between Lys48-diUb and hOTUB1. Based on the structural alignment of OsOTUB1-cat-Met1-diUb-DHA with Ub^{distal}-UbcH5b^{C85S}-hOTUB1^{Δ1-24}-Ub^{prox} (Supplementary Fig. 11a), we found that the structure of OsOTUB1-cat-Ub^{distal} is similar to that of Ub^{distal}-hOTUB1^{Δ1-24}, with an RMSD of 1.53 Å (Supplementary Fig. 11c), proving that OsOTUB1 bears the S1 pocket almost the same as hOTUB1. However, proximal Ub in OsOTUB1-cat-Met1-diUb-DHA adopts a conformation distinct from that in Ub^{distal}-UbcH5b^{C85S}-hOTUB1^{Δ1-24}-Ub^{prox} (Supplementary Fig. 11b). This is mainly due to the distinct localization between M1 and K48 in Ub. For this reason, the residues in proximal Ub involved in the formation of Met1-type chains are totally different from those in Lys48-type chains (Supplementary Fig. 11b). Next, we aimed to illuminate the factors that affect the Lys48 activity of OsOTUB1 according to the structure of Ub^{distal}-UbcH5b^{C85S}-hOTUB1^{Δ1-24}-Ub^{prox}.

The finding that OsOTUB1 has Met1 activity expands the ubiquitin-linkage selectivity in the OTUB subfamily, ranging from Lys48 and Lys63 to Met1. In addition, our large-scale sequence alignment study indicated that two motifs required for Met1 selectivity are ubiquitous across the OTUB subfamily in plants, ranging from lower unicellular plants to higher multicellular plants, but not in animals. Recent studies have sequenced and analysed the genome and transcriptome of

*Mesostigma viride*, a unicellular plant positioned at the base of the Streptophyta clade, and deepened our understanding of the evolution of land plants from their unicellular aquatic ancestors[41]. However, it remains unknown what change triggered the divergence towards plants or animals in the initial stage of single-cell evolution. Classification of primitive unicellular organisms without typical features requires more decipherable genetic markers. Here, taxonomically, the motif features determining members of the OTUB subfamily that can hydrolyse Met1 chains might constitute a molecular standard to distinguish between plants (where these features are conserved) and animals (where they are not).

## Methods

### Bacterial strains and plasmids

A list of primers, plasmids and cell strains is included as Supplementary Data 3. *Escherichia coli* strain DH5α cells were used for DNA cloning purposes, transformed according to standard methods. The *E. coli* BL21 (DE3) was used instead for protein expression. The genes coding for OsOTUB1, Met1-diUb, Met1-diUb Q2A$^{prox}$, Met1-diUb Q2N$^{prox}$, Met1-diUb Q2E$^{prox}$, Met1-diUb Q2H$^{prox}$ and Met1-tetraUb were optimized, synthesized and inserted in pET22b by NdeI/XhoI by Genscript in Nanjing city, PR. China (https://www.genscript.com/). The genes coding for OTULIN-cat (residues 80-352), OTUB1s from *AMAMU* (*Amanita muscaria Koide BX008*), *ZOSMR* (*Zostera marina*), *AMBTC* (*Amborella trichopoda*), *9AGAR* (*Hypholoma sublateritium FD-334 SS-4*), *9TREE* (*Kwoniella pini CBS 10737*), *RHOT1* (*Rhodosporidium toruloides NP11*), *9APHY* (*Daedalea quercina L-15889*), *PHYPA* (*Physcomitrella patens* subsp. patens), *WALI9* (*Wallemia ichthyophaga, EXF-994/CBS 113033*), *GONPE* (*Gonium pectoral*), *OPHP1* (*Ophiostoma piceae, UAMH 11346*), *9PEZI* (*Pseudogymnoascus VKM F-4519*), *TRIHA* (*Trichoderma harzianum*), *SPOSCS* (*Porothrix schenckii 1099-18*) were optimized, synthesized and inserted in pET28a by NcoI/XhoI by Genscript in Nanjing city, PR. China (https://www.genscript.com/). Vectors used to express proteins related to Ub contain no affinity tag, the ones used to express proteins related to OsOTUB1 contain N-terminal 6×His tag followed by HRV 3C protease site (LEVLFQG), while the others contain only N-terminal 6×His tag.

**Expression vector constructions for OsOTUB1 truncation and mutations.** PCR amplified from *OsOTUB1* mentioned above using the oligonucleotides F-OsOTUB1-Δ73 to introduce an NdeI site at the translational start codon and encode a 6×His tag followed by HRV 3C protease site, and R-OsOTUB1-Δ73 to introduce an XhoI site at the end of the ORF. To generate a OsOTUB1-cat recombinant protein (the N-terminal 73 residues were removed) with a hexa-histidine-HRV3C-tagged site at the N-terminus, the corresponding PCR product was extracted from the agarose gel, cloned into the pET22b (Merck), forming the recombinant plasmid pET22b_OsOTUB1-cat. Site-directed mutations were based on pET22b_OsOTUB1-cat according to conventional methods. Briefly, pairs of primers used to generate site-directed mutations (F-G117P/R-117P, F-EDD/R-EDD, F-S93E/R-S93E, F-G94A/R-G94A, and F-S95D/R-S95D) primed the annular amplification of the pET22b_OsOTUB1-cat followed by DpnI digestion. The digestive product was transformed to DH5α and the positive clones were screened on solid LB medium containing 2% agar and 100 μg/mL ampicillin.

**Expression vector constructions for hOTUB1 variants.** PCR amplified from cDNA (purchased from YouBio) using the oligonucleotides F-hOTUB1 to introduce an NcoI site at the translational start codon and encode a 6×His tag, and R-hOTUB1 to introduce an XhoI site at the end of the ORF. To generate a hOTUB1 recombinant protein with a hexa-histidine-tagged site at the N-terminus, the corresponding PCR product was gel-purified, double-digested and cloned into the pET28a (Merck), forming the recombinant plasmid pET28a_hOTUB1. To construct expression vector for hOTUB1-cat (the N-terminal 39 residues

were removed), F-hOTUB1-Δ39 to introduce an NcoI site at the translational start codon and encode a 6×His tag, and R-hOTUB1 to introduce an XhoI site at the end of the ORF. The PCR product was purified and cloned into the pET28a (Merck), forming the recombinant plasmid pET28a_hOTUB1-cat. Site-directed mutation was based on pET28a_hOTUB1-cat and followed the similar procedures with that of *OsOTUB1*, except that the paired primers were used to generate P87G (by F-P87G/R-87G) and SGS (by F-SGS/R-SGS), respectively.

**Expression vector constructions for OsUCH14, OsUCH-L, hUBCH5B and OsUBCH5B.** The protocol is similar with that of *hOTUB1* mentioned above, except that cDNA from human cell (purchased from YouBio) or from rice were used as the PCR templates, and the F-UCH14/R-UCH14, F-UCHL/R-UCHL, F-hUBCH5B/R-hUBCH5B and F-OsUBCH5B/R-OsUBCH5B were used as the paired primers for PCR.

DNA sequences of all genes were verified by Sanger sequencing.

### Expression and purification of proteins

Expressions of all N-terminally hexahistidine-HRV 3C-tagged or hexa-histidine-tagged proteins used in this work were carried out following isopropyl-β-thiogalactoside (IPTG) induction in BL21 (DE3) *E. coli*. Bacteria were grown at 37 °C in 1000 ml LB broth to 0.6–0.7 absorbance at 600 nm. IPTG was then added to 0.3–0.5 mM and the culture was grown for 12 h at 22 °C. Cells were harvested by centrifugation at 3000 × *g* at 4 °C, resuspended in 40 ml of lysis buffer (50 mM Tris•HCl pH 8, 500 mM NaCl, 40 mM imidazole, 2 mM dithiothreitol) and lysed by sonication. After centrifugation (30,000 × *g*, 30 min, 4 °C), the supernatant was recovered and proteins were separated from whole-cell lysates by Ni-NTA agarose chromatography (Qiagen, Inc). After three washing steps with lysis buffer, 6×His-tagged proteins were eluted from the resin with 300 mM imidazole in lysis buffer. In the case of hexa-histidine-HRV 3C-tagged proteins, after three washing steps with lysis buffer, 1 mg HRV 3 C protease and DTT to 2 mM final concentration were added. The mixture was incubated overnight at 16 °C. Proteins were eluted from the resin with 5 mM imidazole in lysis buffer. A final size-exclusion chromatography step (S200-Superdex, GE) was performed for all proteins, pre-equilibrating the column by gel-filtration buffer (50 mM Tris•HCl pH 8, 150 mM NaCl, 2 mM dithiothreitol), and eluting isocratically at 0.4 ml/min with high-performance liquid chromatography (AKTA Purifier, GE). Protein purity was monitored by Coomassie blue staining after SDS-PAGE on a 15% polyacrylamide gel. Protein concentrations were determined by UV spectroscopy.

### Resolution of crystal structures of OsOTUB1-cat and that in complex with Met1-diUb-DHA and with Ub-PA

**Crosslinking complex formation and purification.** OsOTUB1-cat was incubated with 3 folds molar Met1-diUb-DHA and Ub-PA in 30 °C for 3 h, after which the complexes were separated by MonoQ and further by size exclusion chromatography (Superdex 75 16/60) in Gel filtration buffer. The purified proteins were concentrated, flash frozen in liquid nitrogen and stored at −80 °C.

**Acquisition of protein crystals.** The initial crystal screening process was based on method of sitting-drop vapor diffusion. The OsOTUB1-cat (residues 74-324) with a concentration of 10 mg/mL was mixed with the crystallization reservoir at a volume ratio of 1:1 (1 μL: 1 μL), and crystals satisfying the diffraction requirement were finally obtained under the 95# condition of Hapton Index 49-96, which contained 0.1 M Potassium thiocyanate, 30% w/v Polyethylene glycol monomethyl ether 2000. The OsOTUB1-cat-UbPA with a concentration of 7 mg/mL was mixed with the crystallization reservoir at a volume ratio of 1:1 (1 μL: 1 μL), and crystals satisfying the diffraction requirement were finally obtained under the 72# condition of Hapton Index 49-96, which contained 0.2 M Sodium chloride, 0.1 M HEPES pH 7.5, 25% w/v Polyethylene glycol 3,350. The OsOTUB1-cat-Met1-diUb-DHA with a

concentration of 8 mg/mL was mixed with the crystallization reservoir at a volume ratio of 1:1 (1 μL : 1 μL), and crystals satisfying the diffraction requirement were finally obtained under the 70# condition of Hapton Index 49-96, which contained 0.2 M Sodium chloride, 0.1 M BIS-TRIS pH 5.5, 25% w/v Polyethylene glycol 3350. Before collecting the diffraction data, the crystals were firstly cryo-protected with crystallization reservoir containing 30% glycerol.

**Resolution of crystal structures.** The diffraction data were initially collected at the Shanghai Synchrotron Radiation (SSRF) line stations 18U and 19U, and processed using HKL3000. The initial structure of OsOTUB1-cat was constructed by molecular replacement by CCP4, for which the template is the chain-sawed crystal structure of catalytic domain in hOTUB1 (2ZFY). Coot and PHENIX were used alternately for multiple rounds of refinement and model construction. For structures of OsOTUB1-cat-UbPA and OsOTUB1-cat-Met1-diUb-DHA, structures of OsOTUB1-cat and ubiquitin (1UBQ) were used as templates for molecular replacement by CCP4. Coot and PHENIX were used alternately for multiple rounds of refinement and model construction. All structural data were generated by Pymol (www.pymol.org).

**Preparation of Ub-PA**
100 mg MesNa, 5 mg TCEP and 50 mg guanidine hydrochloride per mL was added to the supernatant of cell lysis after acid precipitation by perchloric acid, before which the pH was adjusted to 7.0. After exchanging most of the air with nitrogen, it was sealed with a plastic sealing film, tightly closed, and transferred to a heating shaker at 50 °C for 48 h. After chromatography and mass spectrometry analysis, it was confirmed that the reaction was completed, and then purified by HPLC and lyophilized.

After frozen-dried for several days, the sample was dissolved in 6 M guanidine hydrochloride at pH 2.3, transferred to an ice salt bath and stirred. After the temperature was lowered to −20 °C, sodium nitrite was added in 20-fold equivalent, and oxidation was completed for about 30 min which produced the intermediate product Ub-azide. Next, the pH of the solution is quickly adjusted to basic (usually 9.0 to 10.0) with propargylamine, in which case it is theoretically possible to convert all of the starting materials to Ub-PA probe. After confirming the reaction by chromatography and mass spectrometry, it was purified by HPLC and frozen-dried.

Because Ub-PA was mostly used in cross-linking experiments with DUB, it is necessary to refold the frozen-dried polypeptide. Thus, we first dissolve it with guanidine hydrochloride buffer. Then in a low temperature environment (usually 4 °C) water or neutral pH buffer was gradually added, the ubiquitin is refolded by gradually diluting the concentration of guanidine hydrochloride. Finally, Ub-PA that can be used for the crosslinking reaction was obtained by gel filtration.

**Preparation of Met1-diUb-DHA**
Recombinant Met1-diUb mutant (Gly76 in distal Ub was mutated to Cys) containing N-terminal Avi tag (GLNDIFEAQKIEWHE) (Avi-Met1-diUb-76C) in pET22b was expressed in *E. coli* BL21 (DE3). After purification as mentioned in 1.1, in the concentrated Avi-Met1-diUb-76C (about 8 mg/mL), 1 μM BirA, 2 mM ATP, 10 μM Mg$^{2+}$, 0.2 mM D-Biotin and 10 folds bisamide of the 1,4-dibromobutane were added simultaneously. After adjusting the pH to 9.5 by NaOH, the mixture was placed in 37 °C for about 10 h. The final reaction mixture was further purified by Superdex 75 16/60 in Gel filtration buffer.

**Deubiquitination assay**
DUBs were diluted in Gel filtration buffer and incubated at 37 °C for 5 min. The DUB assays were subsequently carried out where 2 μM of DUB was incubated with 60 μM diUbs in Gel filtration buffer in a reaction volume of 25 μl. The reactions were incubated at 37 °C and 6 μl samples were taken at different time points and directly mixed with

4×SDS sample buffer to stop the reaction. The samples were separated on 4-12% SDS-PAGE gel (Life Technology) and stained by Coomassie brilliant blue G-250.

**Ub-PA crosslinking assay**
DUBs were diluted in Gel filtration buffer and incubated at 37 °C for 5 min. The crosslinking assays were subsequently carried out where 20 μM of DUB was incubated with 100 μM Ub-PA in Gel filtration buffer in a reaction volume of 25 μL. The reactions were incubated at 37 °C and 6 μL samples were taken at different time points and directly mixed with 4×SDS sample buffer to stop the reaction. The samples were separated on 4-12% SDS-PAGE gel (Life Technology) and stained by Coomassie brilliant blue G-250.

**Ub-AMC hydrolysis essay**
Ub-AMC was diluted in Gel filtration buffer to 400 nM. For each reaction 100 μL of diluted substrate in a black 96-well low volume plate (Corning) was mixed with 100 μL of 4 μM DUBs (40 nM for UCHL3) at room temperature. The rate of AMC generation was measured using a Synergy HT detector (Biotek). Fluorescent intensities were recorded following excitation at 340 nm and emission at 440 nm.

**Plant materials and growing conditions**
*OsOTUB1* overexpression lines and knockout mutant were created in Zhonghua11 (ZH11) genetic background as previously described[18]. *Rice* seeds were disinfected in 20% sodium hypochlorite solution for 30 min, thoroughly washed with deionized water, and then germinated in 96-hole plate for 14 days in hydroponic nutrient solution. Paddy-grown plants were spaced 20 cm apart and were grown during the standard growing season in Beijing. Seedlings and young panicles were collected and stored in liquid nitrogen for protein extraction.

**Western blotting**
OsOTUB1 and ubiquitin chains were retrieved from ZH11 and its genetically modified variants (overexpression or knockout of *OsOTUB1*) in the seedling stage (14 days) or young panicle differentiation stage (0.2 cm young panicles), using a buffer composed of 50 mM HEPES (pH 7.5), 150 mM KCl, 1 mM EDTA, 0.5% Triton X-100, 1 mM DTT and a proteinase inhibitor cocktail (Roche LifeScience, Basel, Switzerland). The obtained lysates were subjected to SDS-PAGE, and the separated proteins were transferred to a nitrocellulose membrane (GE Healthcare). OsOTUB1, GFP, OsHSP82 (rice), total ubiquitin conjugates, Met1 ubiquitin chains and Lys48 ubiquitin chains were detected by probing the membrane with an antibody against OsOTUB1 (BGI, antiserum was generated by injecting purified OsOTUB1 into rabbits, dilution 1:5000), GFP (Roche, catalog #11814460001, dilution 1:5000), HSP82 (rice) (Beijing Protein Innovation, catalog #AbM51099-31-PU, dilution 1:10000), ubiquitin (Abcam, catalog #ab134953, dilution 1:5000), Met1 ubiquitin chains (Merck-millipore, catalog #MABS199, dilution 1:2000) and Lys48 ubiquitin chains (Abcam, catalog #ab140601, dilution 1:5000), respectively. Peroxidase-AffiniPure Goat Anti-Rabbit IgG (Jackson 111-035-003, dilution 1:10000) and Peroxidase-AffiniPure Goat Anti-Mouse IgG (Jackson 115-035-003, dilution 1:10000) were used as secondary antibody.

**Surface Plasmon Resonance (SPR)**
SPR was measured using a Biacore 8 K + instrument (Cytiva). The measurements were performed at 25 °C in 10 mM PBS buffer. Series S CM5 sensor chip (Cytiva) was used to immobilize the OsOTUB1-cat-C121A for 2 min, and Met1-diUb variants were used as the analytes and passed over the chips. Met1-diUb variants were diluted in the PBS buffer at concentrations of 0.3125-80 μM and injected at 30 μl/min for 2 min at 25 °C. Biacore Insight Evaluation 3.0.12.15655 was used to fit

the curve in the steady state affinity model and to overlay plot to depict the interaction between OsOTUB1-cat-C121A and Met1-diUb variants.

## Chain cleavage kinetics monitored by High Performance Liquid Chromatography (HPLC)

Mono ubiquitin production upon cleavage of the peptide bond between the distal and proximal ubiquitin molecules was used to derive Michaelis-Menten rates. The cleavage assays were performed at 30 °C in a reaction buffer containing 50 mM Tris·HCl, pH 7.5, 150 mM NaCl and 2 mM DTT. OsOTUB1-cat (4 or 40 μM) was mixed with specified amounts of Met1diUb variants. Aliquots of the reaction mixtures were removed at specified time points, quenched by the addition of 6 M Guanidine·HCl, and monitored by HPLC (Shimadzu LC-2030C 3D Plus, detector RID-20A) with 5 μm column of Symmetry300™ C18 (Waters) following linear gradient elution (reaching 60% of buffer B from 20% in 60 min). Double distilled water and acetonitrile, added with 1‰ trifluoroacetic acid, were used as buffer A and B, respectively. The initial amount of product for each time point was converted from the area of its peak traced by HPLC. Double recordings were made for each substrate concentration. Reaction velocities were fit to the Michaelis–Menten equation with Graphpad Prism 5.

The relationship between the molar amount of product and its peak area in HPLC is established through a standard curve (Supplementary Fig. 5). Mono ubiquitin (50–1500 pmol) in a reaction buffer containing 50 mM Tris·HCl, pH 7.5, 150 mM NaCl and 2 mM DTT, was unfolded by addition of 6 M Guanidine·HCl, and monitored by HPLC with 5 μm column of Symmetry300™ C18 (Waters) following linear gradient elution (reaching 60% of buffer B from 20% in 60 min). Double distilled water and acetonitrile, added with 1‰ trifluoroacetic acid, were used as buffer A and B, respectively. The standard curve was established by fitting the molar amount of mono ubiquitin and its peak area traced by HPLC into the linear regression equation with Graphpad Prism 5. Double recordings were made for each mono ubiquitin concentration.

## Quantification of protein concentration by bicinchoninic acid (BCA)

Protein concentration was quantified by the enhanced BCA protein assay kit produced by Beyotime (Category No. P0010S). The specific experimental steps are carried out in accordance with the manufacturer's instructions.

## Reporting summary

Further information on research design is available in the Nature Research Reporting Summary linked to this article.

## Data availability

Source data are provided with this paper. The mass spectrometry proteomics data generated in this study have been deposited to the ProteomeXchange Consortium via the PRIDE partner repository with the dataset identifiers PXD032822. 6K9N (crystal structure of apo OsOTUB1); 6KBE (crystal structure of OsOTUB1-Met1-diUb-DHA); 6K9P (crystal structure of OsOTUB1-Ub-PA); 1UBQ (crystal structure of ubiquitin); 2ZFY (crystal structure of apo hOTUB1); 3ZNZ (crystal structure of OTULIN OTU domain (C129A) in complex with Met1-diUb); 2W9N (crystal structure of extended Met1-di ubiquitin); 3AXC (crystal structure of compact Met1-di ubiquitin); 4DDG (crystal structure of Ub$^{distal}$-UbcH5b$^{C85S}$-hOTUB1$^{Δ1-24}$-Ub$^{prox}$); sequences of C65 peptidase (updated to January 2020) were retrieved from database Pfam (PF10275) and are supplied in Supplementary Data 1. Source data are provided with this paper.

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

## Acknowledgements
We thank the staffs from BL17B1/BL18U1/BL19U1/BL19U2/BL01B beamline of National Facility for Protein Science in Shanghai (NFPS) at Shanghai Synchrotron Radiation Facility, for assistance during data collection. We thank the Analytical Instrumentation Center in Peking University for assistance with mass spectrometry and Dr. Wen ZHOU for help with data processing and uploading. This work was supported by the National Natural Science Foundation of China Grants (32071269, 31770827 and 21736002 to F.W., 91753205 and 31870791 to Z.M. and 31500980 to S.W.), the National Key Research and Development Program of China (2017YFD0500400 to Z.M.), the funding of the GuangXi Ba-Gui Scholar Program to Z.H., the start-up funding for the introduced talents in Guangxi University to L.L., the special program for the science and technology base and talent of GuangXi (2021AC19338 to L.L.), and the Beijing Institute of Technology Research Fund Program for Young Scholars.

## Author contributions
L.L. initiated the project, L.L., F.W., Z.H. and Z.M. oversaw all aspects of the project. L.L., X.Z., X.L., S.W., L.Z., L.W., X.J., L.J.L., Z.D., Z.L., Y.W., X.F., H.H. and J.W. designed and performed all experiments. L.L., F.W., Z.H. and Z.M. wrote the manuscript.

## Competing interests
The authors declare no competing interests.
