## [Peer Review File · Nature Communications]

REVIEWER COMMENTS

Reviewer #1 (Remarks to the Author):

Using a semisynthetic Met1-linked diubiquitin (Met1-diUb) suicide probe, the authors screened a plant cellular protein extract with the hopes of enriching and identifying deubiquitinases (DUBs) that may recognize Met1-linked polyubiquitin chains. They successfully identified an OTUB1 homolog from *Oryza sativa* (OsOTUB1) which showed a distinct substrate preference for Met1-linked poly-Ub chains. This seems to be the first plant-derived enzyme capable of hydrolyzing Met1-linked chains and having a preference for the same over other linkage types. To understand enzymatic properties and substrate preference they crystallized apo OsOTUB1, the same bound to a Met1-diUb suicide probe (to visualize both S1 and S1' interactions) and a Ub-based suicide probe that targets only the distal site (S1 site) of the enzyme. The structures showed conformational rearrangement of the catalytic residues induced upon binding of the substrate analogs. The authors interpreted these substrate-induced changes in terms of an activation model, they call 'bi-modal activation', and attempted to provide mutagenesis data in support of this model. Furthermore, the authors noticed some unique sequence motifs OsOTUB1 that appear to confer ability to recognize Met1-linked poly-Ub chain substrates in yet to be characterized proteins. They searched a peptidase database and attempted to identify OTUB DUBs from other organisms showing hydrolytic activity toward Met1-linked diubiquitin substrate.

Overall, this paper does provide some interesting insights into mechanism of DUBs and expands our appreciation of their versatility. By showing that a homolog of human OTUB1 can exhibit Met1-linked diubiquitin hydrolysis in addition to cleaving K48-linked chains, the authors are able to demonstrate that OTUB DUBs could be versatile. The evolutionary connections arising from their work appear to provide additional insights into similar enzymes in plants and fungi. The structural and biochemical data presented are in general of high quality and the effort that has gone into the work is commendable.

In the negatives, the structural data are not interpreted with the same level of competence. The paper has a number of inconsistencies and overreaching statements that are not adequately supported by experimental evidence, which applies even to some of the primary conclusions in the paper. Thus, the negatives outweigh the positives in the current version of the paper.

Major Issues:

No explanation was given as to why other DUBs identified in the activity profiling were not considered, even though the score for one of them, OsUCH14 (human USP5 homolog), is higher than OsOTUB1 (Figure 1), the candidate they chose to pursue. It seems the authors abandoned the other DUBs from further analysis because some of them were also identified in a previous activity-based screen in human cells (Weber et al 2017). Somehow the authors favored OTUB1 but no mention of Met1-polyUb activity of UCH14 and UCH-L and biochemical analysis of their DUB properties.

Please include linkage specificity data for OsUCH14, and, if possible, for UCH-L. It might help explain the conundrum in Figure 1D: the knockout is marginally different from the wt seedlings (in Met1-polyUb staining). That could mean that there are other Met1-linked DUBs playing a more prominent role in vivo.

Please explain why do we see enrichment of Poly-Ub and the E1 enzyme. What do these observations indicate about the limitation of the approach?

It seems the catalytic domain alone is better at cleaving K48-linked diUb than the full-length enzyme (Figure 1F). What is the role of the deleted segment? Please explain.

Figure 1D: Please provide corresponding results for K48-linked chains. In the same figure, including the catalytic mutant would help. Without that result, one cannot directly relate the observation to DUB activity. Include data showing levels of protein expression in wt, KO and overexpressed seedlings.

Bi-modal activation:

Please explain clearly the meaning of 'bi-modal activation'. After reading the paper I got the sense that it is a fancy word for the widely observed phenomena of substrate-induced conformational changes in the vicinity of the active site of the enzyme. I am not a stickler for words, but the word bi-modal does not make any sense in the context of the paper.

From the way the activation model is described it seems conformational changes occur in a sequential fashion: first, the distal Ub-induced alignment of active site residues (His alignment), then 'pushing' by Q2 of the proximal Ub. While this sounds interesting, no evidence is provided.

What is the significance of the bi-modal activation mechanism? Is there a biological implication of this type of diUb recognition?

I do not agree with 'activation by both distal and proximal ubiquitin' mentioned in the abstract. The crystals structure with Ub-PA shows the catalytic His is already in place due to changes induced upon distal Ub binding. The cavity created by this movement is simply occupied by Q2 of the proximal Ub. So, instead of Q2 'pushing' the histidine, it is merely occupying an empty area near the active site to maximize interactions with enzyme residues. I am not sure if this interpretation is less plausible than the one provided by the authors. If it cannot be tested, the authors should indicate that this and similar arguments as speculative remarks and should include them in discussion rather than in the abstract, where one expects to see only the main, evidence-backed results.

The observation that in the Ub-PA complex the histidine has already moved to its catalytic competent orientation is a blow to the notion that proximal Ub 'activate' the enzyme. This is a serious inconsistency and should be addressed carefully.

Please provide experimental evidence in support of 'proximal ubiquitin triggers full activity of OsOTUB1 and the hydrolysis of Met1 chains'.

(Line 203) The Q2E mutational result shows the mutant is impaired worse than the Ala mutant. The authors claim this is due to Glu sidechain making an electrostatic interaction with the catalytic His thereby by holding it in an inactive conformation, which would render activation by proximal Ub more difficult. I suspect this is a biased interpretation. One could argue that the additional negative charge of the Q2E mutant may introduce repulsion near the active-site area (for example, with D319 and even with E16 of proximal Ub) explaining the loss of activity.

With regard to Figure 2G, this is exactly what I would expect for a DUB that has activity toward both K48 and Met1-linked diubiquitins. As the authors point out, it is indeed distinct from OTULIN. The reactivity with Ub-PA (and Ub-AMC) cannot be and should not be taken as evidence in favor of activation by distal Ub. Such reactivity is seen in a wide variety of DUBs bearing a catalytic Cys.

M1-specific Motif

In my view, the M1-specific motif has not been validated adequately. Please consider mutating the residues and asking if there are differential changes in hydrolysis of the Met1-linked linked substrate versus the K48-linked one. One expects the overall activity of the mutants to be lower than wt, but its specificity might be altered if the residues do indeed determine Met1-diub specificity.

If the distal and proximal Ub binding sites are conserved between both human OTUB1 and OsOTUB1, how are they so different?

It is hard to understand that the SGS motif (N-handle motif), which barely has any interaction with the proximal Ub (figure 3F), will have a role in conferring M1 hydrolytic activity.

Minor points:

What are the interactions with Met1 of the proximal Ub?

Please consider showing the faint Ub band in Figure 3C is indeed the Ub product resulting from Gly76-Met1 peptide bond cleavage.

Please explain the rationale behind probing longer poly-Ub chains in Figure 3E.

Please change the wordings of 'Two-orientation activation mode of OsOTUB1-cat to Met1 chains'. As a subheading it does not represent the text that follows.

Overall citation to previous work is incomplete. Some examples:

1. Include citations to distal Ub and proximal Ub conformational changes induced upon Ub binding as observed in other family of DUBs, such as USP, UCH and OTU.
2. I think the reference to the first synthesis of dehydroalanine Met1-DiUb using the same desulfurization method would be appropriate (Haj-Yahya N, et al, 2014) in addition to the one provided.

Reviewer #2 (Remarks to the Author):

In this manuscript the authors describe a novel molecular mechanism by which a OTUB1 deubiquitinating enzyme from *Oryza sativa* possesses activity against linear Met1-linked ubiquitin chains. So far cleavage of Met1-Ub linkages has been restricted to human OTULIN, another member of the OTU subfamily of DUBs, but in plants this activity seems to be present in OTUB1. The manuscript also explains the structural determinants for the differences between human OTUB1, which has only activity against K48 and K63 Ub chains, and plant OTUB1, in which the presence of particular structural elements allows cleavage of Met1-Ub substrates. The results presented in this manuscript are interesting and the structural analysis is well done, however there are particular issues that should be addressed.

Major Points:

From a mechanistic point of view, it is confusing the distinction between the activation by distal or by proximal Ub. OsOTUB1 can be activated by a distal Ub alone, which explains the activity for Ub-AMC substrates, but in the absence of the Gln2 residue from the proximal Ub, which seemed necessary to align the catalytic triad, as the authors report. The authors should clarify better this point. What's the relevance of Gln2 ?? Maybe it is only needed to cleave Met1-Ub substrates ?.

Paragraph 115-124. The author conclusions' from western blot reaction in figure 1D are difficult to observe. The different band intensities between lanes in anti-M1Ub are not clear at all. Some statistical quantification should be done.

Additionally it would be interesting to observe the endogenous levels of OsUTUB1 in WT, KO and OE mutants. From these data, the levels of linear Met1Ub do not seem very affected by OsOTUB1, as observed in the WB.

Figure 1F. Please load molecular weight markers in the SDS-PAGE, otherwise it is difficult to correlate the bands with the MW. It is particular relevant when you discuss on the deletion of the N-terminal extension. Have the author's checked whether there is a role for the N-terminal extension in the activity of OsOTUB1 ?

As stated previously, if distal-Ub alone can rearrange the active site of OsOTUB1, then what is really the relevance of Ub proximal Gln2 ? From the activity assays in figure 2E, Q2 seems essential for linear diM1Ub cleavage, but not for hydrolysis of Ub-AMC ??.

Is there not a binding pocket for Met1 in the complex structure ?? A better figure of the active site environment cleavage residues would help to understand the process (including the crosslink).

What is the structural homology between OTULIN and OsOTUB1 ?. Since they belong to the same OTU subfamily and both can interact with the proximal ubiquitin of a linear Met1-diUb,

it would be interesting to show or to discuss whether they share a similar binding mode to the proximal Ub.

Lane 283. Figures should be better introduced in the main text. In ex., fig3D and 3E in the main text (they are confusing).

Lane 286. In figure 3C the intensity band that increases at longer times with the double mutant is very weak and difficult to observe. The conclusion that they confer diM1Ub activity to hOTUB1 is highly questionable. I guess that other (not obvious) structural determinants are present in plant OsOTUB1.

Another issue that could be clarified is about the mechanism of OsOTUB1 to cleave Ub-K48 chains, which seems also as active as for diUbM1 (Fig 1E). Is there any interaction of the proximal Ub with OsOTUB1 in diUbK48 as occurs in the cleavage of Met1-linked chains? Perhaps activity assays with the OsOTUB1 proximal Ub mutants could be conducted with K48-linked chain substrates.

There are several publication on the interaction of OTUB1 with the E2 conjugation enzyme. In particular there seems to be the formation of a N-terminal α -helix that interacts with the proximal Ub and the E2 enzyme in OTUB1 (Wiener et al, 2012, Nature). Additionally E2 binding reciprocally regulates the conjugation-deconjugation cycles. It would be interesting in the discussion to add some information on this point when compare human and OsOTUB1. Is such helix present in OsOTUB1?. Does it bind E2 enzymes ?.

Reviewer #3 (Remarks to the Author):

The authors design a new strategy to synthesise a diUb probe that reacts with deubiquitinating enzymes that cleave Met1 (linear) ubiquitin conjugates. They then take this probe and look for Met1-selective DUBs in plants. The presence of Met1-specific DUBs in plants has not been carefully investigated and it is interesting that the authors identify candidate Met1-selective DUBs, notably OsOTUB1.

Given that the Met1 ubiquitin signal in mammals is predominantly associated with inflammation and the fact that plants have a more diverse and complex ubiquitination system than mammals, it is highly relevant that a Met1-selective DUB has been identified, which will lead the way to future studies to investigate the role of the Met1 ubiquitin signal in plants.

The authors then go on to mechanistically characterise the role of OsOTUB1 in how it recognizes and cleaves Met1 ubiquitin and present three crystal structures detailing how OsOTUB1 functions.

Overall, this reviewer finds this an interesting study and the identification of a plant Met1-selective DUB opens up the field to investigating the Met1 signal in plants. However, the mechanistic insights are poorly translated from the structural analyses. Notably, no experiments decouple the effects of binding mutations (K_d) from catalytic activities (k_{cat}). Descriptive information between the mode of action of OTULIN and OsOTUB1 is made but no solid conclusions can be made to compare the mechanism of OsOTUB1. Overall, this reviewer found the presentation and execution of some of the biochemical experiments as sloppy and not worthy of publication in their current form. It was not clear on how many occasions these experiments had been performed and it was not clear, in some DUB assays, whether the ubiquitin substrates were actually being cleaved or whether contaminating mono ubiquitin was already present in the input.

Without a separation of binding versus catalytic mutants this reviewer does not feel that the authors can make the statement that OsOTUB1 is a bi-modal activated deubiquitinase and further careful structural interpretation and biochemical experiments are required to justify these assertions.

Below are some more specific and minor comments relating to the manuscript.

In the structure of apo OsOTUB1 the catalytic His (Hiscat) is in a non-catalytic configuration whereas in the structure of OsOTUB1 bound to Met1 diUb DHA, Hiscat is in a catalytically competent orientation, as would be expected since this structure represents a trapped intermediate of the reaction. The authors focus their attention on Gln2 of the proximal ubiquitin moiety that occupies the position where the imidazole ring from Hiscat resides in the inactive apo configuration. A series of mutations are made on Gln2 to assess the ability of OsOTUB1 to hydrolyse Met1 ubiquitin variants. Figure 2E shows a DUB assay (with no time zero control) showing different lengths of incubation of the Met1 ubiquitin variants with OsOTUB1. From this experiment it is impossible to

conclude whether the lack of Met1 ubiquitin cleavage is due to a deficiency in binding of Met1 ubiquitin to OsOTUB1 or whether Gln2 directly contributes towards the catalytic site affecting the rate of hydrolysis of Met ubiquitin. At the very least, kinetic information on the efficiency of OsOTUB1 is required to show that the effect is not simply related to K_d but rather k_{cat} .

Confusingly, the authors state that in a previous study they performed on OTULIN (see line 211), that the proximal Glu16 disrupts the inhibitory effect of Asp336 and aligns the catalytic histidine, providing a mechanism of ubiquitin-assisted catalysis. It should be noted that the Keusekotten 2013 paper separated the binding function of Glu16 and the catalytic role of Glu16. Interestingly, Gln2 was also in proximity to the active site but had no effect on catalysis. What are the effects of other residues from the proximal ubiquitin on activity of OsOTUB1? Notably, Glu16 and Lys33 as shown in the figure?

The authors then tested the role of distal Ub binding on the activity of OsOTUB1 through determining a structure of Ub-PA reacted with OsOTUB1 and testing ability to hydrolyse Ub-AMC. OsOTUB1 is able to hydrolyse Ub-AMC, though at a much-reduced rate than UCHL3, suggesting that binding of the distal Ub is sufficient to induce rearrangement of the catalytic triad in a more catalytic configuration. This is not the case for OTULIN as pointed out by the authors and shown in Figure 2G and S Figure 3B. However, the authors then go on to solve the structure of OsOTUB1 bound to Ub-PA. This reviewer could not identify an overlay between the OsOTUB1 Met1 diUB DHA structure with OsOTUB1 Ub-PA structures. However, from Figure 2H it is clear that the Hiscat is in a catalytically active configuration (which would be expected). Therefore, since the proximal ubiquitin is not present in the OsOTUB1 Ub-PA structure and yet it is already active, wouldn't this suggest that the effects of proximal Gln2 on activating OsOTUB1 are not important and hence the entire argument regarding bi-modal activation is incorrect? How are the structures of the two states related to one another? More detailed structural analysis needs to be provided. Again, the inclusion of more quantitative biochemistry is required to deduce the importance of Gln2 for OsOTUB1 activity. Importantly, the authors need to distinguish between reactivity of OsOTUB1 that is mediated through binding of the distal ubiquitin moiety, and activity of OsOTUB1 towards di ubiquitin substrates, which in theory could be dependent on proximal ubiquitin Gln2. Only through k_{cat}/K_M analysis of Ub-AMC cleavage and k_{cat}/K_M measurements on the hydrolysis of Met1 and Lys48-linked di ubiquitin can these assertions be made.

The second part of the manuscript addresses the difference in specificity between hOTUB1 and OsOTUB1. hOTUB1 is specific for hydrolysing Lys48-linked ubiquitin chains, whereas OsOTUB1 has selectivity for Met1 and Lys48-linked chains. Mutations that were introduced in hOTUB1 to facilitate Met1 ubiquitin cleavage had very little effect on hydrolysis of Met1 ubiquitin (Figure 3C), so this reviewer is somewhat sceptical that these mutations can be directly translated into hOTUB1. However, this reviewer acknowledges that it is difficult to obtain gain-of-function mutations to achieve linkage selectivity. Importantly, the controls assessing the effect of these hOTUB1 variants to cleave Lys48-linked ubiquitin are lacking. The phylogenetic analysis of different OTUB1 families is

interesting but Figure 4 is confusing, no description of the sub-panels of Figure 4A is given in the figure legend nor the text. In the first part of Figure 4B, it was impossible to assess the ability of the DUBs to hydrolyse Met1 ubiquitin as no zero-time point is shown for each DUB and the levels of mono ubiquitin vary significantly but do not increase between 5 and 15 minutes, suggesting there is contaminating mono ubiquitin in the di ubiquitin or that the enzyme has ceased functioning during the course of the assay.

From the data statistics provided in Table S1, it seems that the effective resolution extends beyond the reported resolution for the apo and Met1 diUb DHA structures. The authors may wish to further investigate whether their electron density maps can be improved. For the data scaling it is good practice to include CC1/2 values as a metric for data quality and effective resolution. In addition, Ramachandran statistics are not included in Table S1. However, from one PDB validation report submitted with the manuscript, there are a significant number of Ramachandran outliers (4% in one chain). Representative electron density for the outliers should be included in the supplementary figures. However, given that all chains are the same macromolecule and the high degree of variation between all the molecules, this suggests to this reviewer that this structure is not fully refined. The other validation reports had improved model quality statistics.

Minor comments

Abstract mentions that no Met1-selective DUBs have been identified in plants and that the authors identify OTUB1. However, it has been previously reported in the literature that OTUB1 from *Arabidopsis* can hydrolyse Met1 ubiquitin chains see reference (Radjacommaro R. et al., *Front Plant Sci.* 2014).

Naming the versions of the probe from Met1-diUB-1 to Met1-diUb-2 is a little confusing. Could the authors please refer to it as Met1-diUb DHA or something that doesn't read as Ub2?

A lot of the figures omit important information to assess the activity of OsOTUB1, such as concentration of OsOTUB1 used and in some cases length of incubation time (e.g. Figure 1E has no incubation time indicated nor amount of enzyme added).

This reviewer is not a plant expert but wonders what the differences between looking at the levels of Met1 Ub in seedlings is compared to looking at other regions of plants? It would be helpful for a more general audience to explain the rationale for Figure 1D.

Line 136 “FL-OsOTUB1 was not stable (spots on gel)”. To this reviewer, the band corresponding to FL OsOTUB1 looks fine and there are no additional bands on the gel to indicate degradation. The authors should re write this sentence a little more carefully.

For the structural overlays it is hard to deduce which structures are shown, i.e., which PDB accession numbers have been used for the overlay? Often the overall structure is showing only one molecule, but the zoomed-in regions display stick representation of both structures (e.g., see Figure 3). It was hard for the reviewer to work out which residues were from which structure and how well the overall structures superimpose.

hOTUB1 is activated through binding to E2 and inhibited through mono ubiquitin binding (see Wiener R, et al., NSMB 2013 and subsequent publications from the Wolberger group). Does a similar mechanism operate for OsOTUB1 and are these residues conserved amongst plant OTUB1?

Responds to Reviewer #1:

In the negatives, the structural data are not interpreted with the same level of competence. The paper has a number of inconsistencies and overreaching statements that are not adequately supported by experimental evidence, which applies even to some of the primary conclusions in the paper. Thus, the negatives outweigh the positives in the current version of the paper.

Respond:

Thank you for your comments.

Frankly speaking, in the initial version, we failed to sufficiently analyze the related structure of OsOTUB1 and its similarities and differences with the related structures of OTULIN and hOTUB1; we failed to provide sufficient and direct evidence to support the conclusion (such as "proximal ubiquitin activation mechanism", "lack of kinetics evidence"; insufficient evidence for "M1-specific motifs"). Thanks to your suggestions, in the revised version,

① The kinetics data of OsOTUB1 in the presence of different Met1-diUb mutants (Met1-diUb Q2A^{prox}, Met1-diUb Q2N^{prox}, Met1-diUb Q2H^{prox} and Met1-diUb Q2E^{prox}) were measured (Figure 2F and 2G). By further comparing the structures of OsOTUB1~Met1-diUb-DHA and OsOTUB1~Ub-PA (Figure S5) and analyzing the interaction environment in which Q2^{prox} is located (Figure 3B, 3C and S7), we draw the conclusion that, Q2^{prox} affects Met1-diUb's affinity with OsOTUB1, thereby altering the catalysis ability of OsOTUB1, but not hampering Q2^{prox}-mediated activation of OsOTUB1; E mutation on Q2^{prox} will disturb the compatibility between the hydrogen-bonding line in proximal Ub and S1', thus significantly hampering the activity of OsOTUB1.

② The M1-specific mutant of hOTUB1 (hOTUB1-P87G-SGS) was re-expressed and purified, and its ability to hydrolyze Met1-diUb was tested. The results (Figure 4D) showed that, hOTUB1-cat-P87G-SGS can efficiently digest Met1-diUb (about 50% Met1-diUb was digested after 2 hours), while no digestive product was observed for hOTUB1-cat even after 24h.

③ The effect of swapping M1-specific motifs (including N-handle motif and C-handle motif) on the hydrolysis activity of Lys48-diUb by OsOTUB1 and hOTUB1 was observed (Figure S9B and S9C). We found that, C-handle motif (G/P) contributes almost equally to hOTUB1-cat and OsOTUB1-cat in the hydrolysis activity against Lys48-diUb (G favors while P disfavors), while N-handle motif (SGS/EDD) contributes oppositely to that on OsOTUB1-cat and hOTUB1-cat: mutation from EDD to SGS can weaken hOTUB1-cat, but enhance OsOTUB1-cat. In sharp contrast, N-handle motif and C-handle motif are consistent in determining hOTUB1 and OsOTUB1 in the hydrolysis activity against Met1-diUb. Based on this consistency and the fact that M1-specific motifs mutation upon hOTUB1 will significantly endow it with the ability to hydrolyze Met1-diUb (mentioned above in ②), we believe that N-handle motif and C-handle motif are M1-specific motifs.

④ Regarding the selection function mechanism of M1-specific motifs, our results indicated that, the chemical feature of residues in N-handle motif and C-handle motif enables the S1' pocket of OsOTUB1 to accommodate the proximal ubiquitin of Met1-diUb. For SGS in the N-handle

motif, it has neither electrostatic repulsion nor steric hindrance with the proximal ubiquitin. These advantages enable proximal ubiquitin to enter S1'. Therefore, N-handle motif and C-handle motif act jointly as a filter to select ubiquitin chains with different linkage type. This filter has a smaller caliber in hOTUB1, so that it can only accommodate the proximal ubiquitin of Lys48-diUb; while in OsOTUB1, it has a larger caliber, which enables it to accommodate both Lys48 and Met1 chains. Therefore, the N-handle motif in OsOTUB1 works because it allows proximal ubiquitin to enter the S1' pocket, but not because it can bind to proximal ubiquitin.

The above response is only a brief overall response. For a more specific and detailed response, please refer to the followings.

Major Issues:

No explanation was given as to why other DUBs identified in the activity profiling were not considered, even though the score for one of them, OsUCH14 (human USP5 homolog), is higher than OsOTUB1 (Figure 1), the candidate they chose to pursue. It seems the authors abandoned the other DUBs from further analysis because some of them were also identified in a previous activity-based screen in human cells (Weber et al 2017). Somehow the authors favored OTUB1 but no mention of Met1-polyUb activity of UCH14 and UCH-L and biochemical analysis of their DUB properties.

Please include linkage specificity data for OsUCH14, and, if possible, for UCH-L. It might help explain the conundrum in Figure 1D: the knockout is marginally different from the wt seedlings (in Met1-polyUb staining). That could mean that there are other Met1-linked DUBs playing a more prominent role in vivo.

Respond:

Very good question! We forgot to clarify this issue when preparing the manuscript.

We intended to conduct in-depth research on OsOTUB1 due to mainly 2 reasons: (1) the human homologues of OsUCH14 and OsUCH-L are USP5 and UCHL3, respectively. Previous studies have demonstrated that USP5 can cut a total of 6 types of ubiquitin chains including Met1, which means that there is no preferred linkage type for it (Reyes-Turcu, et al., 2008; Dayal, et al., 2009); while UCHL3 mainly removes the C-terminal small peptide conjugates of ubiquitin or short C-terminal extensions of polymeric ubiquitin precursors (Komander et al., 2009). Our *in vitro* experiments have indeed confirmed that OsUCH14 and OsUCH-L have no apparent linkage preference (Fig. S2A). Considering that we want to discover more DUBs with linkage selectivity, OsUCH14 and OsUCH-L are not considered for further study. (2) The human homologue of OsOTUB1 is hOTUB1, which has been proved to strictly digest Lys48 type ubiquitin chains, but not Met1 type (Edelmann, et al., 2009). The unusual selectivity of OsOTUB1 to Met1 type has aroused our great interest. Please refer to the third paragraph in the part of “1. The discovery of plant DUBs that hydrolyze Met1 chains using an efficiently prepared Met1-diUb suicide probe”.

Please explain why do we see enrichment of Poly-Ub and the E1 enzyme. What do these observations indicate about the limitation of the approach?

Respond:

Thank you for your question!

Here, we noticed that in addition to DUBs, PolyUb and E1 were both detected in Weber's work and our study. The mass spectrometry can easily and inevitably detect Ub peptides because the M1-diUb probe covalently crosslinks the target protein. For E1 enzyme, it can not only bind and activate the C-terminus of ubiquitin, but also the proximal Ub in the ubiquitin chains (Van der Heden van Noort et al., 2019; Zheng et al., 2020). Therefore, it is reasonable that Met1-diUb probe can bind and extract E1 enzyme in the lysate. In summary, this method identifies protein candidates by analyzing the peptide sequence of the protein bound by the probe (both covalently bound and non-covalently bound). Therefore, the side effects caused by ubiquitin itself and non-covalently bound proteins cannot be ruled out. When using this method, it is necessary to rely on additional methods such as manual identification and biochemical experimental verification to eliminate false positives as much as possible. Please refer to the last four sentences of the second paragraph in the part of "1. The discovery of plant DUBs that hydrolyze Met1 chains using an efficiently prepared Met1-diUb suicide probe".

It seems the catalytic domain alone is better at cleaving K48-linked diUb than the full-length enzyme (Figure 1F). What is the role of the deleted segment? Please explain.

Respond:

Very good question! We probed the function of the N-terminal extension in OsOTUB1.

In hOTUB1, the N-terminal peptide (A25-N45) forms an α helix which binds to and stabilizes the conformation of the proximal Ub in Lys48-diUb, thereby promoting the cleavage by hOTUB1 (Juang et al., 2013; Wiener et al., 2013), which is also confirmed by our data, both in the presence of hUBCH5B and OsUBCH5B (Fig. S2C). Inspired by this phenomenon, we asked whether activity of OsOTUB1 toward Lys48-diUb can be promoted by OsUBCH5B or hUBCH5B. Therefore, activity of OsOTUB1 against Lys48-diUb was monitored, in the presence of OsUBCH5B or hUBCH5B (25 μ M). The results showed that (Fig. S2C), the activity of OsOTUB1 against Lys48-diUb can be promoted by both hUBCH5B and OsUBCH5B. However, the sequence comparison shows that (Fig. S3), the N-terminal sequence of OsOTUB1 is significantly different from that of hOTUB1. In OsOTUB1, the N-terminal sequence is rich in residues that are prone to form random coils, such as Gly, Ser and Pro, suggesting that this region would not form an α helix to bind and stabilize the proximal Ub in Lys48-diUb. To reveal whether the activity promotion by E2 is dependent on the N-terminus sequence of OsOTUB1, activity of OsOTUB1-cat (OsOTUB1 that deleting N terminus) against Lys48-diUb was monitored, in the presence of OsUBCH5B or hUBCH5B. The results showed that (Fig. S2D), the K48 activity of OsOTUB1-cat can also be promoted by both hUBCH5B and OsUBCH5B. Therefore, stimulation of OsOTUB1 against Lys48-diUb by both hUBCH5B and OsUBCH5B is not (or not completely) dependent on the

N-terminus sequence. The mechanism that OsUBCH5B and hUBCH5B enhance the activity of OsOTUB1 needs to be further revealed. Please refer to the last paragraph in the part of “1. The discovery of plant DUBs that hydrolyze Met1 chains using an efficiently prepared Met1-diUb suicide probe”.

Figure 1D: Please provide corresponding results for K48-linked chains. In the same figure, including the catalytic mutant would help. Without that result, one cannot directly relate the observation to DUB activity. Include data showing levels of protein expression in wt, KO and overexpressed seedlings.

Respond:

Very good advice! We supplemented the effect of overexpression or knockout of OsOTUB1 on the abundance of chains (Figure 1D) and the expression of OsOTUB1 (Figure 1D). Based on three replicates, we used quantification software ImageJ to quantify both Met1 and Lys48 chains. The results showed that,

① Expression of OsOTUB1 was significantly elevated and completely deprived in overexpressed and knockout seedlings, respectively.

② The intensities of the bands corresponding to Ub in wild-type, depleted, and overexpressed OsOTUB1 seedlings were almost the same, indicating that OsOTUB1 does not affect the total Ub in rice. Using the Met1 chains antibody, the intensity of the band corresponding to Met1 chains was significantly increased in OsOTUB1 knockout seedlings, which was about 204% of that of WT, indicating that activity of OsOTUB1 contributes the digestion of Met1 chains *in vivo*. In contrast, the OsOTUB1-overexpressed seedlings showed no obvious decrease of Met1 chains (99% of that of WT), the reason for which is under further investigation.

③ The effect of OsOTUB1 on Lys48 chains was seemingly not significant, since the abundance of Lys48 chains in both OsOTUB1 knockout and overexpression seedlings was similar with that of the wildtype (Figure 1D).

Please refer to the fourth paragraph in the part of “1. The discovery of plant DUBs that hydrolyze Met1 chains using an efficiently prepared Met1-diUb suicide probe”.

Unfortunately, we couldn't introduce inactivating mutations in OsOTUB1, mainly because we currently cannot perform site-directed mutations on specific protein-coding gene at the genetic level. However, our current experimental results have clearly demonstrated that OsOTUB1 can affect the *in vivo* abundance of ubiquitination in rice.

Bi-modal activation:

Please explain clearly the meaning of 'bi-modal activation'. After reading the paper I got the sense that it is a fancy word for the widely observed phenomena of substrate-induced

conformational changes in the vicinity of the active site of the enzyme. I am not a stickler for words, but the word bi-modal does not make any sense in the context of the paper.

From the way the activation model is described it seems conformational changes occur in a sequential fashion: first, the distal Ub-induced alignment of active site residues (His alignment), then 'pushing' by Q2 of the proximal Ub. While this sounds interesting, no evidence is provided.

Respond:

Thank you for your comments. Frankly speaking, in the initial version, this inference sounds interesting, but no evidence is provided.

In order to provide evidence for this inference, we first tried to prove that the distal Ub can activate OsOTUB1 by inducing the conformational change of the catalytic triad, aligning them from inactive state to active state. By resolving the structure of OsOTUB1-cat (Figure 1A and 1B), we find that, in the absence of substrate, the imidazole ring of the catalytic His317 adopts a nearly vertical conformation with Cys121, and $\epsilon^2\text{N}$ is 6.8 Å away from the sulfur atom of C121—much further than the distance between the catalytic His and Cys in the common DUBs (3~5 Å) (Kong et al., 2015; Mevissen et al., 2013), and too far for $\epsilon^2\text{N}$ to deprotonate the catalytic Cys. So the enzyme at this time is in an inactive state. But when it coexists with the substrate, it shows activity, indicating that the substrate causes the rearrangement of the catalytic triad and the enzyme is activated. We investigated whether OsOTUB1-cat could hydrolyze Ub-7-amino-4-methylcoumarin (Ub-AMC), since it was previously reported to be bound by S1 pocket of hOTUB1 (Wiener et al., 2013) and the free coumarin generated through deubiquitination produces readily detectable fluorescence. The results showed that (Fig. 2H), under the action of OsOTUB1-cat, the fluorescence intensity of AMC continued to increase as time extended, demonstrating that OsOTUB1-cat effectively hydrolyzes Ub-AMC. This result was similar with that conducted by UCHL3 which produced more free coumarin as time extended (Fig. 2H) and was previously demonstrated to be activated by distal Ub (Boudreaux et al., 2010). Consistently, OsOTUB1-cat is ready to crosslink with Ub-PA (>90% in 1 minute, Figure 2I) and the structure of OsOTUB1-cat~Ub-PA showed that binding of the distal Ub induce a deflection of approximately 80° for catalytic Cys121 and His317 and a distance reduction from 6.8 Å to 4.1 Å between side chains of the catalytic Cys121 and His317, forming an activated conformation. Therefore, these results confirmed that the distal Ub activates OsOTUB1-cat by inducing major conformational changes. Please refer to the fifth paragraph in the part of “2. Activation of OsOTUB1-cat is induced by distal Ub, but not proximal Ub”.

Subsequently, we tried to prove whether Q2^{prox} promotes the spatial displacement of imidazole ring of catalytic His317. By structural alignment between OsOTUB1-cat~Ub-PA and OsOTUB1-cat~Met1-diUb-DHA (Fig. S5), we find that residues around the catalytic center of OsOTUB1-cat~Ub-PA are already at the almost exact positions with those in the structure of OsOTUB1-cat~Met1-diUb-DHA, seemingly suggesting that imidazole ring of catalytic His317 occurs after binding with distal Ub. However, in the process of OsOTUB1-cat hydrolysis of Met1-diUb, we still cannot rule out the existence of Q2^{prox} promoting the spatial displacement of imidazole ring of catalytic His317, since the inducing activation of distal Ub and the propelling effect of Q2^{prox} (if it does exist) may happen at the same time. To this end, we first tried to

unilaterally limit the activation of distant Ub through mutations, but we were unsuccessful, because these mutations all caused a significant decrease in the binding ability of OsOTUB1-cat to Met1-diUb (data not shown). Considering that if Q2^{prox} could promote the spatial displacement of imidazole ring of catalytic His317 to form an activated conformation, it should show the activation ability of OsOTUB1-cat. Therefore, we investigated the effect of mutations at Q2^{prox} on the activity of OsOTUB1-cat. Here, Q2^{prox} was mutated to residues with short side chain (here are Met1-diUb Q2A^{prox} and Met1-diUb Q2N^{prox}), comparable-length side chain (here is Met1-diUb Q2H^{prox}) or negative charge since it might electrostatically stabilize the imidazole ring of His317 at the inactive state (here is Met1-diUb Q2E^{prox}), following by the enzymatic evaluation. The results showed that (Fig. 2E), compared with wild type Met1-diUb WT (about 50% was hydrolyzed after 1 min), Met1-diUb Q2A^{prox} significantly impairs the activity of OsOTUB1-cat (merely negligible product after 5 min), Met1-diUb Q2N^{prox} moderately reduced the activity of OsOTUB1-cat (about 10% was hydrolyzed after 1 min), Met1-diUb Q2H^{prox} almost recovers the activity of OsOTUB1-cat (about 30% was hydrolyzed after 1 min), while Met1-diUb Q2E^{prox} much more severely impedes the activity of OsOTUB1-cat (about 5% was hydrolyzed even after 30 min). These results indicated that, activity of OsOTUB1-cat is increasing as the length of side chain gradually extends, while much more significantly inhibited when the side chain bears negative charge, seemingly supporting the hypothesis that Q2^{prox} activates OsOTUB1-cat by relocating the imidazole ring of His317. Please refer to the third paragraph in the part of “2. Activation of OsOTUB1-cat is induced by distal Ub, but not proximal Ub”.

However, we still did not dissect the binding function and the catalytic role of Q2^{prox}. Therefore, we measured the kinetics of OsOTUB1-cat in the presence of Met1-diUb WT, Met1-diUb Q2A^{prox}, Met1-diUb Q2N^{prox}, Met1-diUb Q2H^{prox} and Met1-diUb Q2E^{prox}, respectively. The kinetics results showed that (Fig. 2F, G), compared with OsOTUB1-cat in the presence of Met1-diUb WT (0.31 s⁻¹), the turnover number (k_{cat}) of OsOTUB1-cat showed 1.07-fold (0.29 s⁻¹), 2.21-fold (0.14 s⁻¹), 7.21-fold (0.043 s⁻¹) and 413.3-fold reduction (0.0075 s⁻¹) in the presence of Met1-diUb Q2H^{prox}, Met1-diUb Q2N^{prox}, Met1-diUb Q2A^{prox} or Met1-diUb Q2E^{prox}, respectively. This is consistent with the activity of OsOTUB1-cat against different mutants (Fig. 2E), demonstrating that mutation upon Q2^{prox} significantly inhibits the M1 activity of OsOTUB1-cat by reducing its k_{cat} . However, affinity (K_M) of OsOTUB1-cat towards Met1-diUb mutants were not keeping almost unchanged or elevated as seen in OTULIN which was previously demonstrated to be activated by proximal Ub (Keusekotten et al., 2013), but significantly reduced: compared to OsOTUB1-cat with Met1-diUb WT (25.65 μ M), the affinity (K_M) of OsOTUB1-cat showed 1.51-fold (38.76 μ M), 1.69-fold (43.32 μ M), 3.95-fold (101.2 μ M) and 5.11-fold (131.1 μ M) reduction in the presences of Met1-diUb Q2H^{prox}, Met1-diUb Q2N^{prox}, Met1-diUb Q2A^{prox} or Met1-diUb Q2E^{prox}, respectively. These results demonstrated that mutations on Q2^{prox} will affect the affinity of OsOTUB1-cat with substrate, thereby altering the catalysis ability of OsOTUB1-cat, but not hampering Q2^{prox}-mediated activation of OsOTUB1-cat. Please refer to the fourth paragraph in the part of “2. Activation of OsOTUB1-cat is induced by distal Ub, but not proximal Ub”.

Therefore, there is no phenomenon that Q2^{prox} promotes the spatial displacement of imidazole ring of catalytic His317, and thus there is no co-activation of OsOTUB1-cat by distal Ub and proximal Ub.

What is the significance of the bi-modal activation mechanism? Is there a biological implication of this type of diUb recognition?

Respond:

Thank you for your question!

In the modified version, the kinetics results (Figure 2F and 2G) confirmed that there is no dual ubiquitin activation mechanism in OsOTUB1-cat's hydrolysis on Met1-diUb. For detailed discussion, please refer to the reply to your previous question.

I do not agree with 'activation by both distal and proximal ubiquitin' mentioned in the abstract. The crystals structure with Ub-PA shows the catalytic His is already in place due to changes induced upon distal Ub binding. The cavity created by this movement is simply occupied by Q2 of the proximal Ub. So, instead of Q2 'pushing' the histidine, it is merely occupying an empty area near the active site to maximize interactions with enzyme residues. I am not sure if this interpretation is less plausible than the one provided by the authors. If it cannot be tested, the authors should indicate that this and similar arguments as speculative remarks and should include them in discussion rather than in the abstract, where one expects to see only the main, evidence-backed results.

Respond:

Thank you very much for your questions!

In the new version of the manuscript, we re-verified the view of "activation by both distal and proximal ubiquitin" through kinetics experiments, and the results proved that there is indeed no phenomenon that Q2^{prox} promotes the spatial displacement of imidazole ring of catalytic His317, and thus there is no co-activation of OsOTUB1-cat by distal Ub and proximal Ub. For detailed discussion, please refer to the reply to your first question about the "Bi-modal activation".

The observation that in the Ub-PA complex the histidine has already moved to its catalytic competent orientation is a blow to the notion that proximal Ub 'activate' the enzyme. This is a serious inconsistency and should be addressed carefully.

Respond:

Thank you very much for your questions!

In the new version of the manuscript, we re-verified the view of "activation by both distal and proximal ubiquitin" through kinetics experiments, and the results proved that there is indeed no phenomenon that Q2^{prox} promotes the spatial displacement of imidazole ring of catalytic His317, and thus there is no co-activation of OsOTUB1-cat by distal Ub and proximal Ub. For detailed discussion, please refer to the reply to your first question about the "Bi-modal activation".

Please provide experimental evidence in support of 'proximal ubiquitin triggers full activity of

OsOTUB1 and the hydrolysis of Met1 chains'.

Respond:

Thank you for your questions!

In the new version of the manuscript, we re-verified the view of "activation by both distal and proximal ubiquitin" through kinetics experiments, and the results proved that there is indeed no phenomenon that Q2^{prox} promotes the spatial displacement of imidazole ring of catalytic His317, and thus there is no co-activation of OsOTUB1-cat by distal Ub and proximal Ub. For detailed discussion, please refer to the reply to your first question about the "Bi-modal activation".

(Line 203) The Q2E mutational result shows the mutant is impaired worse than the Ala mutant. The authors claim this is due to Glu sidechain making an electrostatic interaction with the catalytic His thereby by holding it in an inactive conformation, which would render activation by proximal Ub more difficult. I suspect this is a biased interpretation. One could argue that the additional negative charge of the Q2E mutant may introduce repulsion near the active-site area (for example, with D319 and even with E16 of proximal Ub) explaining the loss of activity.

Respond:

Thank you very much for your questions! Briefly, the reason why Q2E can cause a severe decrease in OsOTUB1-cat's M1 activity is that the negative charge introduced by Q2E may seriously interfere with the compatibility between the hydrogen bonding line in proximal Ub and S1' pocket. Please refer to the following for a more detailed response.

Initially, we carefully rechecked the interactions between proximal Ub and S1' pocket. In the S1' pocket, hydrogen bonds contribute main interactions between OsOTUB1-cat and proximal Ub of Met1-diUb (Fig. 3B, S7). As the interaction surface spans a large area (approximately 26Å×14Å), we separated it into three sections, named top-, middle- and bottom-handlers, representing interaction surfaces that happening in $\alpha 1$, $\beta 1$ and loop ($\beta 2$ - $\alpha 1$), and loop ($\beta 4$ - $\beta 5$) of proximal Ub, respectively (Fig. S7). Except for side chain of R116 in OsOTUB1-cat interacts with side chain of E64^{prox} in the bottom handler, all the interactions happen in the top- and middle-handlers (Fig. 3B). In the top- and middle-handlers, side chains of E90, R114, R314, D319, T115 and Y273 in OsOTUB1-cat interact with side chains of E16^{prox}, E18^{prox}, K33^{prox} and Q2^{prox} via network of hydrogen bonds, directly or indirectly, including a) Oxygen of the amide group of Q2^{prox} forms a hydrogen bond with hydroxyl of T115 through a water molecule (w1), b) carboxyl of E16^{prox} forms a hydrogen bond directly with R314, and indirectly with R114 and E90 via a water molecule (w2), c) carboxyl of E18^{prox} forms a hydrogen bond with phenolic hydroxyl of Y273 via a water molecule (w3), d) amino of K33^{prox} interacts with E90. Please refer to the third paragraph in the part of "3. Binding of proximal Ub facilitates the Met1 type preference of OsOTUB1-cat".

Furthermore, in top- and middle-handlers, intramolecular interactions also happen, including a) carboxyl of E16^{prox} simultaneously interacts with amino of K33^{prox} and Q2^{prox} by hydrogen bonds, forming a hydrogen bonding line in the proximal Ub, arranging in the order of

basic-acidic-basic, b) guanidyl of R314 interacts with carboxyl of E90 and D319 simultaneously by hydrogen bonds, forming a hydrogen bonding line in the S1' pocket, arranging in the order of acidic-basic-acidic. According to the residue arrangement, it may be reasonable that the compatibility between the two lines is the key for OsOTUB1-cat's efficient binding with Met1-diUb. Mutation of any residue that changes the charge property would significantly affect OsOTUB1-cat's M1 activity. Please refer to the fourth paragraph in the part of "3. Binding of proximal Ub facilitates the Met1 type preference of OsOTUB1-cat".

Finally, to verify these speculations, Q2^{prox}, E16^{prox}, E18^{prox}, K33^{prox} and E64^{prox} were individually mutated and the hydrolytic activity of OsOTUB1-cat toward these mutants was measured and compared (Fig. 3C). The results showed that, mutations upon Q2^{prox}, E16^{prox}, E18^{prox}, K33^{prox} and E64^{prox} will affect the hydrolytic activity of OsOTUB1-cat against them to varying degrees, and the digestive preference order of OsOTUB1-cat is Met1-diUb WT≈Met1-diUb E18A^{prox}>Met1-diUb E16Q^{prox}≈Met1-diUb E16A^{prox}>Met1-diUb E64A^{prox}>Met1-diUb Q2A^{prox}>Met1-diUb K33A^{prox}>Met1-diUb Q2E^{prox}>Met1-diUb K33E^{prox}. These results support the speculation that binding of proximal Ub by S1' pocket of OsOTUB1-cat is important for its M1 activity and compatibility between the hydrogen bonding lines in OsOTUB1-cat and proximal Ub is vital for OsOTUB1-cat's efficient binding and cleavage of Met1-diUb (etc. Q2E^{prox} and K33E^{prox}). Please refer to the fifth paragraph in the part of "3. Binding of proximal Ub facilitates the Met1 type preference of OsOTUB1-cat".

With regard to Figure 2G, this is exactly what I would expect for a DUB that has activity toward both K48 and Met1-linked diubiquitins. As the authors point out, it is indeed distinct from OTULIN. The reactivity with Ub-PA (and Ub-AMC) cannot be and should not be taken as evidence in favor of activation by distal Ub. Such reactivity is seen in a wide variety of DUBs bearing a catalytic Cys.

Respond:

I'm sorry I can't agree with you.

Distal ubiquitin can induce conformational changes of some DUBs, causing their catalytic triad to rearrange and then be activated. This is a proven phenomenon (Abdul Rehman et al., 2016; Boudreaux et al., 2010; Reyes-Turcu et al., 2006). This phenomenon is also well reflected in our manuscript: in the absence of substrate, the imidazole ring of the catalytic His317 adopts a nearly vertical conformation with Cys121, and $\epsilon^2\text{N}$ is 6.8 Å away from the sulfur atom of C121—much further than the distance between the catalytic His and Cys in the common DUBs (3~5 Å) (Kong et al., 2015; Mevissen et al., 2013), and too far for $\epsilon^2\text{N}$ to deprotonate the catalytic Cys. So the enzyme at this time is in an inactive state. But when it coexists with the substrate, it shows activity, indicating that the substrate causes the rearrangement of the catalytic triad and the enzyme is activated. we investigated whether OsOTUB1-cat could hydrolyze Ub-7-amino-4-methylcoumarin (Ub-AMC), since it was previously reported to be bound by S1 pocket of hOTUB1 (Wiener et al., 2013) and the free coumarin generated through deubiquitination produces readily detectable fluorescence. The results showed that (Fig. 2H), under the action of OsOTUB1-cat, the fluorescence intensity of AMC continued to increase as time extended,

demonstrating that OsOTUB1-cat effectively hydrolyzes Ub-AMC. This result was similar with that conducted by UCHL3 which produced more free coumarin as time extended (Fig. 2H) and is previously demonstrated to be activated by distal Ub (Boudreaux et al., 2010). These results proved that OsOTUB1-cat can also be activated by distal Ub.

It should be pointed out that although most members of the OTU family, except for OTULIN, can hydrolyze Ub-AMC, not all of them need to be activated. For example, the catalytic triad of human OTUB2 is already aligned to be active state that does not need substrate activation (Nanao et al., EMBO Rep (2004) 5:783-788), while the human OTUB1 requires substrate activation (Edelmann et al., 2009).

M1-specific Motif

In my view, the M1-specific motif has not been validated adequately. Please consider mutating the residues and asking if there are differential changes in hydrolysis of the Met1-linked linked substrate versus the K48-linked one. One expects the overall activity of the mutants to be lower than wt, but its specificity might be altered if the residues do indeed determine Met1-diub specificity.

Respond:

Thank you for your question! In response to it, we have supplemented the following experiments and obtained the following conclusions/inferences:

1. We investigated the effects of M1-specific motifs mutations in hOTUB1-cat and OsOTUB1-cat on their hydrolysis activity against Lys48-diUb. The results showed that (Fig. S9), C-handle motif (G/P) contributes almost equally to hOTUB1-cat and OsOTUB1-cat in the hydrolysis activity against Lys48-diUb (G favors while P disfavors), while the effect of N-handle motif (SGS/EDD) on hOTUB1-cat is opposite to that on OsOTUB1-cat: mutation from EDD to SGS can weaken hOTUB1-cat, but enhance OsOTUB1-cat. These results indicated that hOTUB1-cat adopts different way from OsOTUB1-cat to bind to and digest Lys48-diUb. In sharp contrast, N-handle motif and C-handle motif are consistent in determining hOTUB1 and OsOTUB1 in the hydrolysis activity against Met1-diUb. Based on this consistency and the fact that M1-specific motifs mutation upon hOTUB1 will significantly endow it with the ability to hydrolyze Met1-diUb (Figure 4D), we believe that N-handle motif and C-handle motif are M1-specific motifs.
2. Furthermore, we proved that introducing P87G-SGS into hOTUB1 can endow it a preference for the Met1 chain instead of Lys48. Given that residues in S1' pocket involved in binding to proximal Ub are also conserved in hOTUB1 (Figure S3), we wondered whether the mutant of hOTUB1(P87G-SGS) that allows proximal Ub to enter S1' shows hydrolysis preference for Met1 linkage, as OsOTUB1(87G-SGS) does. Therefore, we compared the hydrolysis activity of hOTUB1-P87G-SGS against Lys48-diUb and Met1-diUb (Fig. 4G). Under the same conditions, about 20% Met1-diUb was digested within 30 min, while no more than 10% Lys48-diUb was excised within 30 min, proving preference of hOTUB1-P87G-SGS for Met1 linkage. This result implies that hOTUB1 has the potential to cleave Met1 type ubiquitin

chain, due to the conservation of residues responsible for binding to proximal Ub in Met1-diUb (Fig. S3), while N- and C-handle motifs control this switch. Please refer to the sixth paragraph in the part of “4. Two motifs in S1' pocket determines Met1 type preference of OsOTUB1”.

3. In summary, OsOTUB1 may cleave Met1 chains through a two-step adaptation model (Figure 5). The first step is permission. In this step, along with S1 pocket accommodating the distal Ub, the N-handle motif conforming to XGY sequence pattern (SGS) and the C-handle motif confined to Gly enable proximal Ub in Met1-diUb to enter into S1' pocket of OsOUTB1. In contrast, in hOTUB1, EDD in N-handle motif that violates the XGY sequence pattern and P87 with bulk side chain in C-handle motif jointly restrict proximal Ub entry into S1' pocket. The second step is attraction and compatibility. In this step, the intermolecular interaction between S1' pocket and proximal Ub, together with compatibility between the hydrogen bonding line in S1' pocket and that in proximal Ub, confer OsOTUB1 with the preference for Met1 type. Please refer to the seventh paragraph in the part of “4. Two motifs in S1' pocket determines Met1 type preference of OsOTUB1”.

If the distal and proximal Ub binding sites are conserved between both human OTUB1 and OsOTUB1, how are they so different?

Respond:

Thank you for your question.

The binding sites in hOTUB1 and OsOTUB1 that bind to distal ubiquitin are conserved (Figure S3). This is consistent with the conclusion that most members of the OTU family have relatively conservative distal ubiquitin binding pockets (S1 pockets) (Mevisen et al., 2013). However, there are significant differences in the proximal ubiquitin binding pockets (S1' pockets). Among them, the sequence difference between N-handle motif and C-handle motif is particularly significant (EDD^{hOTUB1} versus SGS^{OsOTUB1}, P^{hOTUB1} versus G^{OsOTUB1}, Fig. S8). Our results did prove the role of these two motifs in determining M1 activity. We re-expressed and purified the protein, and repeated the Met1-diUb hydrolysis assay (Fig. 4D). The results showed that wildtype hOTUB1 did not have the ability to hydrolyze Met1-diUb at all (no product is detected after 24h); similarly, the single mutation of C-handle motif (P87G) and of N-handle motif (SGS) could not confer hOTUB1 the ability to hydrolyze Met1-diUb (no hydrolysis product is detected after 24h). But when C-handle motif and N-handle motif were mutated at the same time (P87G-SGS), hOTUB1 could hydrolyze ~20% of Met1-diUb in 30 minutes and ~50% of Met1-diUb in 2 hours. These results proved that changes in C-handle motif and N-handle motif can indeed endow hOTUB1 with the ability to hydrolyze Met1-diUb. Please refer to the first paragraph in the part of “4. Two motifs in S1' pocket determines Met1 type preference of OsOTUB1”.

Regarding the selection mechanism of M1-specific motifs, our results indicated that, the chemical feature of residues in N-handle motif and C-handle motif enables the S1' pocket of OsOTUB1 to accommodate the proximal ubiquitin of Met1-diUb. For SGS in the N-handle motif, it has neither electrostatic repulsion nor steric hindrance with the proximal ubiquitin. These

advantages enable proximal ubiquitin to enter S1'. Therefore, N-handle motif and C-handle motif act jointly as a filter to select ubiquitin chains with different linkage type. This filter has a smaller caliber in hOTUB1, so that it can only accommodate the proximal ubiquitin of Lys48-diUb; while in OsOTUB1, it has a larger caliber, which enables it to accommodate both Lys48 and Met1 chains. Therefore, the N-handle motif in OsOTUB1 works because it allows proximal ubiquitin to enter the S1' pocket, but not because it can bind to proximal ubiquitin.

It is hard to understand that the SGS motif (N-handle motif), which barely has any interaction with the proximal Ub (figure 3F), will have a role in conferring M1 hydrolytic activity.

Respond:

Thank you for your question! In response to this question, the replies are as follows:

1. The chemical feature of residues in N-handle motif and C-handle motif enables the S1' pocket of OsOTUB1 to accommodate the proximal ubiquitin of Met1-diUb. For SGS in the N-handle motif, it has neither electrostatic repulsion nor steric hindrance with the proximal ubiquitin. These advantages enable proximal ubiquitin to enter S1'. Therefore, N-handle motif and C-handle motif act jointly as a filter to select ubiquitin chains with different linkage type. This filter has a smaller caliber in hOTUB1, so that it can only accommodate the proximal ubiquitin of Lys48-diUb; while in OsOTUB1, it has a larger caliber, which enables it to accommodate both Lys48 and Met1 chains. Therefore, the N-handle motif in OsOTUB1 works because it allows proximal ubiquitin to enter the S1' pocket, but not because it can bind to proximal ubiquitin.
2. Our results have indeed proved that, in addition to N-handle motif and C-handle motif, the binding of S1' pocket to proximal ubiquitin also determines the Met1 selectivity of OsOTUB1-cat. **First, we carefully investigated the interactions between OsOTUB1-cat and proximal Ub of Met1-diUb (Fig. 3B).** We found that ① side chains of E90, R114, R314, D319, T115, Y273 and R116 in OsOTUB1-cat interact with side chains of E16^{prox}, E18^{prox}, K33^{prox}, Q2^{prox} and E64^{prox} via network of hydrogen bonds, directly or indirectly (via water molecules), ② in top- and middle-handlers, intramolecular interactions also happen, including a) carboxyl of E16^{prox} simultaneously interacts with amino of K33^{prox} and Q2^{prox} by hydrogen bonds, forming a hydrogen bonding line in the proximal Ub, arranging in the order of basic-acidic-basic, b) guanidyl of R314 interacts with carboxyl of E90 and D319 simultaneously by hydrogen bonds, forming a hydrogen bonding line in the S1' pocket, arranging in the order of acidic-basic-acidic. According to the residue arrangement, it may be reasonable that the compatibility between the two lines is the key for OsOTUB1-cat's efficient binding with Met1-diUb. Mutation of any residue that changes the charge property would significantly affect OsOTUB1-cat's M1 activity. **Second, to verify these speculations, Q2^{prox}, E16^{prox}, E18^{prox}, K33^{prox} and E64^{prox} were individually mutated and the hydrolytic activity of OsOTUB1-cat toward these mutants was measured and compared (Fig. 3C).** The results showed that, mutations upon Q2^{prox}, E16^{prox}, E18^{prox}, K33^{prox} and E64^{prox} will affect the hydrolytic activity of OsOTUB1-cat against them to varying degrees, and the digestive preference order of OsOTUB1-cat is Met1-diUb WT \approx Met1-diUb E18A^{prox}>Met1-diUb E16Q^{prox} \approx Met1-diUb E16A^{prox}>Met1-diUb E64A^{prox}>Met1-diUb

Q2A^{prox}>Met1-diUb K33A^{prox}>Met1-diUb Q2E^{prox}>Met1-diUb K33E^{prox}. These results support the speculation that binding of proximal Ub by S1' pocket of OsOTUB1-cat is important for its M1 activity and compatibility between the hydrogen bonding lines in OsOTUB1-cat and proximal Ub is vital for OsOTUB1-cat's efficient binding and cleavage of Met1-diUb (etc. Q2E^{prox} and K33E^{prox}). Please refer to the third, fourth and fifth paragraph in the part of "3. Binding of proximal Ub facilitates the Met1 type preference of OsOTUB1-cat".

Minor points:

What are the interactions with Met1 of the proximal Ub?

Respond:

Thank you for your question!

We have supplemented the analysis of the surrounding environment of M1^{prox} in the detailed structure of S1' binding to proximal ubiquitin. It showed that (Figure 3B), M1^{prox} barely interacts with OsOTUB1, except for a hydrogen bond formed between its backbone amino and the carboxyl of G316. Side chain of M1^{prox} otherwise projects to the interior of proximal Ub, similar with M1^{prox} in the structure of OTULIN(C129A)-Met1-diUb (Figure S7). Please refer to the last two sentences of the third paragraph in the part of "3. Binding of proximal Ub facilitates the Met1 type preference of OsOTUB1-cat".

Please consider showing the faint Ub band in Figure 3C is indeed the Ub product resulting from Gly76-Met1 peptide bond cleavage.

Respond:

Thank you for your question!

In response to this question, we re-expressed and purified the protein, and repeated the Met1-diUb hydrolysis assay. The results showed that (Fig. 4D), compared to wildtype hOTUB1 and the single mutation of C-handle motif (P87G) and of N-handle motif (SGS) (no hydrolysis product is detected even after 24h), hOTUB1-P87G-SGS could hydrolyze ~20% of Met1-diUb in 30 minutes and ~50% of Met1-diUb in 2 hours, which is far more than "the faint Ub band" and may need no other evidences to show the band is indeed the Ub product resulting from Gly76-Met1 peptide bond cleavage. Please refer to the first paragraph in the part of "4. Two motifs in S1' pocket determines Met1 type preference of OsOTUB1".

Please explain the rationale behind probing longer poly-Ub chains in Figure 3E.

Respond:

Thank you for your question! We have ignored the explanation of this question. Figure 3E has been changed to Figure S9A.

This is mainly because the long-chain polyUb was reported previously and confirmed by our preliminary assay to be more liable to DUB, thus being more sensitive for the detection of hydrolytic activity. Here, the Met1-tetraUb was used as the substrate. The purpose of this experiment was to compare the effects of the N-handle and C-handle motifs on OsOTUB1-cat's M1 activity. The results (Fig. S9A) demonstrated that the C-handle motif more significantly influences the M1 activity of OsOTUB1-cat than the N-handle motif, consistent with the results shown in Figure 4C. Based on this conclusion, we sequentially classified the OUTBs in the following manuscript and finally obtained the G-only group (Fig. 5A-ii). Please refer to the fourth paragraph in the part of "4. Two motifs in S1' pocket determines Met1 type preference of OsOTUB1".

Please change the wordings of 'Two-orientation activation mode of OsOTUB1-cat to Met1 chains'. As a subheading it does not represent the text that follows.

Respond:

Thank you for your suggestion! Based on the kinetics results, we have changed the title to "Activation of OsOTUB1-cat is induced by distal Ub, but not proximal Ub". Please check the revised manuscript.

Overall citation to previous work is incomplete. Some examples:

- 1. Include citations to distal Ub and proximal Ub conformational changes induced upon Ub binding as observed in other family of DUBs, such as USP, UCH and OTU.*
- 2. I think the reference to the first synthesis of dehydroalanine Met1-DiUb using the same desulfurization method would be appropriate (Haj-Yahya N, et al, 2014) in addition to the one provided.*

Respond:

Thank you for your suggestions! We have searched and added relevant references. Please refer to the second sentence of the second paragraph in the part of "Introduction" and the eighth sentence of the first paragraph in the part of "1. The discovery of plant DUBs that hydrolyze Met1 chains using an efficiently prepared Met1-diUb suicide probe".

Reviewer #2 (Remarks to the Author):

In this manuscript the authors describe a novel molecular mechanism by which a OTUB1 deubiquitinating enzyme from *Oryza sativa* possesses activity against linear Met1-linked ubiquitin chains. So far cleavage of Met1-Ub linkages has been restricted to human OTULIN, another member of the OTU subfamily of DUBs, but in plants this activity seems to be present in OTUB1. The manuscript also explains the structural determinants for the differences between human OTUB1, which has only activity against K48 and K63 Ub chains, and plant OTUB1, in which the presence of particular structural elements allows cleavage of Met1-Ub substrates. The results presented in this manuscript are interesting and the structural analysis is well done, however there are particular issues that should be addressed.

Major Points:

From a mechanistic point of view, it is confusing the distinction between the activation by distal or by proximal Ub. OsOTUB1 can be activated by a distal Ub alone, which explains the activity for Ub-AMC substrates, but in the absence of the Gln2 residue from the proximal Ub, which seemed necessary to align the catalytic triad, as the authors report. The authors should clarify better this point. What's the relevance of Gln2 ?? . Maybe it is only needed to cleave Met1-Ub substrates ?.

Respond:

Thank you very much for your questions! These questions helped us to timely and effectively revise the important conclusions in the manuscript!

The proximal ubiquitin activation mechanism proposed in the manuscript lacks direct evidence, which is mainly due to the failure to provide kinetics data that can dissect the binding function and the catalytic role of Q2^{prox}. For this reason, we used HPLC quantitative method to accurately calculate the kinetics data of OsOTUB1-cat in the presence of different Met1-diUb mutants (Fig. 2F, G). The kinetics results showed that, compared with OsOTUB1-cat in the presence of Met1-diUb WT (0.31 s⁻¹), the turnover number (k_{cat}) of OsOTUB1-cat show 1.07-fold (0.29 s⁻¹), 2.21-fold (0.14 s⁻¹), 7.21-fold (0.043 s⁻¹) and 413.3-fold reduction (0.0075 s⁻¹) in the presence of Met1-diUb Q2H^{prox}, Met1-diUb Q2N^{prox}, Met1-diUb Q2A^{prox} or Met1-diUb Q2E^{prox}, respectively. This is consistent with the activity of OsOTUB1-cat against different mutants (Fig. 2E), demonstrating that mutation upon Q2^{prox} significantly inhibits the M1 activity of OsOTUB1-cat by reducing its k_{cat} . However, affinity (K_M) of OsOTUB1-cat towards Met1-diUb mutants were not keeping almost unchanged or elevated as seen in OTULIN which was previously demonstrated to be activated by proximal Ub (Keusekotten et al., 2013), but significantly reduced: compared to OsOTUB1-cat with Met1-diUb WT (25.65 μM), the affinity (K_M) of OsOTUB1-cat showed 1.51-fold (38.76 μM), 1.69-fold (43.32 μM), 3.95-fold (101.2 μM) and 5.11-fold (131.1 μM) reduction in the presence of Met1-diUb Q2H^{prox}, Met1-diUb Q2N^{prox}, Met1-diUb Q2A^{prox} or Met1-diUb Q2E^{prox}, respectively. These results demonstrated that mutations on Q2^{prox} will affect the affinity of OsOTUB1-cat with substrate, thereby altering the catalysis ability of OsOTUB1-cat, but not hampering Q2^{prox}-mediated activation of OsOTUB1-cat. Please refer to the fourth paragraph in the part of “2. Activation of OsOTUB1-cat is induced by distal Ub, but not proximal

Ub”.

In line with the conclusion that OsOTUB1-cat is induced to be active by distal Ub, but not proximal Ub, we find that residues around the catalytic center of OsOTUB1-cat~Ub-PA are already at the almost exact positions with those in the structure of OsOTUB1-cat~Met1-diUb-DHA (Fig. S5). Please refer to the last sentence of the fifth paragraph in the part of “2. Activation of OsOTUB1-cat is induced by distal Ub, but not proximal Ub”.

Q2's real role is to participate in the interaction between proximal ubiquitin and S1' pocket, which was confirmed in further analysis of the detailed structure of the S1' pocket bound to the proximal ubiquitin (Figure 3B). In the S1' pocket, hydrogen bonds contribute main interactions between OsOTUB1-cat and proximal Ub of Met1-diUb (Fig. 3B, S7). As the interaction surface spans a large area (approximately 26Å×14Å), we separated it into three sections, named top-, middle- and bottom-handlers, representing interaction surfaces that happening in $\alpha 1$, $\beta 1$ and loop ($\beta 2$ - $\alpha 1$), and loop ($\beta 4$ - $\beta 5$) of proximal Ub, respectively (Fig. S7). Except for side chain of R116 in OsOTUB1-cat interacts with side chain of E64^{prox} in the bottom handler, all the interactions happen in the top- and middle-handlers (Fig. 3B). In the top- and middle-handlers, side chains of E90, R114, R314, D319, T115 and Y273 in OsOTUB1-cat interact with side chains of E16^{prox}, E18^{prox}, K33^{prox} and Q2^{prox} via network of hydrogen bonds, directly or indirectly, including a) Oxygen of the amide group of Q2^{prox} forms a hydrogen bond with hydroxyl of T115 through a water molecule (w1), b) carboxyl of E16^{prox} forms a hydrogen bond directly with R314, and indirectly with R114 and E90 via a water molecule (w2), c) carboxyl of E18^{prox} forms a hydrogen bond with phenolic hydroxyl of Y273 via a water molecule (w3), d) amino of K33^{prox} interacts with E90. Please refer to the third paragraph in the part of “3. Binding of proximal Ub facilitates the Met1 type preference of OsOTUB1-cat”.

Furthermore, in top- and middle-handlers, intramolecular interactions also happen, including a) carboxyl of E16^{prox} simultaneously interacts with amino of K33^{prox} and Q2^{prox} by hydrogen bonds, forming a hydrogen bonding line in the proximal Ub, arranging in the order of basic-acidic-basic, b) guanidyl of R314 interacts with carboxyl of E90 and D319 simultaneously by hydrogen bonds, forming a hydrogen bonding line in the S1' pocket, arranging in the order of acidic-basic-acidic. According to the residue arrangement, it may be reasonable that the compatibility between the two lines is the key for OsOTUB1-cat's efficient binding with Met1-diUb. Mutation of any residue that changes the charge property would significantly affect OsOTUB1-cat's M1 activity. Please refer to the fourth paragraph in the part of “3. Binding of proximal Ub facilitates the Met1 type preference of OsOTUB1-cat”.

To verify these speculations, Q2^{prox}, E16^{prox}, E18^{prox}, K33^{prox} and E64^{prox} were individually mutated and the hydrolytic activity of OsOTUB1-cat toward these mutants was measured and compared (Fig. 3C). The results showed that, mutations upon Q2^{prox}, E16^{prox}, E18^{prox}, K33^{prox} and E64^{prox} will affect the hydrolytic activity of OsOTUB1-cat against them to varying degrees, and the digestive preference order of OsOTUB1-cat is Met1-diUb WT \approx Met1-diUb E18A^{prox}>Met1-diUb E16Q^{prox} \approx Met1-diUb E16A^{prox}>Met1-diUb E64A^{prox}>Met1-diUb Q2A^{prox}>Met1-diUb K33A^{prox}>Met1-diUb Q2E^{prox}>Met1-diUb K33E^{prox}. These results support the speculation that binding of proximal Ub by S1' pocket of OsOTUB1-cat is important for its M1 activity and compatibility between the hydrogen bonding lines in OsOTUB1-cat and proximal

Ub is vital for OsOTUB1-cat's efficient binding and cleavage of Met1-diUb (etc. Q2E^{prox} and K33E^{prox}). Please refer to the fifth paragraph in the part of "3. Binding of proximal Ub facilitates the Met1 type preference of OsOTUB1-cat".

Paragraph 115-124. The author conclusions' from western blot reaction in figure 1D are difficult to observe. The different band intensities between lanes in anti-M1Ub are not clear at all. Some statistical quantification should be done.

Additionally it would be interesting to observe the endogenous levels of OsUTUB1 in WT, KO and OE mutants. From these data, the levels of linear Met1Ub do not seem very affected by OsOTUB1, as observed in the WB.

Response:

Very good advice!

Based on three replicates, we used quantification software ImageJ to quantify Met1 chains, and supplemented the effect of overexpression or knockout of OsOTUB1 on the expression of OsOTUB1 (Figure 1D). The results showed that,

① Expression of OsOTUB1 was significantly elevated and completely deprived in overexpressed and knockout seedlings, respectively.

② The intensities of the bands corresponding to Ub in wild-type, depleted, and overexpressed OsOTUB1 seedlings were almost the same, indicating that OsOTUB1 does not affect the total Ub in rice. Using the Met1 chains antibody, the intensity of the band corresponding to Met1 chains was significantly increased in OsOTUB1 knockout seedlings, which was about 204% of that of WT, indicating that activity of OsOTUB1 contributes the digestion of Met1 chains *in vivo*. The intensities of the bands corresponding to Ub in wild-type, depleted, and overexpressed OsOTUB1 seedlings were almost the same, indicating that OsOTUB1 does not affect the total Ub in rice. Using the Met1 chains antibody, the intensity of the band corresponding to Met1 chains was significantly increased in OsOTUB1 knockout seedlings, which was about 204% of that of WT, indicating that activity of OsOTUB1 contributes the digestion of Met1 chains *in vivo*. In contrast, the OsOTUB1-overexpressed seedlings showed no obvious decrease of Met1 chains (99% of that of WT), the reason for which is under further investigation.

③ The effect of OsOTUB1 on Lys48 chains was seemingly not significant, since the abundance of Lys48 chains in both OsOTUB1 knockout and overexpression seedlings was similar with that of the wildtype (Figure 1D).

Please refer to the fourth paragraph in the part of "1. The discovery of plant DUBs that hydrolyze Met1 chains using an efficiently prepared Met1-diUb suicide probe".

Figure 1F. Please load molecular weight markers in the SDS-PAGE, otherwise it is difficult to

correlate the bands with the MW. It is particular relevant when you discuss on the deletion of the N-terminal extension. Have the author's checked whether there is a role for the N-terminal extension in the activity of OsOTUB1 ?

Respond:

Thank you for your suggestion and the good question! We have added a protein marker in Figure 1F and probed the function of the N-terminal extension in OsOTUB1.

In hOTUB1, the N-terminal peptide (A25-N45) forms an α helix which binds to and stabilizes the conformation of the proximal Ub in Lys48-diUb, thereby promoting the cleavage by hOTUB1 (Juang et al., 2013; Wiener et al., 2013), which is also confirmed by our data, both in the presence of hUBCH5B and OsUBCH5B (Fig. S2C). Inspired by this phenomenon, we asked whether activity of OsOTUB1 toward Lys48-diUb can be promoted by OsUBCH5B or hUBCH5B. Therefore, activity of OsOTUB1 against Lys48-diUb was monitored, in the presence of OsUBCH5B or hUBCH5B (25 μ M). The results showed that (Fig. S2C), the activity of OsOTUB1 against Lys48-diUb can be promoted by both hUBCH5B and OsUBCH5B. However, the sequence comparison shows that (Fig. S3), the N-terminal sequence of OsOTUB1 is significantly different from that of hOTUB1. In OsOTUB1, the N-terminal sequence is rich in residues that are prone to form random coils, such as Gly, Ser and Pro, suggesting that this region would not form an α helix to bind and stabilize the proximal Ub in Lys48-diUb. To reveal whether the activity promotion by E2 is dependent on the N-terminus sequence of OsOTUB1, activity of OsOTUB1-cat (OsOTUB1 that deleting N terminus) against Lys48-diUb was monitored, in the presence of OsUBCH5B or hUBCH5B. The results showed that (Fig. S2D), the K48 activity of OsOTUB1-cat can also be promoted by both hUBCH5B and OsUBCH5B. Therefore, stimulation of OsOTUB1 against Lys48-diUb by both hUBCH5B and OsUBCH5B is not (or not completely) dependent on the N-terminus sequence. The mechanism that OsUBCH5B and hUBCH5B enhance the activity of OsOTUB1 needs to be further revealed. Please refer to the last paragraph in the part of "1. The discovery of plant DUBs that hydrolyze Met1 chains using an efficiently prepared Met1-diUb suicide probe".

As stated previously, if distal-Ub alone can rearrange the active site of OsOTUB1, then what is really the relevance of Ub proximal Gln2 ? From the activity assays in figure 2E, Q2 seems essential for linear diMIUb cleavage, but not for hydrolysis of Ub-AMC ??.

Respond:

Thank you for your question. Briefly, in the modified version, the kinetics results (Figure 2F and 2G) confirmed that, during the process of OsOTUB1-cat hydrolyzing Met1-diUb, only the distal ubiquitin activation mechanism exists, and there is no Q2^{prox}-mediated proximal ubiquitin activation mechanism. Q2's real role is to participate in the interaction between proximal ubiquitin and S1' pocket. Please refer to the following for a more detailed response.

The proximal ubiquitin activation mechanism proposed in the manuscript lacks direct evidence, which is mainly due to the failure to provide kinetic data that can dissect the binding

function and the catalytic role of Q2^{prox}. For this reason, we used HPLC quantitative method to accurately calculate the kinetics data of OsOTUB1-cat in the presence of different Met1-diUb mutants (Fig. 2F, G). The kinetics results showed that (Fig. 2F, G), compared with OsOTUB1-cat in the presence of Met1-diUb WT (0.31 s⁻¹), the turnover number (k_{cat}) of OsOTUB1-cat show 1.07-fold (0.29 s⁻¹), 2.21-fold (0.14 s⁻¹), 7.21-fold (0.043 s⁻¹) and 413.3-fold reduction (0.0075 s⁻¹) in the presence of Met1-diUb Q2H^{prox}, Met1-diUb Q2N^{prox}, Met1-diUb Q2A^{prox} or Met1-diUb Q2E^{prox}, respectively. This is consistent with the activity of OsOTUB1-cat against different mutants (Fig. 2E), demonstrating that mutation upon Q2^{prox} significantly inhibits the M1 activity of OsOTUB1-cat by reducing its k_{cat} . However, affinity (K_M) of OsOTUB1-cat towards Met1-diUb mutants were not keeping almost unchanged or elevated as seen in OTULIN which was previously demonstrated to be activated by proximal Ub (Keusekotten et al., 2013), but significantly reduced: compared to OsOTUB1-cat with Met1-diUb WT (25.65 μ M), the affinity (K_M) of OsOTUB1-cat showed 1.51-fold (38.76 μ M), 1.69-fold (43.32 μ M), 3.95-fold (101.2 μ M) and 5.11-fold (131.1 μ M) reduction in the presence of Met1-diUb Q2H^{prox}, Met1-diUb Q2N^{prox}, Met1-diUb Q2A^{prox} or Met1-diUb Q2E^{prox}, respectively. These results demonstrated that mutations on Q2^{prox} will affect the affinity of OsOTUB1-cat with substrate, thereby altering the catalysis ability of OsOTUB1-cat, but not hampering Q2^{prox}-mediated activation of OsOTUB1-cat. Please refer to the fourth paragraph in the part of “2. Activation of OsOTUB1-cat is induced by distal Ub, but not proximal Ub”. In line with the conclusion that OsOTUB1-cat is induced to be active by distal Ub, but not proximal Ub, we find that residues around the catalytic center of OsOTUB1-cat~Ub-PA are already at the almost exact positions with those in the structure of OsOTUB1-cat~Met1-diUb-DHA (Fig. S5). Please refer to the last sentence of the fifth paragraph in the part of “2. Activation of OsOTUB1-cat is induced by distal Ub, but not proximal Ub”.

Q2's real role is to participate in the interaction between proximal ubiquitin and S1' pocket, which was confirmed in further analysis of the detailed structure of the S1' pocket bound to the proximal ubiquitin (Figure 3B). In the S1' pocket, hydrogen bonds contribute main interactions between OsOTUB1-cat and proximal Ub of Met1-diUb (Fig. 3B, S7). As the interaction surface spans a large area (approximately 26Å×14Å), we separated it into three sections, named top-, middle- and bottom-handlers, representing interaction surfaces that happening in α 1, β 1 and loop (β 2- α 1), and loop (β 4- β 5) of proximal Ub, respectively (Fig. S7). Except for side chain of R116 in OsOTUB1-cat interacts with side chain of E64^{prox} in the bottom handler, all the interactions happen in the top- and middle-handlers (Fig. 3B). In the top- and middle-handlers, side chains of E90, R114, R314, D319, T115 and Y273 in OsOTUB1-cat interact with side chains of E16^{prox}, E18^{prox}, K33^{prox} and Q2^{prox} via network of hydrogen bonds, directly or indirectly, including a) Oxygen of the amide group of Q2^{prox} forms a hydrogen bond with hydroxyl of T115 through a water molecule (w1), b) carboxyl of E16^{prox} forms a hydrogen bond directly with R314, and indirectly with R114 and E90 via a water molecule (w2), c) carboxyl of E18^{prox} forms a hydrogen bond with phenolic hydroxyl of Y273 via a water molecule (w3), d) amino of K33^{prox} interacts with E90. Please refer to the third paragraph in the part of “3. Binding of proximal Ub facilitates the Met1 type preference of OsOTUB1-cat”.

Furthermore, in top- and middle-handlers, intramolecular interactions also happen, including a) carboxyl of E16^{prox} simultaneously interacts with amino of K33^{prox} and Q2^{prox} by hydrogen bonds, forming a hydrogen bonding line in the proximal Ub, arranging in the order of

basic-acidic-basic, b) guanidyl of R314 interacts with carboxyl of E90 and D319 simultaneously by hydrogen bonds, forming a hydrogen bonding line in the S1' pocket, arranging in the order of acidic-basic-acidic. According to the residue arrangement, it may be reasonable that the compatibility between the two lines is the key for OsOTUB1-cat's efficient binding with Met1-diUb. Mutation of any residue that changes the charge property would significantly affect OsOTUB1-cat's M1 activity. Please refer to the fourth paragraph in the part of "3. Binding of proximal Ub facilitates the Met1 type preference of OsOTUB1-cat".

To verify these speculations, Q2^{prox}, E16^{prox}, E18^{prox}, K33^{prox} and E64^{prox} were individually mutated and the hydrolytic activity of OsOTUB1-cat toward these mutants was measured and compared (Fig. 3C). The results showed that, mutations upon Q2^{prox}, E16^{prox}, E18^{prox}, K33^{prox} and E64^{prox} will affect the hydrolytic activity of OsOTUB1-cat against them to varying degrees, and the digestive preference order of OsOTUB1-cat is Met1-diUb WT \approx Met1-diUb E18A^{prox}>Met1-diUb E16Q^{prox} \approx Met1-diUb E16A^{prox}>Met1-diUb E64A^{prox}>Met1-diUb Q2A^{prox}>Met1-diUb K33A^{prox}>Met1-diUb Q2E^{prox}>Met1-diUb K33E^{prox}. These results support the speculation that binding of proximal Ub by S1' pocket of OsOTUB1-cat is important for its M1 activity and compatibility between the hydrogen bonding lines in OsOTUB1-cat and proximal Ub is vital for OsOTUB1-cat's efficient binding and cleavage of Met1-diUb (etc. Q2E^{prox} and K33E^{prox}). Please refer to the fifth paragraph in the part of "3. Binding of proximal Ub facilitates the Met1 type preference of OsOTUB1-cat".

Therefore, Q2^{prox} is important for Met1-diUb cleavage, but theoretically, maybe not for hydrolysis of Ub-AMC, since Ub-AMC was previously reported to be bound by S1 pocket of hOTUB1 (Wiener et al., 2013).

Is there not a binding pocket for Met1 in the complex structure ?? A better figure of the active site environment cleavage residues would help to understand the process (including the crosslink).

Respond:

Thank you for your question!

We have supplemented the analysis of the surrounding environment of M1^{prox} in the detailed structure of S1' binding to proximal ubiquitin. It showed that (Figure 3B), M1^{prox} barely interacts with OsOTUB1, except for a hydrogen bond formed between its backbone amino and the carboxyl of G316. Side chain of M1^{prox} otherwise projects to the interior of proximal Ub, similar with M1^{prox} in the structure of OTULIN(C129A)-Met1-diUb (Figure S7). Please refer to the last two sentences of the third paragraph in the part of "3. Binding of proximal Ub facilitates the Met1 type preference of OsOTUB1-cat".

What is the structural homology between OTULIN and OsOTUB1 ?. Since they belong to the same OTU subfamily and both can interact with the proximal ubiquitin of a linear Met1-diUb, it would be interesting to show or to discuss whether they share a similar binding mode to the

proximal Ub.

Respond:

Thank you for your question! We compared the structure of OsOTUB1-cat~M1-diUb-DHA with OTULIN(C129A)-Met1-diUb.

The binding of OsOTUB1 to Met1-diUb is accompanied by the distortion of Met1-diUb (Fig. S6C), which is similar to that in OTULIN(C129A)-Met1-diUb. Surprisingly, compared with OTULIN(C129A)-Met1-diUb, proximal Ub in OsOTUB1-cat~M1-diUb-DHA adopts almost the same conformation (Fig. S6A), while distal ubiquitin exhibits a spatial deflection with approximately 80°, which is probably caused by different orientation of the α helix responsible for binding of distal ubiquitin (Fig. S6B). Please refer to the second paragraph in the part of “3. Binding of proximal Ub facilitates the Met1 type preference of OsOTUB1-cat”.

Besides, we structurally compared the binding of proximal Ub by OsOTUB1 with OTULIN (Fig. S7). OsOTUB1 and OTULIN adopt nearly identical conformation to accommodate proximal Ub, with RMSD 1.76 Å. They both interact with proximal Ub via top-, middle- and bottom handlers. However, multiple of obvious differences between their interactions to proximal Ub can be found in detail, both in the configuration and interaction number. In the top-handler, OsOTUB1 employs E90 to interact with K33^{prox}, while OTULIN extensively interacts with D32^{prox}, K33^{prox} and E34^{prox} by E95, R97, G98, T100 and K102, directly or indirectly. In the middle-handler, OsOTUB1 makes use of R314 to directly bind to E16^{prox}, while OTULIN uses the catalytic residue N341 to directly, and other 5 residues (Y91, E335, I333, D336 and R338) to indirectly bind with E16^{prox}. OsOTUB1 binds to proximal Ub through totally 8 hydrogen bonds, while OTULIN employs 16 hydrogen bonds to interact with proximal Ub. Taken together, although OsOTUB1 and OTULIN share similar mode in accommodating proximal Ub, they employ residues with different orientation and quantity to accomplish these interactions. Please refer to the last paragraph in the part of “3. Binding of proximal Ub facilitates the Met1 type preference of OsOTUB1-cat”.

Lane 283. Figures should be better introduced in the main text. In ex., fig3D and 3E in the main text (they are confusing).

Respond:

Thank you for your suggestion! We have adjusted Figure 3D and 3E to Figure 4C and S9A, respectively, and well introduced them in the main text. Please refer to the first and fourth paragraph in the part of “4. Two motifs in S1' pocket determines Met1 type preference of OsOTUB1”.

Lane 286. In figure 3C the intensity band that increases at longer times with the double mutant is very weak and difficult to observe. The conclusion that they confer diMIUb activity to hOTUB1 is

highly questionable. I guess that other (not obvious) structural determinants are present in plant *OsOTUB1*.

Respond:

Thank you for your question!

In response to this question, we re-expressed and purified the protein, and repeated the Met1-diUb hydrolysis assay (Fig. 4D). The results showed that wildtype hOTUB1 did not have the ability to hydrolyze Met1-diUb at all (no product is detected after 24h); similarly, the single mutation of C-handle motif (P87G) and of N-handle motif (SGS) could not confer hOTUB1 the ability to hydrolyze Met1-diUb (no hydrolysis product is detected after 24h). But when C-handle motif and N-handle motif were mutated at the same time (P87G-SGS), hOTUB1 could hydrolyze ~20% of Met1-diUb in 30 minutes and ~50% of Met1-diUb in 2 hours. These results proved that changes in C-handle motif and N-handle motif can indeed endow hOTUB1 with the ability to hydrolyze Met1-diUb. Please refer to the first paragraph in the part of “4. Two motifs in S1’ pocket determines Met1 type preference of OsOTUB1”.

Another issue that could be clarified is about the mechanism of OsOTUB1 to cleave Ub-K48 chains, which seems also as active as for diUbM1 (Fig 1E). Is there any interaction of the proximal Ub with OsOTUB1 in diUbK48 as occurs in the cleavage of Met1-linked chains? Perhaps activity assays with the OsOTUB1 proximal Ub mutants could be conducted with K48-linked chain substrates.

Respond:

Thank you for your question!

We have been exploring the molecular mechanism of OsOTUB1 hydrolyzing Lys48 chains by structural biology. However, we couldn't crystallize OsOTUB1 in complex with Lys48 chains albeit we've tried several strategies. Fortunately, Juang *et al.* has been reported the crystal structure of Ub~UbcH5b^{C85S}~hOTUB1^{Δ1-24} complexed with free Ub (Ub^{distal}~UbcH5b^{C85S}~hOTUB1^{Δ1-24}-Ub^{prox}, Juang et al. 2012), which includes two Ubs adopting Lys48-diUb configuration and may provide clues for interaction between Lys48-diUb and hOTUB1. Based on the structure alignment between OsOTUB1-cat~M1-diUb-DHA and Ub^{distal}~UbcH5b^{C85S}~hOTUB1^{Δ1-24}-Ub^{prox} (Figure S10A), we found that OsOTUB1-cat~Ub^{distal} is similar with that of Ub^{distal}~hOTUB1^{Δ1-24} (RMSD 1.53Å) (Figure S10C), proving that OsOTUB1 shares almost the same S1 with hOTUB1. However, proximal Ub in OsOTUB1-cat~M1-diUb-DHA adopts distinct conformation with that in Ub^{distal}~UbcH5b^{C85S}~hOTUB1^{Δ1-24}-Ub^{prox} (Figure S10B). This is mainly due to the distinguish different localization between M1 and K48 in Ub. And also for this reason, the residues in proximal Ub that participate in the formation of Met1-type chains do not contribute interactions in the formation of Lys48-type (Figure S10B). Therefore, we didn't conduct the similar activity assays on Lys48 chains as that on Met1 chains. Please refer to the third paragraph in the part of

“Discussion”.

There are several publication on the interaction of OTUB1 with the E2 conjugation enzyme. In particular there seems to be the formation of a N-terminal α -helix that interacts with the proximal Ub and the E2 enzyme in OTUB1 (Wiener et al, 2012, Nature). Additionally E2 binding reciprocally regulates the conjugation-deconjugation cycles. It would be interesting in the discussion to add some information on this point when compare human and OsOTUB1. Is such helix present in OsOTUB1?. Does it bind E2 enzymes ?.

Respond:

Very good question! We probed the function of the N-terminal extension in OsOTUB1.

In hOTUB1, the N-terminal peptide (A25-N45) forms an α helix which binds to and stabilizes the conformation of the proximal Ub in Lys48-diUb, thereby promoting the cleavage by hOTUB1 (Juang et al., 2013; Wiener et al., 2013), which is also confirmed by our data, both in the presence of hUBCH5B and OsUBCH5B (Fig. S2C). Inspired by this phenomenon, we asked whether activity of OsOTUB1 toward Lys48-diUb can be promoted by OsUBCH5B or hUBCH5B. Therefore, activity of OsOTUB1 against Lys48-diUb was monitored, in the presence of OsUBCH5B or hUBCH5B (25 μ M). The results showed that (Fig. S2C), the activity of OsOTUB1 against Lys48-diUb can be promoted by both hUBCH5B and OsUBCH5B. However, the sequence comparison shows that (Fig. S3), the N-terminal sequence of OsOTUB1 is significantly different from that of hOTUB1. In OsOTUB1, the N-terminal sequence is rich in residues that are prone to form random coils, such as Gly, Ser and Pro, suggesting that this region would not form an α helix to bind and stabilize the proximal Ub in Lys48-diUb. To reveal whether the activity promotion by E2 is dependent on the N-terminus sequence of OsOTUB1, activity of OsOTUB1-cat (OsOTUB1 that deleting N terminus) against Lys48-diUb was monitored, in the presence of OsUBCH5B or hUBCH5B. The results showed that (Fig. S2D), the K48 activity of OsOTUB1-cat can also be promoted by both hUBCH5B and OsUBCH5B. Therefore, stimulation of OsOTUB1 against Lys48-diUb by both hUBCH5B and OsUBCH5B is not (or not completely) dependent on the N-terminus sequence. The mechanism that OsUBCH5B and hUBCH5B enhance the activity of OsOTUB1 needs to be further revealed. Please refer to the last paragraph in the part of “1. The discovery of plant DUBs that hydrolyze Met1 chains using an efficiently prepared Met1-diUb suicide probe”.

Reviewer #3 (Remarks to the Author):

The authors design a new strategy to synthesise a diUb probe that reacts with deubiquitinating enzymes that cleave Met1 (linear) ubiquitin conjugates. They then take this probe and look for Met1-selective DUBs in plants. The presence of Met1-specific DUBs in plants has not been carefully investigated and it is interesting that the authors identify candidate Met1-selective DUBs, notably OsOTUB1.

Given that the Met1 ubiquitin signal in mammals is predominantly associated with inflammation and the fact that plants have a more diverse and complex ubiquitination system than mammals, it is highly relevant that a Met1-selective DUB has been identified, which will lead the way to future studies to investigate the role of the Met1 ubiquitin signal in plants.

The authors then go on to mechanistically characterise the role of OsOTUB1 in how it recognizes and cleaves Met1 ubiquitin and present three crystal structures detailing how OsOTUB1 functions.

Overall, this reviewer finds this an interesting study and the identification of a plant Met1-selective DUB opens up the field to investigating the Met1 signal in plants. However, the mechanistic insights are poorly translated from the structural analyses. Notably, no experiments decouple the effects of binding mutations (Kd) from catalytic activities (kcat). Descriptive information between the mode of action of OTULIN and OsOTUB1 is made but no solid conclusions can be made to compare the mechanism of OsOTUB1.

Overall, this reviewer found the presentation and execution of some of the biochemical experiments as sloppy and not worthy of publication in their current form. It was not clear on how many occasions these experiments had been performed and it was not clear, in some DUB assays, whether the ubiquitin substrates were actually being cleaved or whether contaminating mono ubiquitin was already present in the input.

Respond:

Thank you very much for your comments.

Firstly, we have added more replicates to prove the reproducibility of the experiments. For almost all the experiments, included with Figure 1D, 1E, 1F, 2E, 2H, 2I, 3A, 3C, 4C, 4D, 4F, 4G, 6B, S2A, S2B, S2C, S2D, S9A, S9B and S9C, three replicates were performed. Please find the replicates in the “Source Data”. All the replicates showed the consistent results. For the quantitative assay, “based on three replicates” was stated in the legends for Figure 1D, 3A and S2B, and “based on two replicates” was stated in the legends for Figure 2F and 2G.

Secondly, to verify that the appeared bands with smaller molecular weight was produced by DUB treatment of diUb, almost all the digestion experiments were re-performed and the zero-time results were added, included with Figure 1F, 2E, 3C, 4D, 4G, 6B, S2C, S2D, S9A, S9B and S9C. Please find the SDS-PAGE results in both the main text and the supporting information.

Without a separation of binding versus catalytic mutants this reviewer does not feel that the authors can make the statement that OsOTUB1 is a bi-modal activated deubiquitinase and further careful structural interpretation and biochemical experiments are required to justify these assertions.

Respond:

Thank you very much for your comments. The followings are brief responses.

According to your comments, we initially monitored the K_m and K_{cat} of OsOTUB1-cat in the presence of Q2^{prox} variants of Met1-diUb by HPLC. The results showed that (Fig. 2F and 2G), mutations on Q2^{prox} will affect the affinity of OsOTUB1-cat with substrate, thereby altering the catalysis ability of OsOTUB1-cat, but not hampering Q2^{prox}-mediated activation of OsOTUB1-cat.

Q2's real role is to participate in the interaction between proximal ubiquitin and S1' pocket, which was confirmed by further analysis of the detailed structure of S1' pocket bound to the proximal ubiquitin, as well as the digestion assay against mutants in the proximal ubiquitin (Fig. 3B, 3C and S7).

For more detail information, please refer to the following specific responses.

Below are some more specific and minor comments relating to the manuscript.

In the structure of apo OsOTUB1 the catalytic His (Hiscat) is in a non-catalytic configuration whereas in the structure of OsOTUB1 bound to Met1 diUb DHA, Hiscat is in a catalytically competent orientation, as would be expected since this structure represents a trapped intermediate of the reaction. The authors focus their attention on Gln2 of the proximal ubiquitin moiety that occupies the position where the imidazole ring from Hiscat resides in the inactive apo configuration. A series of mutations are made on Gln2 to assess the ability of OsOTUB1 to hydrolyse Met1 ubiquitin variants. Figure 2E shows a DUB assay (with no time zero control) showing different lengths of incubation of the Met1 ubiquitin variants with OsOTUB1. From this experiment it is impossible to conclude whether the lack of Met1 ubiquitin cleavage is due to a deficiency in binding of Met1 ubiquitin to OsOTUB1 or whether Gln2 directly contributes towards the catalytic site affecting the rate of hydrolysis of Met ubiquitin. At the very least, kinetic information on the efficiency of OsOTUB1 is required to show that the effect is not simply related to K_d but rather k_{cat} .

Confusingly, the authors state that in a previous study they performed on OTULIN (see line 211), that the proximal Glu16 disrupts the inhibitory effect of Asp336 and aligns the catalytic histidine, providing a mechanism of ubiquitin-assisted catalysis. It should be noted that the Keusekotten 2013 paper separated the binding function of Glu16 and the catalytic role of Glu16. Interestingly, Gln2 was also in proximity to the active site but had no effect on catalysis. What are the effects of other residues from the proximal ubiquitin on activity of OsOTUB1? Notably, Glu16 and Lys33 as shown in the figure?

The authors then tested the role of distal Ub binding on the activity of OsOTUB1 through determining a structure of Ub-PA reacted with OsOTUB1 and testing ability to hydrolyse Ub-AMC. OsOTUB1 is able to hydrolyse Ub-AMC, though at a much-reduced rate than UCHL3, suggesting that binding of the distal Ub is sufficient to induce rearrangement of the catalytic triad in a more

catalytic configuration. This is not the case for OTULIN as pointed out by the authors and shown in Figure 2G and S Figure 3B. However, the authors then go on to solve the structure of OsOTUB1 bound to Ub-PA. This reviewer could not identify an overlay between the OsOTUB1 Met1 diUB DHA structure with OsOTUB1 Ub-PA structures. However, from Figure 2H it is clear that the Hiscat is in a catalytically active configuration (which would be expected). Therefore, since the proximal ubiquitin is not present in the OsOTUB1 Ub-PA structure and yet it is already active, wouldn't this suggest that the effects of proximal Gln2 on activating OsOTUB1 are not important and hence the entire argument regarding bi-modal activation is incorrect? How are the structures of the two states related to one another? More detailed structural analysis needs to be provided. Again, the inclusion of more quantitative biochemistry is required to deduce the importance of Gln2 for OsOTUB1 activity. Importantly, the authors need to distinguish between reactivity of OsOTUB1 that is mediated through binding of the distal ubiquitin moiety, and activity of OsOTUB1 towards di ubiquitin substrates, which in theory could be dependent on proximal ubiquitin Gln2. Only through k_{cat}/K_M analysis of Ub-AMC cleavage and k_{cat}/K_M measurements on the hydrolysis of Met1 and Lys48-linked di ubiquitin can these assertions be made.

Respond:

Thank you very much for your questions! These questions helped us to timely and effectively revise the important conclusions in the manuscript!

The proximal ubiquitin activation mechanism proposed in the manuscript lacks direct evidence, which is mainly due to the failure to provide kinetic data that can dissect the binding function and the catalytic role of Q2^{prox}. For this reason, we used HPLC quantitative method to accurately calculate the kinetic data of OsOTUB1-cat in the presence of different Met1-diUb mutants (Fig. 2F, G). The kinetics results showed that (Fig. 2F, G), compared with OsOTUB1-cat in the presence of Met1-diUb WT (0.31 s^{-1}), the turnover number (k_{cat}) of OsOTUB1-cat show 1.07-fold (0.29 s^{-1}), 2.21-fold (0.14 s^{-1}), 7.21-fold (0.043 s^{-1}) and 413.3-fold reduction (0.0075 s^{-1}) in the presence of Met1-diUb Q2H^{prox}, Met1-diUb Q2N^{prox}, Met1-diUb Q2A^{prox} or Met1-diUb Q2E^{prox}, respectively. This is consistent with the activity of OsOTUB1-cat against different mutants (Fig. 2E), demonstrating that mutation upon Q2^{prox} significantly inhibits the M1 activity of OsOTUB1-cat by reducing its k_{cat} . However, affinity (K_M) of OsOTUB1-cat towards Met1-diUb mutants were not keeping almost unchanged or elevated as seen in OTULIN which was previously demonstrated to be activated by proximal Ub (Keusekotten et al., 2013), but significantly reduced: compared to OsOTUB1-cat with Met1-diUb WT ($25.65 \text{ }\mu\text{M}$), the affinity (K_M) of OsOTUB1-cat showed 1.51-fold ($38.76 \text{ }\mu\text{M}$), 1.69-fold ($43.32 \text{ }\mu\text{M}$), 3.95-fold ($101.2 \text{ }\mu\text{M}$) and 5.11-fold ($131.1 \text{ }\mu\text{M}$) reduction in the presences of Met1-diUb Q2H^{prox}, Met1-diUb Q2N^{prox}, Met1-diUb Q2A^{prox} or Met1-diUb Q2E^{prox}, respectively. These results demonstrated that mutations on Q2^{prox} will affect the affinity of OsOTUB1-cat with substrate, thereby altering the catalysis ability of OsOTUB1-cat, but not hampering Q2^{prox}-mediated activation of OsOTUB1-cat. **Please refer to the fourth paragraph in the part of “2. Activation of OsOTUB1-cat is induced by distal Ub, but not proximal Ub”.**

In line with the conclusion that OsOTUB1-cat is induced to be active by distal Ub, but not proximal Ub, we find that residues around the catalytic center of OsOTUB1-cat~Ub-PA are already at the almost exact positions with those in the structure of

OsOTUB1-cat~Met1-diUb-DHA (Fig. S5). Please refer to the last sentence of the fifth paragraph in the part of “2. Activation of OsOTUB1-cat is induced by distal Ub, but not proximal Ub”.

Q2's real role is to participate in the interaction between proximal ubiquitin and S1' pocket, which was confirmed in further analysis of the detailed structure of the S1' pocket bound to the proximal ubiquitin (Figure 3B). In the S1' pocket, hydrogen bonds contribute main interactions between OsOTUB1-cat and proximal Ub of Met1-diUb (Fig. 3B, S7). As the interaction surface spans a large area (approximately 26Å×14Å), we separated it into three sections, named top-, middle- and bottom-handlers, representing interaction surfaces that happening in $\alpha 1$, $\beta 1$ and loop ($\beta 2$ - $\alpha 1$), and loop ($\beta 4$ - $\beta 5$) of proximal Ub, respectively (Fig. S7). Except for side chain of R116 in OsOTUB1-cat interacts with side chain of E64^{prox} in the bottom handler, all the interactions happen in the top- and middle-handlers (Fig. 3B). In the top- and middle-handlers, side chains of E90, R114, R314, D319, T115 and Y273 in OsOTUB1-cat interact with side chains of E16^{prox}, E18^{prox}, K33^{prox} and Q2^{prox} via network of hydrogen bonds, directly or indirectly, including a) Oxygen of the amide group of Q2^{prox} forms a hydrogen bond with hydroxyl of T115 through a water molecule (w1), b) carboxyl of E16^{prox} forms a hydrogen bond directly with R314, and indirectly with R114 and E90 via a water molecule (w2), c) carboxyl of E18^{prox} forms a hydrogen bond with phenolic hydroxyl of Y273 via a water molecule (w3), d) amino of K33^{prox} interacts with E90. Please refer to the third paragraph in the part of “3. Binding of proximal Ub facilitates the Met1 type preference of OsOTUB1-cat”.

Furthermore, in top- and middle-handlers, intramolecular interactions also happen, including a) carboxyl of E16^{prox} simultaneously interacts with amino of K33^{prox} and Q2^{prox} by hydrogen bonds, forming a hydrogen bonding line in the proximal Ub, arranging in the order of basic-acidic-basic, b) guanidyl of R314 interacts with carboxyl of E90 and D319 simultaneously by hydrogen bonds, forming a hydrogen bonding line in the S1' pocket, arranging in the order of acidic-basic-acidic. According to the residue arrangement, it may be reasonable that the compatibility between the two lines is the key for OsOTUB1-cat's efficient binding with Met1-diUb. Mutation of any residue that changes the charge property would significantly affect OsOTUB1-cat's M1 activity. Please refer to the fourth paragraph in the part of “3. Binding of proximal Ub facilitates the Met1 type preference of OsOTUB1-cat”.

To verify these speculations, Q2^{prox}, E16^{prox}, E18^{prox}, K33^{prox} and E64^{prox} were individually mutated and the hydrolytic activity of OsOTUB1-cat toward these mutants was measured and compared (Fig. 3C). The results showed that, mutations upon Q2^{prox}, E16^{prox}, E18^{prox}, K33^{prox} and E64^{prox} will affect the hydrolytic activity of OsOTUB1-cat against them to varying degrees, and the digestive preference order of OsOTUB1-cat is Met1-diUb WT \approx Met1-diUb E18A^{prox}>Met1-diUb E16Q^{prox} \approx Met1-diUb E16A^{prox}>Met1-diUb E64A^{prox}>Met1-diUb Q2A^{prox}>Met1-diUb K33A^{prox}>Met1-diUb Q2E^{prox}>Met1-diUb K33E^{prox}. These results support the speculation that binding of proximal Ub by S1' pocket of OsOTUB1-cat is important for its M1 activity and compatibility between the hydrogen bonding lines in OsOTUB1-cat and proximal Ub is vital for OsOTUB1-cat's efficient binding and cleavage of Met1-diUb (etc. Q2E^{prox} and K33E^{prox}). Please refer to the fifth paragraph in the part of “3. Binding of proximal Ub facilitates the Met1 type preference of OsOTUB1-cat”.

The second part of the manuscript addresses the difference in specificity between hOTUB1 and OsOTUB1. hOTUB1 is specific for hydrolysing Lys48-linked ubiquitin chains, whereas OsOTUB1 has selectivity for Met1 and Lys48-linked chains. Mutations that were introduced in hOTUB1 to facilitate Met1 ubiquitin cleavage had very little effect on hydrolysis of Met1 ubiquitin (Figure 3C), so this reviewer is somewhat sceptical that these mutations can be directly translated into hOTUB1. However, this reviewer acknowledges that it is difficult to obtain gain-of-function mutations to achieve linkage selectivity. Importantly, the controls assessing the effect of these hOTUB1 variants to cleave Lys48-linked ubiquitin are lacking. The phylogenetic analysis of different OTUB1 families is interesting but Figure 4 is confusing, no description of the sub-panels of Figure 4A is given in the figure legend nor the text. In the first part of Figure 4B, it was impossible to assess the ability of the DUBs to hydrolyse Met1 ubiquitin as no zero-time point is shown for each DUB and the levels of mono ubiquitin vary significantly but do not increase between 5 and 15 minutes, suggesting there is contaminating mono ubiquitin in the di ubiquitin or that the enzyme has ceased functioning during the course of the assay.

Respond:

Thank you for your question! These questions are critical. In response to these questions, we have supplemented the following experiments.

1. We re-expressed and purified the protein, and repeated the Met1-diUb hydrolysis assay (Fig. 4D). The results showed that wildtype hOTUB1 did not have the ability to hydrolyze Met1-diUb at all (no product is detected after 24h); similarly, the single mutation of C-handle motif (P87G) and of N-handle motif (SGS) could not confer hOTUB1 the ability to hydrolyze Met1-diUb (no hydrolysis product is detected after 24h). But when C-handle motif and N-handle motif were mutated at the same time (P87G-SGS), hOTUB1 could hydrolyze ~20% of Met1-diUb in 30 minutes and ~50% of Met1-diUb in 2 hours. These results proved that changes in C-handle motif and N-handle motif can indeed endow hOTUB1 with the ability to hydrolyze Met1-diUb. Please refer to the first paragraph in the part of “4. Two motifs in S1’ pocket determines Met1 type preference of OsOTUB1”.
2. We don’t agree that the effect of these hOTUB1 variants on cleavage of Lys48 chains should be considered as controls, because hOTUB1 may not interact with Met1 and Lys48 chains in absolutely different way, and these mutations may simultaneously influence hOTUB1’s interaction with both Met1 and Lys48 chains. This assumption was confirmed by our experiments. The results showed that (Fig. S9), for the C-handle motif, displacement of P with G promoted the hydrolysis activity of hOTUB1-cat against Lys48-diUb; while for the N-handle motif, mutation from EDD to SGS can weaken that of hOTUB1-cat.

Meanwhile, the effect of OsOTUB1 variants on cleavage of Lys48-linked ubiquitin was investigated. The results showed that (Fig. S9), for the C-handle motif, displacement of P with G promoted the hydrolysis activity against Lys48-diUb, which is similar with hOTUB1; while for the N-handle motif, mutation from EDD to SGS can strengthen that of OsOTUB1-cat, which is opposite to hOTUB1.

These results indicate that hOTUB1-cat adopts different, but not absolutely different, way from OsOTUB1-cat to bind to and digest Lys48-diUb. Yet, the exact molecular mechanism needs further investigations. Please refer to the fifth paragraph in the part of “4. Two motifs in S1’ pocket determines Met1 type preference of OsOTUB1”.

3. Figure 4A has been replaced with Figure 6A and the brief description of the sub-panels is given in the text. Please refer to the first paragraph in the part of “4. The influence of the N-handle and C-handle motifs on M1 activity is observed in the OTUB subfamily from other species”.
4. We have re-added the results at zero-time point. And in order to make the hydrolysis activity more obvious, we have uniformly extended the hydrolysis time to 60 min (Fig. 6B). After repeated verifications, we found that almost all the sequences comprising the patterns SGS, NGS, K/R-GS, G/A-GS and XGN hydrolyzed Met1-diUb, while in contrast, almost all the sequences incorporating the XAD pattern (against XGY pattern) hydrolyzed undetectable or negligible Met1-diUb within 60 minutes. Therefore, these results demonstrated that sequences comprising XGY pattern can hydrolyze Met1-diUb, but those disobeying XGY pattern cannot. However, there are also some exceptions here, 1) we couldn’t detect any activity of sequence from ZOSMR (belonging to SGS), albeit we have rechecked the coding sequence, re-purified the protein for several times and tested the activity even when protein concentration was elevated to 100 folds (data not shown here); 2) sequence from OPHP1 (belonging to N/Q-AD) showed effective catalytic activity (cleaving 100% Met1-diUb in 30 min). Therefore, it is conclusive that the XGY pattern principle may be applicable to determine the hydrolysis activity against Met1 chains across the OTUB subfamily members. But it is not the whole story. There may exist other factors that need to be explored. Please refer to the second paragraph in the part of “5. The influence of the N-handle and C-handle motifs on M1 activity is observed in the OTUB subfamily from other species”.

From the data statistics provided in Table S1, it seems that the effective resolution extends beyond the reported resolution for the apo and Met1 diUb DHA structures. The authors may wish to further investigate whether their electron density maps can be improved. For the data scaling it is good practice to include CCI/2 values as a metric for data quality and effective resolution. In addition, Ramachandran statistics are not included in Table S1. However, from one PDB validation report submitted with the manuscript, there are a significant number of Ramachandran outliers (4% in one chain). Representative electron density for the outliers should be included in the supplementary figures. However, given that all chains are the same macromolecule and the high degree of variation between all the molecules, this suggests to this reviewer that this structure is not fully refined. The other validation reports had improved model quality statistics.

Respond:

Thank you for your questions!

- ① According to your suggestion, the percentages of Ramachandran outliers are included in Table

S1. The percentages of Ramachandran outliers for 6K9N, 6K9P and 6KBE are 0.3%, 0% and 0.5%, respectively. Please refer to Table S1 in the “Supporting Information”, together with the validation reports for 6K9N, 6K9P and 6KBE.

- ② We have refined the structure. The disallowed region of Ramachandran plot is now 1%, and the outlier percentage of C-chain in 6K9N, which was 4%, is now 0% (please refer to the latest PDB validation report for 6K9N). These adjustments do not affect all conclusions in the manuscript.

Minor comments

Abstract mentions that no Met1-selective DUBs have been identified in plants and that the authors identify OTUB1. However, it has been previously reported in the literature that OTUB1 from Arabidopsis can hydrolyse Met1 ubiquitin chains see reference (Radjacomare R. et al., Front Plant Sci. 2014).

Respond:

Thank you for your suggestions! We did not find this paper when we were preparing the manuscript.

We have adjusted the corresponding content in the part of “Abstract” to “However, few plant-derived deubiquitinases (DUBs) against Met1 ubiquitin chains has been reported, and the selection mechanism for linkage-type of ubiquitin chains remains elusive, which greatly limits our understanding of the functions of plant-derived DUBs.” please refer to the part of “Abstract”.

Additionally, we have also supplemented related texts and citations in the part of “Introduction”, that is, “In plant, physiological roles of Met1 chain and directly correlated DUBs have rarely been reported (Radjacomare et al., 2014; Zang et al., 2020; Majumdar et al., 2020). With the exception of *Arabidopsis*, there are no DUBs digesting Met1 chains are reported in other plants. Importantly, how plant DUBs recognize selectively and hydrolyze Met1 linkage is still enigmatic.” Please refer to the last three sentences of the second paragraph in the part of “Introduction”.

Naming the versions of the probe from Met1-diUB-1 to Met1-diUb-2 is a little confusing. Could the authors please refer to it as Met1-diUb DHA or something that doesn't read as Ub2?

Respond:

Thank you for your suggestion!

“Met1-diUb-2” cannot be simply changed to “Met1-diUb DHA”, as the probe used to obtain the cross-linked complex has been named “M1-diUb-DHA”. According to your suggestion,

“Met1-diUb-1” and “Met1-diUb-2” have been changed to “Avi-M1-diUb (G76C^{distal})” and “Biotin-Avi-M1-diUb-DHA”, respectively. Please refer to the updated manuscript.

A lot of the figures omit important information to assess the activity of OsOTUB1, such as concentration of OsOTUB1 used and in some cases length of incubation time (e.g. Figure 1E has no incubation time indicated nor amount of enzyme added).

Respond:

Thank you for your suggestions!

We have added the concentration, reaction temperature and reaction time of DUB and Ub chains in the figure, please refer to the Figure 1E.

In addition, we also uniformly stated the concentration and reaction temperature of DUB and Ub chains in all enzymatic digestion experiments in the manuscript, that is, “In addition, in order to ensure the uniformity, all the following enzymatic digestion experiments were carried out under 37 °C and the concentration of DUB and Ub chains were 2 μM and 60 μM, respectively, unless otherwise noted.” Please refer to the last sentence of the fifth paragraph in the part of “1. The discovery of plant DUBs that hydrolyze Met1 chains using an efficiently prepared Met1-diUb suicide probe”.

This reviewer is not a plant expert but wonders what the differences between looking at the levels of Met1 Ub in seedlings is compared to looking at other regions of plants? It would be helpful for a more general audience to explain the rationale for Figure 1D.

Respond:

Thank you for your suggestions!

I am not very clear about what you mean. I understand that you want to know the changes of Met1 chains in rice at different growth periods. To this end, we investigated the effect of knocking out and overexpressing OsOTUB1 on the abundance of Met1 chains and Lys48 chains at the young panicle differentiation stage. The results showed that (Figure S2B), the effects of OsOTUB1 on Met1 and Lys48 chains became insignificant at the young panicle differentiation stage, as there were no obvious differences among WT, KO and OE, for both Met1 and Lys48 chains (Fig. S2B). The underlying mechanism needs to be further resolved. Please refer to the last two sentences of the fourth paragraph in the part of “1. The discovery of plant DUBs that hydrolyze Met1 chains using an efficiently prepared Met1-diUb suicide probe”.

Line 136 “FL-OsOTUB1 was not stable (spots on gel)”. To this reviewer, the band corresponding

to FL OsOTUB1 looks fine and there are no additional bands on the gel to indicate degradation. The authors should re write this sentence a little more carefully.

Respond:

Thank you for your suggestions!

In our follow-up experiments, we did find that FL-OsOTUB1 is prone to degradation (see Figure 1F and S2C). But for the sake of safety, we changed the wording to “Notably, FL-OsOTUB1 appeared smear on the gel (Fig. 1F, S2C), suggesting that it may be prone to degrade into variants with similar molecular weight, thus challenging the following quantification and enzymatic assay – perhaps due to the instability of the flexible, glycine-rich region on the N-terminus of full-length protein (Fig. S3).” **Please refer to the fifth paragraph in the part of “1. The discovery of plant DUBs that hydrolyze Met1 chains using an efficiently prepared Met1-diUb suicide probe”.**

For the structural overlays it is hard to deduce which structures are shown, i.e., which PDB accession numbers have been used for the overlay? Often the overall structure is showing only one molecule, but the zoomed-in regions display stick representation of both structures (e.g., see Figure 3). It was hard for the reviewer to work out which residues were from which structure and how well the overall structures superimpose.

Respond:

Thank you for your suggestions!

We have revised and supplemented all the figures in the manuscript that involve structural comparison. In these figures, we added the code of PDB used for structural comparison, the whole-structure comparison result, together with the corresponding RMSD. **Please refer to Figure 2B, 2C, 2J, 4A, S5, S7 and S10.**

hOTUB1 is activated through binding to E2 and inhibited through mono ubiquitin binding (see Wiener R, et al., NSMB 2013 and subsequent publications from the Wolberger group). Does a similar mechanism operate for OsOTUB1 and are these residues conserved amongst plant OTUB1?

Respond:

Very good question! We probed the function of the N-terminal extension in OsOTUB1.

In hOTUB1, the N-terminal peptide (A25-N45) forms an α helix which binds to and stabilizes the conformation of the proximal Ub in Lys48-diUb, thereby promoting the cleavage by hOTUB1 (Juang et al., 2013; Wiener et al., 2013), which is also confirmed by our data, both in the

presence of hUBCH5B and OsUBCH5B (Fig. S2C). Inspired by this phenomenon, we asked whether activity of OsOTUB1 toward Lys48-diUb can be promoted by OsUBCH5B or hUBCH5B. Therefore, activity of OsOTUB1 against Lys48-diUb was monitored, in the presence of OsUBCH5B or hUBCH5B (25 μ M). The results showed that (Fig. S2C), the activity of OsOTUB1 against Lys48-diUb can be promoted by both hUBCH5B and OsUBCH5B. However, the sequence comparison shows that (Fig. S3), the N-terminal sequence of OsOTUB1 is significantly different from that of hOTUB1. In OsOTUB1, the N-terminal sequence is rich in residues that are prone to form random coils, such as Gly, Ser and Pro, suggesting that this region would not form an α helix to bind and stabilize the proximal Ub in Lys48-diUb. To reveal whether the activity promotion by E2 is dependent on the N-terminus sequence of OsOTUB1, activity of OsOTUB1-cat (OsOTUB1 that deleting N terminus) against Lys48-diUb was monitored, in the presence of OsUBCH5B or hUBCH5B. The results showed that (Fig. S2D), the K48 activity of OsOTUB1-cat can also be promoted by both hUBCH5B and OsUBCH5B. Therefore, stimulation of OsOTUB1 against Lys48-diUb by both hUBCH5B and OsUBCH5B is not (or not completely) dependent on the N-terminus sequence. The mechanism that OsUBCH5B and hUBCH5B enhance the activity of OsOTUB1 needs to be further revealed. Please refer to the last paragraph in the part of “1. The discovery of plant DUBs that hydrolyze Met1 chains using an efficiently prepared Met1-diUb suicide probe”.

REVIEWER COMMENTS

Reviewer #1 (Remarks to the Author):

The preference shown by the catalytic construct of OsOTUB1 for M1-linked diUb over K48-diUb is rather marginal (Figure 1F; barely any difference in 4 minutes of reaction between M1 and K48 diUb substrates. Experiments shown in Figure 1F need to be run in triplicate and the product quantified to show clearly the preference for M1 chain, as this is a pivotal point of the manuscript.) Intriguingly, the full-length construct seems better at cleaving M1 than K48 substrate, which implies the missing segment may contribute to binding.

The marginal difference observed in Figure 1F could be offset by concentration effects under physiological settings. For example, K48 chains are likely to be much more abundant than linear polyUb chains in vivo (Figure 1D). In all likelihood, the observed preference will have no relevance under that situation.

A better way to present the paper would be to argue that OsOTUB1, unlike the human homolog, shows dual specificity toward both K48 and linear Ub chains. This could be an interesting point in itself. Then, the discussion pertaining to evolution (later part of the manuscript) can be rewritten, simplified, around this observation. Their structures explain linear Ub chain hydrolysis, the basis of this activity. Nearly all mutational results presented can be simply interpreted as explaining how M1-diUb is accepted as a substrate. It seems there is a predetermined notion of chain preference and the rest of the paper has been stretched to fit that idea. That may be why the paper is so hard to read and follow.

The preference argument is hard to sustain because the actual data do not comply with that (Figure 1F). Figure 3 also shows a similar modest effect in binding affinities for the three types of substrates. I do not think the K_d 's are different enough to be meaningful in the context of the argument.

The model presented in Figure 5 is mostly anecdotal. It does not follow from the results presented in the paper. None of the data seem to suggest there is a two-step recognition of the substrate.

The paper has numerous mistakes in grammar, sentence construction and inconsistencies. I wish the authors showed more care in preparing the final draft for submission. It is very hard to follow the arguments presented in the paper, even with a reasonable level of familiarity on this subject.

In spite of sincere additional efforts put into this edition compared to the previous submission, key points of the paper remain substantively unchanged. I do not think it ascends to the level of the journal.

Reviewer #2 (Remarks to the Author):

In this revised version of the manuscript the authors have included a significant number of new experiments, modified figures and they have concluded with a different hypothesis on the action mechanism of OsOTUB1 to cleave Met1-Ubiquitin substrates. In particular, they have discarded the “Bi-modal Activation”, present in the first version of the manuscript, for the “two-step adaptation” model, in the current version. Fortunately in my view (shared by the other reviewers), they have rejected the idea of the dual activation by the proximal and distal ubiquitin. As they show in the current version, the data did not support that “fancy” hypothesis about the “dual” activation.

This version of the manuscript has significantly improved compared to the previous one. The authors have basically conducted all the experiments required by the three reviewers, which have supposed an important effort in time and resources, but I think that it was worth as seen in the final version. The biochemical and mutagenesis experiments in the current version have significantly improved and support better the mechanistic hypothesis proposed by the authors.

A will only raise a few comments or issues.

Although the “bi-modal activation” has been discarded (from the title), it has been replaced by the “two-step adaptation” mechanism in the present manuscript. I don’t know whether such two-step mechanism has been properly demonstrated (please, check also the sentence in the abstract, since it is a little confusing to me and I would remove it from there). I don’t see the need to add such sequential steps details (and never in the title), unless it has been demonstrated by experimental or theoretical means (molecular dynamics). From my view binding of the distal ubiquitin already activates the enzyme, triggering an active site rearrangement to align the catalytic triad, and then only K48 or Met1-linked Ubiquitin in the proximal position can be accommodated in the S1’, including all the described contacts by the handles nicely proven in the manuscript. I don’t see the need to use terms such as “adaptation” in the mechanism which can lead to misunderstandings.

In the paragraph describing the structure (lines 253 to 275), the authors still insist in the fact of the activation of the catalytic triad by Proximal Ub Gln2. I thought that such idea had been discarded in this version. Binding of distal Ub is sufficient to remodel the catalytic triad...., without the need of Ub Gln2.

In the kinetic analysis I think that the Kcat reduction for Q2E is 41.3-fold, not 413.3 (please check the numbers). The results indicate that both catalysis and binding are affected, which is normal, since Q2 is essential for the proximal Ub binding and is quite close to the active site Histidine...

In line 335, I don't see that OsOTUB1 prefers Met1 over K48 and K63... In figure 1 the activity for K48 is quite significant. Please correct or explain better.

To my view the binding assay with SPR is not so informative and can be misleading. The Kd differences are too small to explain the strong different activities for Met1, K48 and K63 substrates, in particular for K63, which has a Kd quite similar to K48, but it has a weaker catalytic activity.

Sentence in line 528-529 is confusing (I think that you are comparing OsOTUB1 and hOTUB1). An issue a bit concerning: the structures of Ub diK48 and Ub diM1-linear are different, but since the binding of the distal Ub must be similar in both, the position of the proximal Ub should be different. I was expecting that the handles are specific for binding linear proximal Ub, but it seems from your results (lines 520-540) that they are also affecting the activity of K48 proximal Ub. From your model of diUb K48 with OsOTUB1, how can be similar to the binding of linear diUb, since the structure of diUb K48 is conformationally restricted?

Again, I am not sure on the two-step adaptation model, not sure on the sequential events...

In line 705, correct the sentence ("has been reported should be reported").

Reviewer #3 (Remarks to the Author):

In the revised manuscript the authors have enhanced the manuscript by the inclusion of kinetic and binding data to relate their structural observations of different structures of OsOTUB1 alone or in complex with Ub-PA or Met1 diUb-DHA. In addition, it was pleasing to see the effects of addition of either human or plant Ubch5b on the activity of OsOTUB1. Finally, the rationale for why hOTUB1 is unable to cleave Met1 chains can be clearly explained, and this justifies the logic of the sequence analysis looking at other species containing OTUB1. In addition, this reviewer had concerns regarding reproducibility and the inclusion of appropriate controls in the assays have addressed these concerns. As has the refinement of one of the OsOTUB1 structures.

However, this reviewer is still not in agreement about some of the conclusions reached by the authors presented in the main text and the rebuttal letter. In particular, the importance of distal Ub activation is not particularly conclusive and the model presented in Figure 5 represents what would be expected for a linkage selective DUB.

It is the opinion of this reviewer that this is an interesting study and the data presented is worthy of publication in Nature Communications. However, I do not understand why the authors still wish to pursue a narrative that now discusses a two-step adaptation method for OsOTUD1 function (previously was bi-modal activation). I still don't understand what this two-step adaptation method is.

The apo structure of OsOTUD1 has the catalytic His in an inactive configuration and consequently the orientation of the catalytic Cys is not in a configuration to be deprotonated and hence assumed not to be nucleophilic. Structures of Met1 diUb-DHA and Ub-PA reveal the catalytic His is in the correct configuration that would permit deprotonation of the Cys for nucleophilic attack. I agree with the authors that this is likely to happen upon distal Ub binding (that has to occur to allow the Ub-PA probe to react or facilitate cleavage of Ub-AMC). However, to relate the crystal structure of apo OsOTUB1 to a truly inactive state would require measuring the pKa of the catalytic Cys or another method to show the nucleophilicity of that Cys was impaired in an apo state. It is true that there are now several examples of DUBs that appear inactive in their apo structures, but one should always caution that the apo structure may not necessarily relate to the structure adopted in solution.

Surely the new model proposed in Figure 5 represents how any linkage selective DUB can work? in that specificity is driven by the S1' site of the DUB? Therefore, what is different about OsOTUB1?

The inclusion of the kinetic data shows different roles for the Ub proximal Gln2. Loss of the side chain to Ala or inclusion of a negative charge does impact the k_{cat} by several orders of magnitude and to some extent the K_M is affected by 2-4 orders of magnitude. The fact that distal Ub binding is sufficient for activity clearly means the Ub prox Gln2 is not required for catalysis (in contrast to

OTULIN where Ub prox Glu16 is absolutely required). However, the authors should not dismiss this all as an effect on binding to S1' owing to the very large differences in kcat. Hence, the authors should alter their conclusions accordingly.

The overall specificity of OsOTUB1 towards Met1 and a lesser extent Lys48 chains cannot solely be explained by binding as the differences in Kd between Met1, Lys48 and Lys63 chains is smaller than the observed hydrolysis of different diUb shown in Figure1. There is only a 2.5-fold difference in binding between Lys63 chains that are barely hydrolysed. Thus, it would not be surprising if there were additional effects on Kcat (see my earlier point above). The authors may want to reflect on this in their conclusions rather than attempt to show this experimentally, which would probably require kinetic analysis of various S1' mutants at the least, which would likely demonstrate that most binding "probably" occurs through the S1 site. Again, measuring binding of monoUb may give similar affinities to Lys63 diUb. If not, then some catalytic effect by the proximal Ub moiety should not be ruled out.

Finally, the inclusion of data demonstrating additional enhancement in OsOTUB1 activity by h/OsUbch5B is very interesting. Although, beyond the scope of the study, is there a reason why the activity was only tested against Lys48 chains? Wouldn't a similar activation be expected for Met1 chains too?

REVIEWER COMMENTS

Reviewer #1 (Remarks to the Author):

The preference shown by the catalytic construct of OsOTUB1 for M1-linked diUb over K48-diUb is rather marginal (Figure 1F; barely any difference in 4 minutes of reaction between M1 and K48 diUb substrates. Experiments shown in Figure 1F need to be run in triplicate and the product quantified to show clearly the preference for M1 chain, as this is a pivotal point of the manuscript.) Intriguingly, the full-length construct seems better at cleaving M1 than K48 substrate, which implies the missing segment may contribute to binding.

Response:

Thank you for your comments.

According to your suggestion, based on three replicates, we quantified the product percentage when M1-diUb and K48-diUb were used as the substrates of OsOTUB1^{cat}. The results (not shown in the manuscript) showed that, there was no significant activity difference of OsOTUB1^{cat} against M1-diUb and K48-diUb, demonstrating that OsOTUB1^{cat} shows no linkage preference for M1 and K48. As seen in the three replicates (please refer to the source data), it's true that OsOTUB1-FL prefers M1-diUb to K48-diUb, suggesting that the N-terminus part of OsOTUB1-FL plays roles in discriminating M1-diUb and K48-diUb, the mechanism of which needs further investigations.

Therefore, in the 5th. paragraph of the part “1. The discovery of plant DUBs that hydrolyze Met1 chains using an efficiently prepared Met1-diUb suicide probe”, the conclusion was changed to “However, OsOTUB1-cat showed marginal preference between Met1- and Lys48-diUb (hydrolysing almost the same percentage of diUbs within 4 min, Fig. 1F), indicating that the N-terminal part of OsOTUB1-FL plays roles in discriminating Met1-diUb and Lys48-diUb, the mechanism of which needs further investigation. Since we aimed to uncover the mechanism by which OsOTUB1 hydrolyses the Met1 chains, OsOTUB1-cat was used in all subsequent work, unless otherwise stated”. Please refer to the last 3 sentences in the 5th. paragraph of the part “1. The discovery of plant DUBs that hydrolyze Met1 chains using an efficiently prepared Met1-diUb suicide probe”.

Furthermore, the conclusions related to the concept that OsOTUB1-cat prefers Met1 to Lys48 were removed. We've restructured the manuscript in the following areas: ① most part of “3. Binding of proximal Ub facilitates the Met1 type preference of OsOTUB1-cat” was removed, while the last paragraph that structurally compared the binding of proximal Ub by OsOTUB1 with OTULIN was moved to the second paragraph of the part of “Discussion”; ② the title of the “4. Two motifs in S1' pocket determines Met1 type preference of OsOTUB1” was changed to “3. Two motifs in the S1' pocket determine the Met1 activity of OsOTUB1-cat”; ③ the part illuminating the N-handle motif and C-handle motif with XGY and G sequence mode, respectively, together with the effect of alterations in N-handle and C-handle motifs on the Lys48 activity for OsOTUB1-cat and hOTUB1-cat, were separated apart and formed the part of “4. The N- and C-handle motifs with XGY and G sequence modes, respectively, constitute the Met1-specific motif”; ④ the “Figure 4G” that compares Met1 and Lys48 activity of

hOTUB1-cat-P87G-SGS, the “Figure 5 Two-step adaptation during Met1-diUb cleavage by OsOTUB1” as well as its following interpretation in the part of “4. Two motifs in S1’ pocket determines Met1 type preference activity of OsOTUB1” were removed.

The marginal difference observed in Figure 1F could be offset by concentration effects under physiological settings. For example, K48 chains are likely to be much more abundant than linear polyUb chains in vivo (Figure 1D). In all likelihood, the observed preference will have no relevance under that situation.

Response:

Thank you for your comments.

It is true that K48 chains are much more abundant than Met1 chains *in vivo*, based on our results (Figure 1D) and previous reports. Thus, it is reasonable that the observed preference would have no relevance under that situation, which promotes us to remove the notion of linkage preference in the new version of manuscript.

Therefore, we’ve restructured the manuscript in the following areas: ① the conclusion that OsOTUB1^{cat} prefers Met1-diUb to Lys48-diUb in the fifth paragraph of the part “1. The discovery of plant DUBs that hydrolyze Met1 chains using an efficiently prepared Met1-diUb suicide probe” was removed; ② most part of “3. Binding of proximal Ub facilitates the Met1 type preference of OsOTUB1-cat” was removed, while the last paragraph that structurally compared the binding of proximal Ub by OsOTUB1 with OTULIN was moved to the second paragraph of the part of “Discussion”; ③ the title of the “4. Two motifs in S1’ pocket determines Met1 type preference of OsOTUB1” was changed to “3. Two motifs in the S1’ pocket determine the Met1 activity of OsOTUB1-cat”; ④ the part illuminating the N-handle motif and C-handle motif with XGY and G sequence mode, respectively, together with the effect of alterations in N-handle and C-handle motifs on the Lys48 activity for OsOTUB1-cat and hOTUB1-cat, were separated apart and formed the part of “4. The N- and C-handle motifs with XGY and G sequence modes, respectively, constitute the Met1-specific motif”; ⑤ the “Figure 4G” that compares Met1 and Lys48 activity of hOTUB1-cat-P87G-SGS, the “Figure 5 Two-step adaptation during Met1-diUb cleavage by OsOTUB1” as well as its following interpretation in the part of “4. Two motifs in S1’ pocket determines Met1 type preference activity of OsOTUB1” were removed.

A better way to present the paper would be to argue that OsOTUB1, unlike the human homolog, shows dual specificity toward both K48 and linear Ub chains. This could be an interesting point in itself. Then, the discussion pertaining to evolution (later part of the manuscript) can be rewritten, simplified, around this observation. Their structures explain linear Ub chain hydrolysis, the basis of this activity. Nearly all mutational results presented can be simply interpreted as explaining how M1-diUb is accepted as a substrate. It seems there is a predetermined notion of chain preference and the rest of the paper has been stretched to fit that idea. That may be why the paper is so hard to read and follow.

Response:

Thank you for your suggestions. I'm not sure if I fully understand your advice, so I tried to respond to you as follows.

Based on further confirmation, OsOTUB1^{cat} showed no linkage preference for Met1 and Lys48. Therefore, in this revised version, all the conclusions involved in linkage preference of OsOTUB1^{cat} were removed. The main thread of the revised version is to elucidate the determinants underlying the Met1 activity of OsOTUB1^{cat}.

It is true that the part of “4. Two motifs in S1' pocket determines Met1 type preference of OsOTUB1” was too complicated to be clearly understood. So this part was restructured as following: ① the title was changed to “3. Two motifs in the S1' pocket determine the Met1 activity of OsOTUB1-cat”; ② the discussion pertaining to evolution of hOTUB1 was moved to the third paragraph, which just follows the effect of alterations in N-handle motif and C-handle motif on hOTUB1's Met1 activity; ③ the part illuminating the N-handle motif and C-handle motif with XGY and G sequence mode, respectively, together with the effect of alterations in N-handle and C-handle motifs on the Lys48 activity for OsOTUB1-cat and hOTUB1-cat, were separated apart and formed the part of “4. The N- and C-handle motifs with XGY and G sequence modes, respectively, constitute the Met1-specific motif”; ④ the “Figure 4G” that compares Met1 and Lys48 activity of hOTUB1-cat-P87G-SGS, the “Figure 5 Two-step adaptation during Met1-diUb cleavage by OsOTUB1” as well as its following interpretation in the part of “4. Two motifs in S1' pocket determines Met1 type preference activity of OsOTUB1” were removed.

Although both OsOTUB1 and hOTUB1 contain S1' pockets, it is certain that these S1' pockets are different when adapting to the proximal ubiquitin of ubiquitin chains with different linkage types, due to the different spatial conformation of the proximal ubiquitin. In this manuscript, the structure based biochemical results indeed demonstrated that the S1' pocket consisting of the N-handle and the C-handle motifs determine the M1 activity of OsOTUB1 and hOTUB1 (Figure 3C and 3D). And further, the S1' pocket determined by the N-handle and C-handle motifs was adapted to the proximal ubiquitin of Met1 chains, but not the one adapted to other linkage types including at least Lys48 (Figure S9B and S9C). Therefore, other experimental results in the rest of this paper are re-verifications for this model, rather than been stretched to fit this model.

The preference argument is hard to sustain because the actual data do not comply with that (Figure 1F). Figure 3 also shows a similar modest effect in binding affinities for the three types of substrates. I do not think the Kd's are different enough to be meaningful in the context of the argument.

Response:

Thank you for your comments.

According to your suggestion, based on three replicates, we quantified the product percentage when M1-diUb and K48-diUb were used as the substrates of OsOTUB1^{cat}. The results (not shown in the manuscript) showed that, there was no significant activity difference of OsOTUB1^{cat} against

M1-diUb and K48-diUb. When it comes to Lys63-diUb, it is true that the 2.5-fold reduction of binding cannot account for the significant decrease in hydrolytic activity. There should be other determinants.

Therefore, the main thread of the revised version is to elucidate the determinants underlying the Met1 activity of OsOTUB1^{cat}. We've restructured the manuscript in the following areas: ① the conclusion that OsOTUB1^{cat} prefers Met1-diUb to Lys48-diUb in the fifth paragraph of the part "1. The discovery of plant DUBs that hydrolyze Met1 chains using an efficiently prepared Met1-diUb suicide probe" was removed; ② most part of "3. Binding of proximal Ub facilitates the Met1 type preference of OsOTUB1-cat" was removed, while the last paragraph that structurally compared the binding of proximal Ub by OsOTUB1 with OTULIN was moved to the second paragraph of the part of "Discussion"; ③ the title of the "4. Two motifs in S1' pocket determines Met1 type preference of OsOTUB1" was changed to "3. Two motifs in the S1' pocket determine the Met1 activity of OsOTUB1-cat"; ④ the part illuminating the N-handle motif and C-handle motif with XGY and G sequence mode, respectively, together with the effect of alterations in N-handle and C-handle motifs on the Lys48 activity for OsOTUB1-cat and hOTUB1-cat, were separated apart and formed the part of "4. The N- and C-handle motifs with XGY and G sequence modes, respectively, constitute the Met1-specific motif"; ⑤ the "Figure 4G" that compares Met1 and Lys48 activity of hOTUB1-cat-P87G-SGS, the "Figure 5 Two-step adaptation during Met1-diUb cleavage by OsOTUB1" as well as its following interpretation in the part of "4. Two motifs in S1' pocket determines Met1 type preference activity of OsOTUB1" were removed.

The model presented in Figure 5 is mostly anecdotal. It does not follow from the results presented in the paper. None of the data seem to suggest there is a two-step recognition of the substrate.

Response:

Thank you for your comments.

I have to admit that the two-step adaptation model is based mainly on speculation, while the only experimental evidence that hOTUB1-cat-P87G-SGS prefers Met1-diUb to Lys48-diUb is also unconvincing. Thus, in this revised version, the main thread is to elucidate the determinants underlying the Met1 activity of OsOTUB1-cat. All the conclusions involved in linkage preference of OsOTUB1^{cat} and the model of two-step recognition of the substrate were removed.

You will find that, the revised version was restructured in the following areas: ① the conclusion that OsOTUB1^{cat} prefers Met1-diUb to Lys48-diUb in the fifth paragraph of the part "1. The discovery of plant DUBs that hydrolyze Met1 chains using an efficiently prepared Met1-diUb suicide probe" was removed; ② most part of "3. Binding of proximal Ub facilitates the Met1 type preference of OsOTUB1-cat" was removed, while the last paragraph that structurally compared the binding of proximal Ub by OsOTUB1 with OTULIN was moved to the second paragraph of the part of "Discussion"; ③ the title of the "4. Two motifs in S1' pocket determines Met1 type preference of OsOTUB1" was changed to "3. Two motifs in the S1' pocket determine the Met1 activity of OsOTUB1-cat"; ④ the part illuminating the N-handle motif and C-handle motif with XGY and G sequence mode, respectively, together with the effect of alterations in N-handle and C-handle motifs on the Lys48 activity for OsOTUB1-cat and hOTUB1-cat, were

separated apart and formed the part of “4. The N- and C-handle motifs with XGY and G sequence modes, respectively, constitute the Met1-specific motif”; ⑤ the “Figure 4G” that compares Met1 and Lys48 activity of hOTUB1-cat-P87G-SGS, the “Figure 5 Two-step adaptation during Met1-diUb cleavage by OsOTUB1” as well as its following interpretation in the part of “4. Two motifs in S1’ pocket determines Met1 type preference activity of OsOTUB1” were removed.

The paper has numerous mistakes in grammar, sentence construction and inconsistencies. I wish the authors showed more care in preparing the final draft for submission. It is very hard to follow the arguments presented in the paper, even with a reasonable level of familiarity on this subject.

Response:

Thank you for your comments. I’m sorry for causing so much trouble to your reading due to my poor English. And thank you very much for your patience upon my manuscript.

According to your suggestion, the language of the revised manuscript was edited by the Nature Research Editing Service (please refer to the Editing_Certificate). Please re-read the manuscript and I would appreciate if you could give further advice.

In spite of sincere additional efforts put into this edition compared to the previous submission, key points of the paper remain substantively unchanged. I do not think it ascends to the level of the journal.

Response:

Thank you for your comments.

I’ve substantially restructured the manuscript. In the revised version, the main thread is to elucidate the determinants underlying the Met1 activity of OsOTUB1-cat. All the conclusions involved in linkage preference of OsOTUB1^{cat} and the model of two-step recognition of the substrate were removed. For detailed information, please refer to the responses to your previous comments/questions.

Reviewer #2 (Remarks to the Author):

In this revised version of the manuscript the authors have included a significant number of new experiments, modified figures and they have concluded with a different hypothesis on the action mechanism of OsOTUB1 to cleave Met1-Ubiquitin substrates. In particular, they have discarded the “Bi-modal Activation”, present in the first version of the manuscript, for the “two-step adaptation” model, in the current version. Fortunately in my view (shared by the other reviewers), they have rejected the idea of the dual activation by the proximal and distal ubiquitin. As they show in the current version, the data did not support that “fancy” hypothesis about the “dual” activation.

This version of the manuscript has significantly improved compared to the previous one. The authors have basically conducted all the experiments required by the three reviewers, which have supposed an important effort in time and resources, but I think that it was worth as seen in the final version. The biochemical and mutagenesis experiments in the current version have significantly improved and support better the mechanistic hypothesis proposed by the authors.

Response:

Thank you for your comments.

Thanks to your reviewers' suggestions, we've made the following improvements:

1. Based on three replicates, we quantified the product percentage when Met1-diUb and Lys48-diUb were used as the substrates of OsOTUB1-cat. The results (not shown in the manuscript) showed that, there was no significant activity difference of OsOTUB1-cat against M1-diUb and K48-diUb, demonstrating that OsOTUB1-cat shows no linkage preference for M1 and K48, which is consistent with the reasonable judgement that the Kd differences would not account for the different activity of OsOTUB1-cat against Met1- and Lys48-diUb, especially for Lys63-diUb. Therefore, in the revised manuscript, all the figures and conclusions involved in linkage preference of OsOTUB1-cat were removed, while the main thread is to elucidate the determinants underlying the Met1 activity of OsOTUB1-cat. You will find that, the revised version was restructured in the following areas: ① the conclusion that OsOTUB1^{cat} prefers Met1-diUb to Lys48-diUb in the fifth paragraph of the part “1. The discovery of plant DUBs that hydrolyze Met1 chains using an efficiently prepared Met1-diUb suicide probe” was removed; ② most part of “3. Binding of proximal Ub facilitates the Met1 type preference of OsOTUB1-cat” was removed, while the last paragraph that structurally compared the binding of proximal Ub by OsOTUB1 with OTULIN was moved to the second paragraph of the part of “Discussion”; ③ the title of the “4. Two motifs in S1' pocket determines Met1 type preference of OsOTUB1” was changed to “3. Two motifs in the S1' pocket determine the Met1 activity of OsOTUB1-cat”; ④ the part illuminating the N-handle motif and C-handle motif with XGY and G sequence mode, respectively, together with the effect of alterations in N-handle and C-handle motifs on the Lys48 activity for OsOTUB1-cat and hOTUB1-cat, were separated apart and formed the part of “4. The N- and C-handle motifs with XGY and G sequence modes, respectively, constitute the Met1-specific motif”; ⑤ the “Figure 4G” that compares Met1 and Lys48 activity of hOTUB1-cat-P87G-SGS, the “Figure 5 Two-step adaptation during Met1-diUb cleavage by

OsOTUB1” as well as its following interpretation in the part of “4. Two motifs in S1’ pocket determines Met1 type preference activity of OsOTUB1” were removed.

2. I have to admit that the two-step adaptation model is based mainly on speculation, while the only experimental evidence that hOTUB1-cat-P87G-SGS prefers Met1-diUb to Lys48-diUb is also unconvincing. Thus, in this revised version, the title was changed to “Met1-specific Motifs conserved in OTUB subfamily of green plants enable rice OTUB1 to Hydrolyse Met1 Ubiquitin Chains”, the sentences involved in “two-step adaptation” in the “Abstract”, the “Figure 4G” that compares Met1 and Lys48 activity of hOTUB1-cat-P87G-SGS, the “Figure 5 Two-step adaptation during Met1-diUb cleavage by OsOTUB1” as well as its following interpretation in the part of “4. Two motifs in S1’ pocket determines Met1 type preference activity of OsOTUB1” were removed.

In conclusion, in the revised version, the main thread is to elucidate the determinants underlying the Met1 activity of OsOTUB1-cat. All the conclusions involved in linkage preference of OsOTUB1-cat and the model of two-step recognition of the substrate were removed.

A will only raise a few comments or issues.

Although the “bi-modal activation” has been discarded (from the title), it has been replaced by the “two-step adaptation” mechanism in the present manuscript. I don’t know whether such two-step mechanism has been properly demonstrated (please, check also the sentence in the abstract, since it is a little confusing to me and I would remove it from there). I don’t see the need to add such sequential steps details (and never in the title), unless it has been demonstrated by experimental or theoretical means (molecular dynamics). From my view binding of the distal ubiquitin already activates the enzyme, triggering an active site rearrangement to align the catalytic triad, and then only K48 or Met1-linked Ubiquitin in the proximal position can be accommodated in the S1’, including all the described contacts by the handles nicely proven in the manuscript. I don’t see the need to use terms such as “adaptation” in the mechanism which can lead to misunderstandings.

Response:

Thank you for your comments.

I have to admit that the two-step adaptation model is based mainly on speculation, while the only experimental evidence that hOTUB1-cat-P87G-SGS prefers Met1-diUb to Lys48-diUb is also unconvincing. Thus, in this revised version, the “Figure 4G” that compares Met1 and Lys48 activity of hOTUB1-cat-P87G-SGS, the “Figure 5 Two-step adaptation during Met1-diUb cleavage by OsOTUB1” as well as its following interpretation in the part of “4. Two motifs in S1’ pocket determines Met1 type preference activity of OsOTUB1” were removed. In the revised version, the main thread is to elucidate the determinants underlying the Met1 activity of OsOTUB1-cat. All the conclusions involved in linkage preference of OsOTUB1-cat and the model of two-step recognition of the substrate were removed.

Therefore, the title of the updated manuscript was changed to “Met1-specific Motifs conserved in OTUB subfamily of green plants enable rice OTUB1 to Hydrolyse Met1 Ubiquitin Chains”, and

the sentences involved in “two-step adaptation” were removed. Please refer to the title and the part of “Abstract”.

In the paragraph describing the structure (lines 253 to 275), the authors still insist in the fact of the activation of the catalytic triad by Proximal Ub Gln2. I thought that such idea had been discarded in this version. Binding of distal Ub is sufficient to remodel the catalytic triad...., without the need of Ub Gln2.

Response:

Thank you for your comments.

It’s true that the concept of activation of the catalytic triad by Proximal Ub Gln2 was denied. The paragraph (lines 253 to 275) described the effect of mutations at Q2^{prox} on the activity of OsOTUB1-cat, which showed that activity of OsOTUB1-cat is increasing as the length of side chain gradually extends, while much more significantly inhibited when the side chain bears negative charge, seemingly supporting the hypothesis that Q2^{prox} activates OsOTUB1-cat by relocating the imidazole ring of His317. However, the following experiments measuring the kinetics of OsOTUB1-cat in the presence of Met1-diUb mutants did not support this concept. But we still cannot completely rule out the possibility that Q2^{prox} may play roles in activating OsOTUB1-cat, due to the seemingly unparallel decrease between k_{cat} and K_M , especially for Met1-diUb Q2E^{prox} (k_{cat} decreased 41.33-fold while K_M decreased 5.11-fold). Thus, in this revised version, the title of the 2nd part was changed to “2. Activation of OsOTUB1-cat is likely induced by distal Ub but may not by proximal Ub”. Please refer to the part of “Activation of OsOTUB1-cat is likely induced by distal Ub but may not by proximal Ub”.

In the kinetic analysis I think that the Kcat reduction for Q2E is 41.3-fold, not 413.3 (please check the numbers). The results indicate that both catalysis and binding are affected, which is normal, since Q2 is essential for the proximal Ub binding and is quite close to the active site Histidine...

Response:

Thank you for your comments.

Sorry for such a low-level error. In the revised version, the value has been corrected to 41.33-fold. Please refer to the sixth sentence of the fourth paragraph in the part of “Activation of OsOTUB1-cat is likely induced by distal Ub but may not by proximal Ub”.

In line 335, I don’t see that OsOTUB1 prefers Met1 over K48 and K63... In figure 1 the activity for K48 is quite significant. Please correct or explain better.

Response:

Thank you for your comments.

In response to the linkage preference of OsOTUB1-cat, we further quantified the product percentage when Met1-diUb and Lys48-diUb were used as the substrates of OsOTUB1-cat, based on three replicates. The results (not shown in the manuscript) showed that, there was no significant activity difference of OsOTUB1-cat against Met1-diUb and Lys48-diUb, demonstrating that OsOTUB1-cat shows no linkage preference for Met1 and Lys48. When it comes to Lys63-diUb, it is true that the 2.5-fold reduction of binding cannot account for the significant decrease in hydrolytic activity. There should be other determinants.

Therefore, in the 5th. paragraph of the part “1. The discovery of plant DUBs that hydrolyse Met1 chains using an efficiently prepared Met1-diUb suicide probe”, the conclusion was changed to “However, OsOTUB1-cat showed marginal preference between Met1- and Lys48-diUb (hydrolysing almost the same percentage of diUbs within 4 min, Fig. 1F), indicating that the N-terminal part of OsOTUB1-FL plays roles in discriminating Met1-diUb and Lys48-diUb, the mechanism of which needs further investigation. Since we aimed to uncover the mechanism by which OsOTUB1 hydrolyses the Met1 chains, OsOTUB1-cat was used in all subsequent work, unless otherwise stated”. Please refer to the last 3 sentences in the 5th. paragraph of the part “1. The discovery of plant DUBs that hydrolyse Met1 chains using an efficiently prepared Met1-diUb suicide probe”.

Furthermore, the conclusions related to the concept that OsOTUB1-cat prefers Met1 to Lys48 were removed, while the main thread is to elucidate the determinants underlying the Met1 activity of OsOTUB1-cat. We’ve restructured the manuscript in the following areas: ① the conclusion that OsOTUB1^{cat} prefers Met1-diUb to Lys48-diUb in the fifth paragraph of the part “1. The discovery of plant DUBs that hydrolyze Met1 chains using an efficiently prepared Met1-diUb suicide probe” was removed; ② most part of “3. Binding of proximal Ub facilitates the Met1 type preference of OsOTUB1-cat” was removed, while the last paragraph that structurally compared the binding of proximal Ub by OsOTUB1 with OTULIN was moved to the second paragraph of the part of “Discussion”; ③ the title of the “4. Two motifs in S1’ pocket determines Met1 type preference of OsOTUB1” was changed to “3. Two motifs in the S1’ pocket determine the Met1 activity of OsOTUB1-cat”; ④ the part illuminating the N-handle motif and C-handle motif with XGY and G sequence mode, respectively, together with the effect of alterations in N-handle and C-handle motifs on the Lys48 activity for OsOTUB1-cat and hOTUB1-cat, were separated apart and formed the part of “4. The N- and C-handle motifs with XGY and G sequence modes, respectively, constitute the Met1-specific motif”; ⑤ the “Figure 4G” that compares Met1 and Lys48 activity of hOTUB1-cat-P87G-SGS, the “Figure 5 Two-step adaptation during Met1-diUb cleavage by OsOTUB1” as well as its following interpretation in the part of “4. Two motifs in S1’ pocket determines Met1 type preference activity of OsOTUB1” were removed.

To my view the binding assay with SPR is not so informative and can be misleading. The Kd differences are too small to explain the strong different activities for Met1, K48 and K63 substrates, in particular for K63, which has a Kd quite similar to K48, but it has a weaker catalytic activity.

Response:

Thank you for your comments.

Based on three replicates, we quantified the product percentage when Met1-diUb and Lys48-diUb were used as the substrates of OsOTUB1-cat. The results (not shown in the manuscript) showed that, there was no significant activity difference of OsOTUB1-cat against M1-diUb and K48-diUb, demonstrating that OsOTUB1-cat shows no linkage preference for M1 and K48. When it comes to Lys63-diUb, it is true that the 2.5-fold reduction of binding cannot account for the significant decrease in hydrolytic activity. There should be other determinants.

Therefore, the conclusions related to the concept that OsOTUB1-cat prefers Met1 to Lys48 were removed, while the main thread is to elucidate the determinants underlying the Met1 activity of OsOTUB1-cat. We've restructured the manuscript in the following areas: ① the conclusion that OsOTUB1^{cat} prefers Met1-diUb to Lys48-diUb in the fifth paragraph of the part "1. The discovery of plant DUBs that hydrolyze Met1 chains using an efficiently prepared Met1-diUb suicide probe" was removed; ② most part of "3. Binding of proximal Ub facilitates the Met1 type preference of OsOTUB1-cat" was removed, while the last paragraph that structurally compared the binding of proximal Ub by OsOTUB1 with OTULIN was moved to the second paragraph of the part of "Discussion"; ③ the title of the "4. Two motifs in S1' pocket determines Met1 type preference of OsOTUB1" was changed to "3. Two motifs in the S1' pocket determine the Met1 activity of OsOTUB1-cat"; ④ the part illuminating the N-handle motif and C-handle motif with XGY and G sequence mode, respectively, together with the effect of alterations in N-handle and C-handle motifs on the Lys48 activity for OsOTUB1-cat and hOTUB1-cat, were separated apart and formed the part of "4. The N- and C-handle motifs with XGY and G sequence modes, respectively, constitute the Met1-specific motif"; ⑤ the "Figure 4G" that compares Met1 and Lys48 activity of hOTUB1-cat-P87G-SGS, the "Figure 5 Two-step adaptation during Met1-diUb cleavage by OsOTUB1" as well as its following interpretation in the part of "4. Two motifs in S1' pocket determines Met1 type preference activity of OsOTUB1" were removed.

Sentence in line 528-529 is confusing (I think that you are comparing OsOTUB1 and hOTUB1). An issue a bit concerning: the structures of Ub diK48 and Ub diM1-linear are different, but since the binding of the distal Ub must be similar in both, the position of the proximal Ub should be different. I was expecting that the handles are specific for binding linear proximal Ub, but it seems from your results (lines 520-540) that they are also affecting the activity of K48 proximal Ub. From your model of diUb K48 with OsOTUB1, how can be similar to the binding of linear diUb, since the structure of diUb K48 is conformationally restricted?

Response:

Thank you for your comments.

- ① Very sorry for causing your confusing due to my imprecise expression. According to your confusing, in the part of "3. Two motifs in the S1' pocket determine Met1 activity of OsOTUB1-cat", the first sentence "As the residues binding to proximal Ub in OsOTUB1 are also conserved in hOTUB1 (Fig. S3), it appears there might exist extra determinant to confer Met1 activity on OsOTUB1 other than hOUTB1." was revised to "As the residues binding to the proximal Ub of Met1-diUb in OsOTUB1 are also conserved in hOTUB1 (Fig. S3), it

appears that there might be an extra determinant to confer Met1 activity on OsOTUB1 other than hOUTB1.”

- ② It's true that the N-handle motif and C-handle motif with XGY and G sequence mode, respectively, are Met1-specific, but not Lys48-specific, because our results (Fig. S9B and S9C) proved that C-handle motif (G/P) and N-handle motif (SGS/EDD) contributes equally to activity against Met1-diUb, but oppositely to that of Lys48-diUb. These results indicate that OsOTUB1-cat and hOTUB1-cat-P87G-SGS adopt different strategies to bind to Met1-diUb and Lys48-diUb, probably due to the distinct configuration between the two linkage-type chains (Please refer to the last two sentences in the part of “4. The N- and C-handle motifs with XGY and G sequence modes, respectively, constitute the Met1-specific motif”). This assumption could also be seen in the structural alignment of OsOTUB1-cat~Met1-diUb-DHA with Ub^{distal}~UbcH5b^{C85S}~hOTUB1^{Δ1-24}-Ub^{prox} (Fig. S10A) as explained in the 4th. paragraph of the part of “Discussion”.
- ③ In summary, although both OsOTUB1 and hOTUB1 contain S1' pockets, it is certain that these S1' pockets are different when adapting to the proximal ubiquitin of ubiquitin chains with different linkage types, due to the different spatial conformation of the proximal ubiquitin. In this manuscript, the structure based biochemical results indeed demonstrated that the S1' pocket consisting of the N-handle and the C-handle motifs determine the M1 activity of OsOTUB1 and hOTUB1 (Figure 3C and 3D). And further, the S1' pocket determined by the N-handle and C-handle motifs was adapted to the proximal ubiquitin of Met1 chains, but not the one adapted to other linkage types including at least Lys48 (Figure S9B and S9C). These results indicated the differences between the S1' pockets that adapted to Met1 chains and Lys48 chains. However, they are unlikely completely separated from each other. There may inevitably be overlap between them, which would result in different consequences for Met1 activity and Lys48 activity of OsOTUB1 and hOTUB1 when the C-handle motif (G/P) and N-handle motif (SGS/EDD) are mutated.

Again, I am not sure on the two-step adaptation model, not sure on the sequential events...

Response:

Thank you for your comments.

I have to admit that the two-step adaptation model is based mainly on speculation, while the only experimental evidence that hOTUB1-cat-P87G-SGS prefers Met1-diUb to Lys48-diUb is also unconvincing. Thus, in this revised version, the “Figure 4G” that compares Met1 and Lys48 activity of hOTUB1-cat-P87G-SGS, the “Figure 5 Two-step adaptation during Met1-diUb cleavage by OsOTUB1” as well as its following interpretation in the part of “4. Two motifs in S1' pocket determines Met1 type preference activity of OsOTUB1” were removed.

In line 705, correct the sentence (“has been reported should be reported”).

Response:

Thank you for your comments.

But I'm sorry that I cannot find "has been reported" in line 705 ("not in animals. Recent studies have sequenced and analyzed the genome and transcriptome of"), which may probably due to some mistake for the number labeling.

Therefore, in the old version I've checked all the sentences containing "has been reported". These sentences are ① line 7-10 "However, few plant-derived deubiquitinases (DUBs) against Met1 ubiquitin chains has been reported, and the selection mechanism for linkage-type of ubiquitin chains remains elusive, which greatly limits our understanding of the functions of plant-derived DUBs."; ② line 671-674 "It has been reported that hOTUB1 specifically hydrolyzes Lys48 chains (Edelmann et al., 2009), and hOTUB2 hydrolyzes both Lys48 and Lys63 chains (Altun et al., 2015), but neither can hydrolyze Met1 chains."; ③ line 687-691 "Fortunately, Juang *et al.* has been reported the crystal structure of Ub~UbcH5b^{C85S}~hOTUB^{Δ1-24} complexed with free Ub (Ub^{distal}~UbcH5b^{C85S}~hOTUB1^{Δ1-24}-Ub^{prox}, Juang et al. 2012), which includes two Ubs adopting Lys48-diUb configuration and may provide clues for interaction between Lys48-diUb and hOTUB1.". I think the "has been reported" in ③ is probably the most likely candidate. So I've corrected the ③ to "Fortunately, Juang *et al.* reported the crystal structure of Ub~UbcH5b^{C85S}~hOTUB^{Δ1-24} complexed with free Ub (Ub^{distal}~UbcH5b^{C85S}~hOTUB1^{Δ1-24}-Ub^{prox}, Juang et al. 2012), which includes two Ubs adopting the Lys48-diUb configuration and may provide clues for the interaction between Lys48-diUb and hOTUB1.". Please refer to the 3rd sentence of the 4th paragraph in the part of "Discussion" in the revised version.

Reviewer #3 (Remarks to the Author):

In the revised manuscript the authors have enhanced the manuscript by the inclusion of kinetic and binding data to relate their structural observations of different structures of OsOTUB1 alone or in complex with Ub-PA or Met1 diUb-DHA. In addition, it was pleasing to see the effects of addition of either human or plant Ubch5b on the activity of OsOTUB1. Finally, the rationale for why hOTUB1 is unable to cleave Met1 chains can be clearly explained, and this justifies the logic of the sequence analysis looking at other species containing OTUB1. In addition, this reviewer had concerns regarding reproducibility and the inclusion of appropriate controls in the assays have addressed these concerns. As has the refinement of one of the OsOTUB1 structures.

However, this reviewer is still not in agreement about some of the conclusions reached by the authors presented in the main text and the rebuttal letter. In particular, the importance of distal Ub activation is not particularly conclusive and the model presented in Figure 5 represents what would be expected for a linkage selective DUB.

Response:

Thank you for your comments. The followings are responses to the two comments.

For the distal Ub activation, it is true that the snap shot of the crystal structure of apo OsOTUB1-cat that reflects the inactivation state (Fig. 2B) may not be that in solution. Therefore, according to your comments, I've toned down the statement of distal Ub activation: ① the title of "2. Activation of OsOTUB1-cat is induced by distal Ub, but not proximal Ub" was replaced by "2. Activation of OsOTUB1-cat is likely induced by distal Ub but may not by proximal Ub"; ② a brief discussion was supplemented at the end of the part of "2. Activation of OsOTUB1-cat is likely induced by distal Ub but may not by proximal Ub", that is, "However, given the condition differences between the crystal and solution, the snap shot of the crystal structure of apo OsOTUB1-cat that reflects the inactivation state (Fig. 2B) may not be that in solution."

For the two-step adaptation model presented by Figure 5, I have to admit that the two-step adaptation model is based mainly on speculation, while the only experimental evidence that hOTUB1-cat-P87G-SGS prefers Met1-diUb to Lys48-diUb is also unconvincing. Thus, in this revised version, the "Figure 4G" that compares Met1 and Lys48 activity of hOTUB1-cat-P87G-SGS, the "Figure 5 Two-step adaptation during Met1-diUb cleavage by OsOTUB1" as well as its following interpretation in the part of "4. Two motifs in S1' pocket determines Met1 type preference activity of OsOTUB1" were removed.

It is the opinion of this reviewer that this is an interesting study and the data presented is worthy of publication in Nature Communications. However, I do not understand why the authors still wish to pursue a narrative that now discusses a two-step adaption method for OsOTUD1 functon(previously was bi-modal activation). I still don't understand what this two-step adaption method is.

Response:

Thank you very much for your comments.

I have to admit that the two-step adaptation model is based mainly on speculation, while the only experimental evidence that hOTUB1-cat-P87G-SGS prefers Met1-diUb to Lys48-diUb is also unconvincing. Thus, in this revised version, the main thread is to elucidate the determinants underlying the Met1 activity of OsOTUB1-cat, while the two-step adaptation was abandoned. You can find that the “Figure 4G” that compares Met1 and Lys48 activity of hOTUB1-cat-P87G-SGS, the “Figure 5 Two-step adaptation during Met1-diUb cleavage by OsOTUB1” as well as its following interpretation in the part of “4. Two motifs in S1’ pocket determines Met1 type preference activity of OsOTUB1” were removed.

The apo structure of OsOTUD1 has the catalytic His in an inactive configuration and consequently the orientation of the catalytic Cys is not in a configuration to be deprotonated and hence assumed not to be nucleophilic. Structures of Met1 diUb-DHA and Ub-PA reveal the catalytic His is in the correct configuration that would permit deprotonation of the Cys for nucleophilic attack. I agree with the authors that this is likely to happen upon distal Ub binding (that has to occur to allow the Ub-PA probe to react or facilitate cleavage of Ub-AMC). However, to relate the crystal structure of apo OsOTUB1 to a truly inactive state would require measuring the pKa of the catalytic Cys or another method to show the nucleophilicity of that Cys was impaired in an apo state. It is true that there are now several examples of DUBs that appear inactive in their apo structures, but one should always caution that the apo structure may not necessarily relate to the structure adopted in solution.

Response:

Thank you very much for your comments! These comments demonstrate your excellent rigor. Our manuscript benefits greatly from them.

I have to admit that your comments illuminate the logic problems that exist in many studies revealing the activation mechanism of DUB, included with the case of our manuscript. This is probably due to the fact that many related works only stayed at the stage of explaining phenomena with phenomena, rather than revealing phenomena with chemical nature.

According to your comments, I’ve toned down the statement of distal Ub activation: ① the title of “2. Activation of OsOTUB1-cat is induced by distal Ub, but not proximal Ub” was replaced by “2. Activation of OsOTUB1-cat is likely induced by distal Ub but may not by proximal Ub”; ② a brief discussion was supplemented at the end of the part of “2. Activation of OsOTUB1-cat is likely induced by distal Ub but may not by proximal Ub”, that is, “However, given the condition differences between the crystal and solution, the snap shot of the crystal structure of apo OsOTUB1-cat that reflects the inactivation state (Fig. 2B) may not be that in solution.”.

Surely the new model proposed in Figure 5 represents how any linkage selective DUB can work? In that specificity is driven by the S1’ site of the DUB? Therefore, what is different about OsOTUB1?

Response:

Thank you for your questions.

I have to admit that the two-step adaptation model is based mainly on speculation, while the only experimental evidence that hOTUB1-cat-P87G-SGS prefers Met1-diUb to Lys48-diUb is also unconvincing. Thus, in this revised version, the main thread is to elucidate the determinants underlying the Met1 activity of OsOTUB1-cat, while the two-step adaptation was abandoned. You can find that the “Figure 4G” that compares Met1 and Lys48 activity of hOTUB1-cat-P87G-SGS, the “Figure 5 Two-step adaptation during Met1-diUb cleavage by OsOTUB1” as well as its following interpretation in the part of “4. Two motifs in S1’ pocket determines Met1 type preference activity of OsOTUB1” were removed.

The inclusion of the kinetic data shows different roles for the Ub proximal Gln2. Loss of the side chain to Ala or inclusion of a negative charge does impact the k_{cat} by several orders of magnitude and to some extent the K_M is affected by 2-4 orders of magnitude. The fact that distal Ub binding is sufficient for activity clearly means the Ub prox Gln2 is not required for catalysis (in contrast to OTULIN where Ub prox Glu16 is absolutely required). However, the authors should not dismiss this all as an effect on binding to S1’ owing to the very large differences in k_{cat} . Hence, the authors should alter their conclusions accordingly.

Response:

Thank you very much for your comments! These comments demonstrate your excellent rigor. Our manuscript benefits greatly from them.

According to your comments, I’ve toned down the statement that proximal ubiquitin cannot activate OsOTUB1-cat. Thus, ① the title of “2. Activation of OsOTUB1-cat is induced by distal Ub, but not proximal Ub” was replaced with “2. Activation of OsOTUB1-cat is likely induced by distal Ub but may not by proximal Ub”; ② at the last of the fourth paragraph in the part of “2. Activation of OsOTUB1-cat is likely induced by distal Ub but may not by proximal Ub”, “These results demonstrated that mutations on Q2^{prox} will affect the affinity of OsOTUB1-cat with substrate, thereby altering the catalysis ability of OsOTUB1-cat, but not hampering Q2^{prox}-mediated activation of OsOTUB1-cat.” was replaced by “These results implied that mutations in Q2^{prox} may affect the affinity of OsOTUB1-cat for the substrate, thereby altering the catalytic ability of OsOTUB1-cat.”; ③ a short discussion was added at the end of the fourth paragraph in the part of “2. Activation of OsOTUB1-cat is likely induced by distal Ub but may not by proximal Ub”, that is, “However, we still cannot completely rule out the possibility that Q2^{prox} may play roles in activating OsOTUB1-cat due to the seemingly unparallel decrease between k_{cat} and K_M , especially for Met1-diUb Q2E^{prox} (k_{cat} decreased 41.33-fold while K_M elevated only 5.11-fold).

The overall specificity of OsOTUB1 towards Met1 and a lesser extent Lys48 chains cannot solely be explained by binding as the differences in K_d between Met1, Lys48 and Lys63 chains is smaller than the observed hydrolysis of different diUb shown in Figure1. There is only a 2.5-fold difference in binding between Lys63 chains that are barely hydrolysed. Thus, it would not be surprising if there were additional effects on K_{cat} (see my earlier point above). The authors may

want to reflect on this in their conclusions rather than attempt to show this experimentally, which would probably require kinetic analysis of various S1' mutants at the least, which would likely demonstrate that most binding “probably” occurs through the S1 site. Again, measuring binding of monoUb may give similar affinities to Lys63 diUb. If not, then some catalytic effect by the proximal Ub moiety should not be ruled out.

Response:

Thank you very much for your inspiring comments!

To re-check the activity difference of OsOTUB1-cat against Met1-diUb and Lys48-diUb, we quantified the product percentage when Met1-diUb and Lys48-diUb were used as the substrates of OsOTUB1-cat. The results (not shown in the manuscript) showed that, there was no significant activity difference of OsOTUB1-cat against Met1-diUb and Lys48-diUb, demonstrating that OsOTUB1-cat shows no linkage preference for Met1 and Lys48, which is consistent with the reasonable judgement that the Kd differences would not account for the different activity of OsOTUB1-cat against Met1- and Lys48-diUb. When it comes to Lys63-diUb, it is true that the 2.5-fold reduction of binding cannot account for the significant decrease in hydrolytic activity, the mechanism of which needs further investigation.

Therefore, in the revised manuscript, the main thread is to elucidate the determinants underlying the Met1 activity of OsOTUB1-cat, while all the figures and conclusions involved in linkage preference of OsOTUB1-cat were removed. You will find that, the revised version was restructured in the following areas: ① the conclusion that OsOTUB1^{cat} prefers Met1-diUb to Lys48-diUb in the fifth paragraph of the part “1. The discovery of plant DUBs that hydrolyze Met1 chains using an efficiently prepared Met1-diUb suicide probe” was removed; ② most part of “3. Binding of proximal Ub facilitates the Met1 type preference of OsOTUB1-cat” was removed, while the last paragraph that structurally compared the binding of proximal Ub by OsOTUB1 with OTULIN was moved to the second paragraph of the part of “Discussion”; ③ the title of the “4. Two motifs in S1' pocket determines Met1 type preference of OsOTUB1” was changed to “3. Two motifs in the S1' pocket determine the Met1 activity of OsOTUB1-cat”; ④ the part illuminating the N-handle motif and C-handle motif with XGY and G sequence mode, respectively, together with the effect of alterations in N-handle and C-handle motifs on the Lys48 activity for OsOTUB1-cat and hOTUB1-cat, were separated apart and formed the part of “4. The N- and C-handle motifs with XGY and G sequence modes, respectively, constitute the Met1-specific motif”; ⑤ the “Figure 4G” that compares Met1 and Lys48 activity of hOTUB1-cat-P87G-SGS, the “Figure 5 Two-step adaptation during Met1-diUb cleavage by OsOTUB1” as well as its following interpretation in the part of “4. Two motifs in S1' pocket determines Met1 type preference activity of OsOTUB1” were removed.

Finally, the inclusion of data demonstrating additional enhancement in OsOTUB1 activity by h/OsUbch5B is very interesting. Although, beyond the scope of the study, is there a reason why the activity was only tested against Lys48 chains? Wouldn't a similar activation be expected for Met1 chains too?

Response:

Thank you for your inspiring question!

Based on your question, we tested the effect of h/OsUBCH5B on the activity of OsOTUB1 against Met1-diUb. The results showed that neither OsUBCH5B nor hUBCH5B affected OsOTUB1's activity against Met1-diUb (Fig. S2E), implying that OsOTUB1 adopts a different way to interact with Met1-diUb and Lys48-diUb, similar to the conclusion demonstrated above that the N-terminal part of OsOTUB1 plays roles in discriminating Met1-diUb and Lys48-diUb (Fig. 1F). Please refer to the last sentence in the part "1. The discovery of plant DUBs that hydrolyse Met1 chains using an efficiently prepared Met1-diUb suicide probe".

REVIEWERS' COMMENTS

Reviewer #3 (Remarks to the Author):

The authors have addressed all my comments.